# The disparities and development trajectories of nations in achieving the sustainable development goals

Fengmei Ma [1,2,3], Heming Wang [2,3,4] ✉, Asaf Tzachor [5,6] ✉, César A. Hidalgo [2,7,8], Heinz Schandl [3,9], Yue Zhang[4], Jingling Zhang[10], Wei-Qiang Chen [1,11] ✉, Yanzhi Zhao[12], Yong-Guan Zhu [1,11,13] & Bojie Fu [13]

The Sustainable Development Goals (SDGs) provide a comprehensive framework for societal progress and planetary health. However, it remains unclear whether universal patterns exist in how nations pursue these goals and whether key development areas are being overlooked. Here, we apply the product space methodology, widely used in development economics, to construct an 'SDG space of nations'. The SDG space models the relative performance and specialization patterns of 166 countries across 96 SDG indicators from 2000 to 2022. Our SDG space reveals a polarized global landscape, characterized by distinct groups of nations, each specializing in specific development indicators. Furthermore, we find that as countries improve their overall SDG scores, they tend to modify their sustainable development trajectories, pursuing different development objectives. Additionally, we identify orphaned SDG indicators − areas where certain country groups remain under-specialized. These patterns, and the SDG space more broadly, provide a high-resolution tool to understand and evaluate the progress and disparities of countries towards achieving the SDGs.

Adopted by United Nations member states in 2015, the 2030 Agenda for Sustainable Development constitutes a comprehensive framework of 17 Sustainable Development Goals (SDGs) and 169 targets to inspire and guide policies for eradicating poverty, protecting planetary ecosystems, and promoting peace and prosperity for humankind[1]. The year of 2023 marked the halfway point in the implementation of the

SDGs[2], prompting us to question whether universal patterns underpinned nations' sustainable development trajectories, and whether some SDG indicators were neglected or overlooked (referred to herein as 'orphaned') in specific areas, across nations, and over time.

Indeed, previous analyses[3–6] have revealed that different countries are performing differently in targets and pursuing alternative

[1]Key Lab of Urban Environment and Health, Institute of Urban Environment, Chinese Academy of Sciences, Xiamen 335400, China. [2]Center for Collective Learning, CIAS, Corvinus University of Budapest, Közraktár u. 4-6, 1093 Budapest, Hungary. [3]Commonwealth Scientific and Industrial Research Organisation (CSIRO), Canberra, ACT 2601, Australia. [4]State Environmental Protection Key Laboratory of Eco-Industry, Northeastern University, Shenyang 110819, China. [5]School of Sustainability, Reichman University (IDC Herzliya), Herzliya 4610101, Israel. [6]Centre for the Study of Existential Risk (CSER), University of Cambridge, Cambridge CB2 1SB, United Kingdom. [7]Center for Collective Learning, IAST, Toulouse School of Economics & Université de Toulouse Capitole, 1 Esp. de l'Université, 31000 Toulouse, France. [8]Alliance Manchester Business School, University of Manchester, Booth St W, Manchester M15 6PB, United Kingdom. [9]Graduate School of Environmental Studies, Nagoya University, Nagoya, Japan. [10]LEREPS - Laboratoire d'Etude et de Recherche sur l'Economie, les Politiques et les Systèmes Sociaux; Institut d'Études Politiques [IEP], Toulouse 31000, France. [11]University of Chinese Academy of Sciences, Beijing 101408, China. [12]Institute of Carbon Neutrality Technology and Policy, Shenyang University, Shenyang 110044, China. [13]State Key Laboratory of Urban and Regional Ecology, Research Center for Eco-Environmental Sciences, Chinese Academy of Sciences, Beijing, China. ✉e-mail: wanghm@mail.neu.edu.cn; atzachor@runi.ac.il; wqchen@iue.ac.cn

sustainable development paths. Rwanda, for instance, has nearly realized SDG 13 (Climate action) while faltering on SDGs 1 and 4 (No poverty and Quality education), while Russia had pursued the opposite sustainable development trajectory, meeting SDGs 1 and 4 while making less progress in SDG 13. China has performed well in SDG 2 (No hunger) yet had lower performance in SDG 14 (Life below water), while Chile has performed poorer on the former than China and better on the latter.

Using historical data, recent studies have also ranked and evaluated countries by their sustainability performance[6–8], explored key development dimensions (e.g., socioeconomic development, environment, and equality)[9–11], and mapped the interactions (synergies and trade-offs) among the SDGs[12–19]. However, the trajectories of countries pursuing SDGs have been poorly explored from a comparative lens, that is, comparing different goals or targets against each other to understand the relative performance or characteristics of each entity in relation to the others. Moreover, it remains unclear whether there are universal patterns underlying the sustainable development trajectories of nations across targets and over time. Specifically, we examined whether countries have so-called 'orphaned' SDG indicators in certain development areas and whether there are underlying patterns or rules governing these 'orphaned' areas.

Uncovering such structural patterns and properties — should they exist — can assist in identifying 'orphaned' targets or areas where progress has been insufficient (not necessarily intentionally or purposefully); forewarning future challenges in realizing SDGs; and, inspiring future development policies at multiple scales (international, regional, and national). Such a nuanced understanding of sustainable development trajectories offers a basis for precise efforts to realize the SDGs, ensuring no area, and consequently no community, is left behind.

In this work, we use the 'product space' method to inform these questions[20]. This method employs network analysis to reveal the relatedness, or affinity, between economies and activities[21,22], and can be used to investigate the clustering and evolution of specialization patterns of regions and nations. For instance, it can help reveal how countries transition from light-manufacturing (e.g., garments) to electronics and which countries tend to specialize in which products. The product space approach, along with the economic complexity index, has proven effective in explaining and anticipating variations in performance across various dimensions and domains[23,24], including industry-, occupation-, research- and technology-spaces[25–29], as well as addressing broader issues like inequality, resource efficiency, and regional sustainability[30–32].

Applying the product space and economic complexity framework, we are able to investigate whether countries' performance follows distinct patterns, and whether countries under-specialize in certain SDG areas. To do so, we devise an 'SDG space' to reveal the sustainable development trajectories of 166 nations for the period 2000–2022. In other words, we consider performance across 96 SDG indicators as types of 'products', and use measures of 'specialization' to quantify their sustainable development trajectories. The revealed comparative advantage (RCA)[33] is used to measure which country specializes in which area. For example, if a country's score for a specific SDG indicator constitutes a higher proportion of its total score across all indicators compared to the world average, that country is considered to be specialized in that SDG indicator. Inspired by the economic complexity index (ECI) and product complexity index (PCI) set out by Hidalgo and Hausmann[23], we use a method equivalent to a clustering algorithm[21,22] to calculate the country sustainability index (CSI) and goal sustainability index (GSI) based on countries' RCA in SDG indicators. The rationale behind the calculation of GSI and CSI indicates that high-CSI countries, are more likely to dominate high-GSI indicators, whereas low-CSI countries tend to dominate low-GSI indicators overall (see Methods).

The indicators covered in this study are carefully indexed and evaluated in the Sustainable Development Report (SDR) 2023[6], with details provided in Tables 1 and 2, including their relationship to the official United Nations Statistics Division (UNSD) SDG targets. For clarity in the figures, the SDG indicator abbreviations used in this study (e.g., 1.A, Poverty headcount ratio at 3.65 US Dollars ($)/day) are simplified versions of the indicators provided in the SDR report. They do not correspond directly to the UNSD SDG targets. National 'SDG spaces' are further quantified for each country, including data on SDG indicator clusters and RCAs (see Supplementary Information). Moreover, we have made available the visualization of 3818 SDG spaces for 166 nations spanning the years 2000–2022 on a dedicated website[34].

Our results reveal a polarized global landscape within the SDG space, with countries clustering into distinct groups, each specializing in different development targets. The SDG space shows that as countries improve their overall, absolute, and aggregate SDG scores, they modify their sustainable development trajectories, pursuing different development ends. Additionally, the SDG space emphasizes orphaned SDG indicators—areas under-specialized by certain country groups. The SDG space and these findings provide a country-level tool to understand and evaluate the progress and disparities of countries in achieving the SDGs.

## Results

### Underlying structures of the SDG space of nations

We adopt the concept of 'product space'[20,23] to construct an SDG space (Fig. 1 and Supplementary Fig. 1), representing the sustainability specialization of nations. The global SDG space of nations shows a 'dumbbell' structure with two main tight clusters (Fig. 1a, b and Supplementary Fig. 2), highlighted here in blue (high-GSI indicators) and red (low-GSI indicators). The blue cluster of SDG indicators (more commonly specialized by countries with high SDG scores, see Supplementary Table 1) is located on the right side of the SDG space. It includes indicators mainly related to SDGs 1 (No Poverty), 3–5 (Good Health and Well-Being, Quality Education, Gender Equality), 9–11 (Industry, Innovation and Infrastructure, Reduced Inequalities, Sustainable Cities and Communities), and 16 (Peace, Justice and Strong Institutions). The red cluster of SDG indicators (mainly specialized by countries with low SDG scores, see Supplementary Table 1) is on the left side of the SDG space. It includes indicators mainly related to SDGs 12 (Responsible Consumption and Production) and 13 (Climate Action). The indicators in these two clusters have relatively higher node degrees, indicating that they are connected to a larger number of edges (Fig. 1a).

Notably, countries at different stages can have different specialisation patterns for specific SDGs. Countries with high SDG scores tend to have relative advantages in indicators related to poverty reduction (1.A), hunger reduction (2.E, 2.G), good health and well-being (3.B, 3.D, 3.F, 3.G, 3.I, 3.K, 3.M), wastewater and air pollution treatment (3.H, 6.D, 14.B), access to clean energy and water (6.B, 6.E, 7.A), industry, innovation and infrastructure (Goal 9), as well as government administration (16.B, 16.C, 16.I, 17.B, 17.C). In contrast, countries with low SDG scores show relative advantages in achieving SDG indicators related to overnutrition (2.B, 2.F), embodied social and environmental impacts in international trade (6.C, 8.C, 8.D, 12.D, 12.F, 13.C, 14.A, 15.A, 16.A), resource use, waste and emissions (6.A, 7.B, 12.A, 12.C, 12.E, 12.G, 13.B), and issues related to corporate tax havens (17.A).

The 'dumbbell' structure of the SDG space reaffirms the unevenness in global sustainable development that has been previously observed[3]; however, from a dynamic evolutionary perspective, it is encouraging to notice that the trend of unevenness is weakening, as shown in Supplementary Figs. 3–4. In addition to the global SDG space, we constructed a 'country space' based on countries' performance on SDG indicators (see Methods on how to build the country space and

 

**Table 1 | Environment-related SDG indicators ranked by the Goal Sustainability Index (GSI) and their alignment with the SDG taxonomies of United Nations Statistics Division (UNSD)**

| Group | GSI in 2022 | Indicator code in this study | Indicator code in SDR 2023 | Indicator definitions | UNSD target | UNSD match |
|---|---|---|---|---|---|---|
| 1 | 1.79 | 6.D | sdg6_wastewat | Anthropogenic wastewater that receives treatment | 6.3.1 | Match |
| | 1.08 | 7.A | sdg7_cleanfuel | Population with access to clean fuels and technology for cooking | 7.1.2 | Closely aligned |
| | 0.83 | 3.H | sdg3_pollmort | Age-standardized death rate attributable to household air pollution and ambient air pollution | 3.9.1 | Match |
| | 0.79 | 14.B | sdg14_cleanwat | Ocean Health Index: Clean Waters score | 14.1.1 | Closely aligned |
| 2 | 0.27 | 14.C | sdg14_cpma | Mean area that is protected in marine sites important to biodiversity | 14.5.1 | Closely aligned |
| | 0.27 | 2.C | sdg2_pestexp | Exports of hazardous pesticides | 3.9 | Closely aligned |
| | 0.19 | 15.D | sdg15_forchg | Permanent deforestation | 15.2 | Closely aligned |
| | 0.05 | 7.D | sdg7_renewcon | Renewable energy share in total final energy consumption | 7.2.1 | Match |
| | −0.03 | 11.B | sdg11_pm25 | Annual mean concentration of particulate matter of less than 2.5 microns in diameter (PM2.5) | 11.6.2 | Match |
| | −0.21 | 14.E | sdg14_fishstocks | Fish caught from overexploited or collapsed stocks | 14.4.1 | Closely aligned |
| | −0.27 | 15.C | sdg15_cpta | Mean area that is protected in terrestrial sites important to biodiversity | 15.1.2 | Match |
| | −0.30 | 15.B | sdg15_cpfa | Mean area that is protected in freshwater sites important to biodiversity | 15.1.2 | Match |
| | −0.34 | 14.D | sdg14_discard | Fish caught that are then discarded | 14.4 | Closely aligned |
| | −0.34 | 14.F | sdg14_trawl | Fish caught by trawling or dredging | 14.4 | Closely aligned |
| | −0.57 | 15.E | sdg15_redlist | Red List Index of species survival | 15.5.1 | Match |
| | −0.65 | 2.D | sdg2_snmi | Sustainable Nitrogen Management Index | 2.4 | Closely aligned |
| | −0.72 | 13.A | sdg13_co2export | CO2 emissions embodied in fossil fuel exports | 13.2 | Closely aligned |
| | −0.82 | 6.A | sdg6_freshwat | Freshwater withdrawal | 6.4.2 | Match |
| | −0.95 | 7.B | sdg7_co2twh | CO2 emissions from fuel combustion per total electricity output | 7.2 | Closely aligned |
| 3 | −1.26 | 12.B | sdg12_explastic | Exports of plastic waste | 12.4 | Closely aligned |
| | −1.40 | 14.A | sdg14_biomar | Marine biodiversity threats embodied in imports | 14.4 | Closely aligned |
| | −1.54 | 12.E | sdg12_nprod | Production-based nitrogen emissions | 9.4 | Closely aligned |
| | −1.59 | 12.D | sdg12_nimport | Nitrogen emissions embodied in imports | 9.4 | Closely aligned |
| | −1.59 | 13.C | sdg13_co2import | CO2 emissions embodied in imports | 13.2 | Closely aligned |
| | −1.62 | 12.A | sdg12_ewaste | Electronic waste | 12.4.2 | Match |
| | −1.64 | 15.A | sdg15_biofrwter | Terrestrial and freshwater biodiversity threats embodied in imports | 15.5 | Closely aligned |
| | −1.66 | 12.F | sdg12_so2import | SO2 emissions embodied in imports | 9.4 | Closely aligned |
| | −1.67 | 6.C | sdg6_scarcew | Scarce water consumption embodied in imports | 6.4 | Closely aligned |
| | −1.67 | 12.C | sdg12_msw | Municipal solid waste | 12.5 | Closely aligned |
| | −1.76 | 13.B | sdg13_co2gcp | CO2 emissions from fossil fuel combustion and cement production | 13.2.2 | Closely aligned |
| | −1.84 | 12.G | sdg12_so2prod | Production-based SO2 emissions | 9.4 | Closely aligned |

The environment-related indicators are divided into three groups based on their GSI values. Group 1: *GSI* > 0.5. Group 2: −1.0 ≤ *GSI* ≤ 0.5. Group 3: *GSI* < −1.0. To simplify the representation of individual indicator names, we have labeled the indicators for each goal in alphabetical order, such as 1.A and 1.B.

Supplementary Fig. 5), which provides more detailed insights into the world's unevenness in sustainable development.

Figure 2 presents strong regional specialization patterns within the SDG space. The SDG indicators with *RCA* > 0 for each region are color-coded, indicating that the region has a relative advantage in these indicators. Africa occupies the red cluster on the left side of the SDG space (Fig. 2a), primarily associated with overnutrition, embodied social and environmental impacts in international trade, and waste and emissions. In contrast, Europe occupies the blue cluster on the right side (Fig. 2d), which includes indicators related to poverty and hunger reduction, good health and well-being, education, innovation, and government administration. The Americas, Asia, and Oceania regions exhibit relatively similar patterns (Fig. 2b, c, e), with a more even distribution and smaller RCA values for their specialized indicators. However, Oceania has more advantages in high-GSI indicators within the blue cluster. The SDG space of all 166 nations ranked by SDG scores in 2000, 2015, and 2022, are shown in Supplementary Figs. 12–177. It is notable that the findings and insights derived from this method may not be universally applicable to all countries in each region, given the local variations in socio-economic conditions, culture, governance, and policy environments.

## Sustainable development paths of nations

Through a moving-window technique[18,35] (see Methods), Fig. 3 reveals a notable pattern: as countries progress in their development, measured in their absolute and aggregate SDG scores, their trajectories tend to manifest an X-shaped pattern over time (Fig. 3a–c, and detailed results in Supplementary Tables 2–13). With improvement in countries' aggregate SDG scores, indicators positioned at the top and bottom of the y-axis start to diverge, moving in opposite directions and sequentially crossing at around the 80th and 40th window groups of countries.

The dynamics of RCA values of SDG indicators is mainly characterized by three trends. The first trend is characterized by a continuous decrease in specialization in low-GSI indicators represented in red in Fig. 3a–c. These indicators mainly represent issues related to the overnutrition, embodied social and environmental impacts in trade, and waste and emissions. Specifically, Fig. 4 referring to the year 2022, associates this trend with indicators 2.B, 2.C, and 2.F in Goal 2 (No Hunger), 12.A-G in Goal 12 (Responsible Consumption and Production), 13.A-C in Goal 13 (Climate Action), 14.A and 14.D-F in Goal 14 (Life Below Water), 15.A in Goal 15 (Life on Land), 16.K in Goal 16 (Peace, Justice and Strong Institutions), and 17.A in Goal 17 (Partnerships for

**Table 2 | Non-environment-related SDG indicators ranked by the Goal Sustainability Index (GSI) and their alignment with UN SDG taxonomies**

| Group | GSI in 2022 | Indicator code in this study | Indicator code in SDR 2023 | Indicator definitions | UNSD target | UNSD match |
|---|---|---|---|---|---|---|
| 4 | 1.98 | 9.E | sdg9_rdex | Expenditure on research and development | 9.5.1 | Match |
| | 1.87 | 9.A | sdg9_articles | Articles published in academic journals | 9.5 | Closely aligned |
| | 1.30 | 3.G | sdg3_neonat | Neonatal mortality rate | 3.2.2 | Match |
| | 1.29 | 3.M | sdg3_uhc | Universal health coverage (UHC) index of service coverage | 3.8.1 | Match |
| | 1.23 | 8.A | sdg8_accounts | Adults with an account at a bank or other financial institution or with a mobile-money-service provider | 8.10.2 | Match |
| | 1.17 | 11.C | sdg11_slums | Proportion of urban population living in slums | 11.1.1 | Match |
| | 1.08 | 9.G | sdg9_uni | The Times Higher Education Universities Ranking: Average score of top 3 universities | – | Not in UNSTATS |
| | 1.08 | 3.I | sdg3_swb | Subjective well-being | 3.4 | Closely aligned |
| | 1.08 | 9.C | sdg9_lpi | Logistics Performance Index: Quality of trade and transport-related infrastructure | 9.1 | Closely aligned |
| | 1.07 | 16.C | sdg16_cpi | Corruption Perceptions Index | 16.5.1, 16.5.2 | Closely aligned |
| | 1.06 | 3.K | sdg3_traffic | Traffic deaths | 3.6.1 | Match |
| | 1.05 | 1.A | sdg1_lmicpov | Poverty headcount ratio at 3.65 US Dollars ($)/day | 1.1.1 | Match |
| | 1.04 | 4.D | sdg4_second | Lower secondary completion rate | 4.1.2 | Match |
| | 1.03 | 3.D | sdg3_lifee | Life expectancy at birth | 3.1:3.9 | Closely aligned |
| | 1.02 | 9.B | sdg9_intuse | Population using the internet | 17.8.1 | Match |
| | 1.01 | 9.F | sdg9_roads | Rural population with access to all-season roads | 9.1.1 | Match |
| | 1.01 | 4.A | sdg4_earlyedu | Participation rate in pre-primary organized learning | 4.2.2 | Closely aligned |
| | 0.98 | 3.B | sdg3_fertility | Adolescent fertility rate | 3.7.2 | Match |
| | 0.97 | 6.B | sdg6_sanita | Population using at least basic sanitation services | 6.2.1 | Closely aligned |
| | 0.95 | 2.E | sdg2_stunting | Prevalence of stunting in children under 5 years of age | 2.2.1 | Match |
| | 0.94 | 9.D | sdg9_mobuse | Mobile broadband subscriptions | 9.c.1, 17.6.1 | Closely aligned |
| | 0.91 | 2.G | sdg2_undernsh | Prevalence of undernourishment | 2.1.1 | Match |
| | 0.84 | 17.C | sdg17_statperf | Statistical Performance Index | 17.18.1: 17.19.2 | Closely aligned |
| | 0.81 | 16.E | sdg16_exprop | Expropriations are lawful and adequately compensated | 16.6 | Closely aligned |
| | 0.75 | 3.F | sdg3_ncds | Age-standardized death rate due to cardiovascular disease, cancer, diabetes, or chronic respiratory disease in adults aged 30–70 years | 3.4.1 | Match |
| | 0.66 | 2.A | sdg2_crlyld | Cereal yield | 2.3, 2.4 | Closely aligned |
| | 0.66 | 17.B | sdg17_govex | Government spending on health and education | 1.a.1 | Closely aligned |
| | 0.58 | 16.B | sdg16_clabor | Children involved in child labor | 8.7.1 | Closely aligned |
| | 0.52 | 8.E | sdg8_rights | Fundamental labor rights are effectively guaranteed | 8.8.2 | Match |
| | 0.51 | 16.I | sdg16_safe | Population who feel safe walking alone at night in the city or area where they live | 16.1.4 | Match |
| 5 | 0.47 | 10.B | sdg10_palma | Palma ratio | 10.1 | Closely aligned |
| | 0.47 | 16.D | sdg16_detain | Unsentenced detainees | 16.3.2 | Match |
| | 0.42 | 1.B | sdg1_wpc | Poverty headcount ratio at 2.15 US Dollars ($)/day | 1.1.1 | Match |
| | 0.42 | 5.B | sdg5_familypl | Demand for family planning satisfied by modern methods | 3.7.1 | Match |
| | 0.39 | 16.H | sdg16_rsf | Press Freedom Index | 16.1 | Closely aligned |
| | 0.37 | 16.F | sdg16_homicides | Homicides | 16.1.1 | Match |
| | 0.36 | 16.G | sdg16_justice | Access to and affordability of justice | 16.3.1, 16.3.3 | Closely aligned |
| | 0.30 | 6.E | sdg6_water | Population using at least basic drinking water services | 6.1.1 | Closely aligned |
| | 0.28 | 2.H | sdg2_wasting | Prevalence of wasting in children under 5 years of age | 2.2.2 | Match |
| | 0.26 | 16.A | sdg16_admin | Timeliness of administrative proceedings | 16.6 | Closely aligned |
| | 0.05 | 11.A | sdg11_pipedwat | Access to improved water source, piped | 11.1 | Closely aligned |
| | 0.05 | 3.N | sdg3_vac | Surviving infants who received 2 WHO-recommended vaccines | 3.b.1 | Closely aligned |
| | 0.00 | 8.F | sdg8_slavery | Victims of modern slavery | 8.7 | Closely aligned |
| | −0.01 | 16.J | sdg16_u5reg | Birth registrations with civil authority | 16.9.1 | Match |
| | −0.07 | 3.L | sdg3_u5mort | Mortality rate, under-5 | 3.2.1 | Match |
| | −0.13 | 8.B | sdg8_adjgrowth | Adjusted GDP growth | 8.1.1 | Closely aligned |

**Table 2 (continued) | Non-environment-related SDG indicators ranked by the Goal Sustainability Index (GSI) and their alignment with UN SDG taxonomies**

| Group | GSI in 2022 | Indicator code in this study | Indicator code in SDR 2023 | Indicator definitions | UNSD target | UNSD match |
|---|---|---|---|---|---|---|
| | −0.15 | 7.C | sdg7_elecac | Population with access to electricity | 7.1.1 | Match |
| | −0.15 | 5.A | sdg5_edat | Ratio of female-to-male mean years of education received | 4.5.1 | Closely aligned |
| | −0.15 | 10.A | sdg10_gini | Gini coefficient | 10.1 | Closely aligned |
| | −0.15 | 5.D | sdg5_parl | Seats held by women in national parliament | 5.5.1 | Match |
| | −0.19 | 11.D | sdg11_transport | Satisfaction with public transport | 11.2.1 | Closely aligned |
| | −0.19 | 4.B | sdg4_literacy | Literacy rate | 4.6.1 | Match |
| | −0.27 | 3.C | sdg3_hiv | New HIV infections | 3.3.1 | Match |
| | −0.37 | 3.A | sdg3_births | Births attended by skilled health personnel | 3.1.2 | Match |
| | −0.39 | 3.E | sdg3_matmort | Maternal mortality rate | 3.1.1 | Match |
| | −0.51 | 3.J | sdg3_tb | Incidence of tuberculosis | 3.3.2 | Match |
| | −0.57 | 4.C | sdg4_primary | Net primary enrollment rate | 4.1.2 | Closely aligned |
| | −0.82 | 5.C | sdg5_lfpr | Ratio of female-to-male labor force participation rate | 5.5 | Closely aligned |
| 6 | −1.06 | 8.G | sdg8_unemp | Unemployment rate | 8.5.2 | Match |
| | −1.45 | 17.A | sdg17_cohaven | Corporate Tax Haven Score | – | Not in UNSTATS |
| | −1.60 | 8.D | sdg8_impslav | Victims of modern slavery embodied in imports | 8.7 | Closely aligned |
| | −1.67 | 8.C | sdg8_impacc | Fatal work-related accidents embodied in imports | 8.8.1 | Closely aligned |
| | −1.68 | 2.F | sdg2_trophic | Human Trophic Level | – | Not in UNSTATS |
| | −1.69 | 16.K | sdg16_weaponsexp | Exports of major conventional weapons | 16.1 | Closely aligned |
| | −1.81 | 2.B | sdg2_obesity | Prevalence of obesity, BMI ≥ 30 | 2.2 | Closely aligned |

The non-environment-related indicators are divided into three groups based on their GSI values. Group 4: $GSI > 0.5$. Group 5: $−1.0 \leq GSI \leq 0.5$. Group 6: $GSI < −1.0$. To simplify the representation of individual indicator names, we have labeled the indicators for each goal in alphabetical order, such as 1.A and 1.B.

the Goals). These indicators, predominantly found in the left cluster (red cluster) of the SDG space (Fig. 1), are identified as 'orphaned' indicators, indicating areas where there is significant need for enhanced focus and policy intervention.

The second trend is associated with the early increase in specialization of indicators with a high GSI, represented by light blue in Fig. 3a–c. This trend primarily focuses on the areas of poverty eradication, basic health, basic education, and livelihood security. Specifically, in the year 2022, as illustrated in Fig. 4, this trend includes indicators 1.A and 1.B in Goal 1 (No Poverty), 2.A and 2.E in Goal 2 (No Hunger), 3.B, 3.G, 3.I, 3.K, and 3.M in Goal 3 (Good Health and Well-Being), 4.A and 4.D in Goal 4 (Quality Education), 6.B and 6.E in Goal 6 (Clean Water and Sanitation), 7.A and 7.C in Goal 7 (Affordable and Clean Energy), and 8.A in Goal 8 (Decent Work and Economic Growth). This upward trend in specialization on high-GSI indicators reflects a prioritization of foundational aspects of sustainable development.

The third trend pertains to the follow-up increasing specialization in objectives that have been identified as 'deep blue indicators' distinguished by their higher GSI values as depicted in Fig. 3a–c. They are mainly related to wastewater treatment, labor rights, and research & development, such as indicators 6.D in Goal 6 (Clean Water and Sanitation), 8.E in Goal 8 (Decent Work and Economic Growth), and 9.A and 9.E-F in Goal 9 (Industry, Innovation and Infrastructure) in 2022 (Fig. 4). These indicators predominantly fall within the right cluster (blue cluster with high GSI values) of the SDG space (Fig. 1). From a dynamic analysis spanning the years 2000 to 2022, it is observed that while the first two trends have remained relatively stable, the third trend - particularly the emphasis on goals related to innovation - has advanced, appearing in earlier window groups.

In exploring the evolutionary patterns underpinning sustainable development trajectories, the product space and economic complexity approach allow to divide SDG indicators into nations' orphaned indicators (red ones with low GSI values) and specialized indicators (blue ones with high GSI values). To identify more refined

trends, we categorize the SDG indicators into six groups based on their GSI values (Fig. 5): three groups of environment-related indicators (Table 1) and three groups of non-environment-related indicators (Table 2). Our findings reveal that environment-related indicators with high GSI values primarily involve wastewater treatment, air pollution control, and access to clean energy, which tend to be specialized by countries with high SDG scores. On the other hand, indicators with low GSI values, such as trade-related pollution and emissions, as well as waste and emissions from production and living, are more commonly associated with countries with low SDG scores.

Similarly, non-environment-related indicators with high GSI values predominantly include no poverty, zero hunger, good health and well-being, quality education, industry, innovation and infrastructure, and effective government administration. In contrast, those with low GSI values mainly involve overnutrition and the social impacts embodied in international trade.

For both environment-related and non-environment-related SDG indicators, the RCA values for higher GSI indicators increase with the growth of the national SDG index, while the RCA values for lower GSI indicators tend to decrease as the national SDG index rises.

However, when analyzing absolute SDG scores outside the comparative lens, we observe an ambiguous convergence between two clusters of indicators (Fig. 3d–f and Supplementary Fig. 6). This ambiguous convergence presents a challenge in discerning the relative advantages and disadvantages of countries at various stages of sustainable development, particularly when the scores of distinct indicators are only narrowly separated. For example, in 2022, indicator 16.A (Timeliness of administrative proceedings) and indicator 12.C (Municipal solid waste) in the 1st window group of countries both registered an average SDG score of 65 (Supplementary Table 10). Despite their identical scores, a comparative analysis reveals a stark contrast in their global standings, with their average RCA rankings positioned at 10 and 90, respectively, out of 96 SDG indicators. This disparity underscores the value of comparative analysis in unveiling the relative strengths

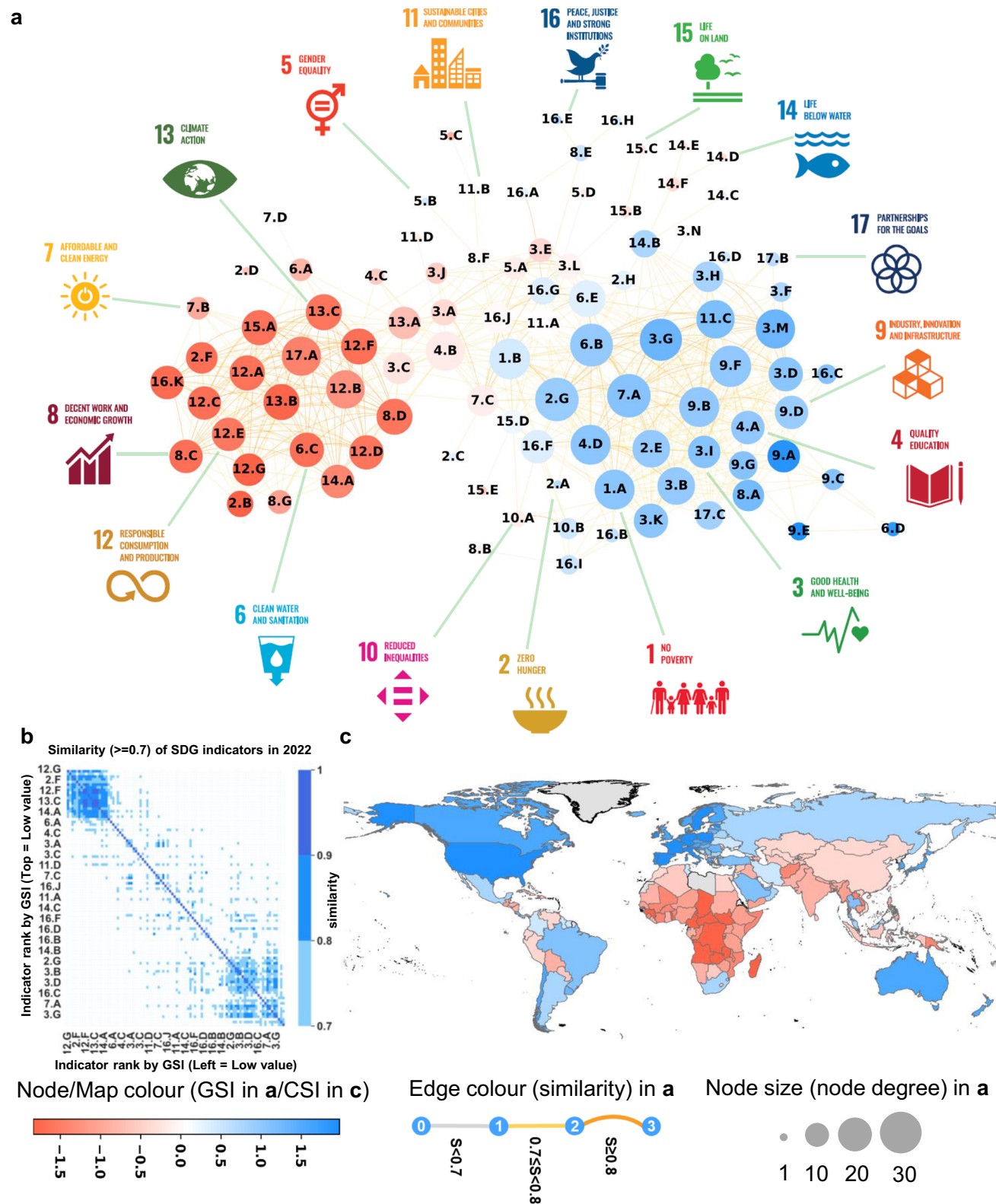

**b** Similarity (>=0.7) of SDG indicators in 2022

**Node/Map colour (GSI in a/CSI in c)**

**Edge colour (similarity) in a**

**Node size (node degree) in a**

and weaknesses of nations on their path of designing future sustainability strategies.

## 'Orphaned' indicators in the SDGs

To examine 'orphaned' indicators that have received insufficient attention among nations across different development stages, we grouped countries into quartiles based on their SDG score rankings (Supplementary Table 1): Stage 1 (Primary stage), Stage 2 (Low medium stage), Stage 3 (High medium stage), and Stage 4 (Advanced stage). Figure 6a–d show that when countries progress from Stage 1 to Stage 4, they shift their relative disadvantages (weaknesses, measured in low RCA with red color) from the right cluster (indicators with low-GSI values), mainly poverty reduction, education, and innovation goals, to the left cluster (indicators with high-GSI values), mainly environmental and climate goals. The move of relative advantages (measured in high RCA with blue color) is the opposite.

**Fig. 1 | The SDG space in 2022.** Panel **a**, Network representation of the SDG space. The node color represents goal sustainability index (GSI), the edge color and shape represent the similarity between two indicators regarding revealed comparative advantage (RCA), and the node size is the node degree, representing the number of edges connected to the node. Panel **b**, The heat map of similarity (> = 0.7) between SDG indicators regarding RCA. Panel **c**, The country sustainability index (CSI) of nations. The map color represents CSI. In Panel **a**, the proximity between SDG indicators in the nation-indicator bipartite network is defined by the conditional probability that two indicators are co-specialized within a nation. RCA is used to assess which country specializes or has relative advantage in which area. If a country $c$ has a higher share of SDG indicator $i$ than the world average, then

$RCA_{c,i} > 0$, which means that this country is considered to have relative advantage in SDG indicator $i$. GSI and CSI are twin clustering indicators, which are calculated based on RCA of indicators using the method developed by Hidalgo and Hausmann[23] (see Methods). The algorithm is equivalent to finding the eigenvalues of a matrix and is related to a spectral clustering algorithm that divides SDG indicators (or countries) into two groups: those with higher and lower GSI values (or countries with higher and lower CSI values). As a result of this reflections algorithm, countries with high CSI values tend to dominate the high-GSI indicators, and vice versa. Most high-CSI countries (blue background countries in **c**) have high SDG scores, and most low-CSI countries have low SDG scores (Supplementary Table 1).

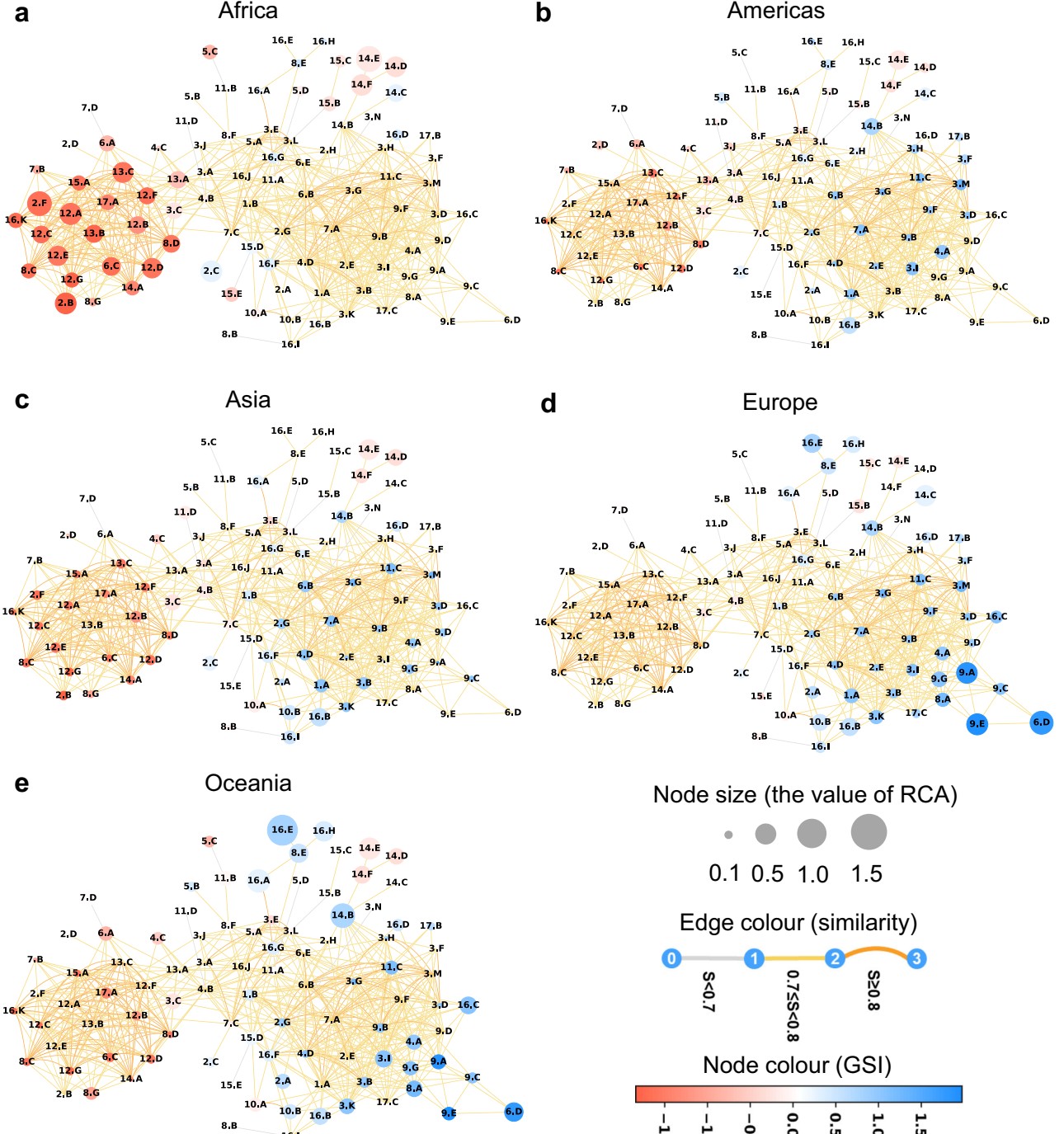

**Fig. 2 | The SDG space of different regions of the world in 2022.** Panel **a**, Africa. Panel **b**, Americas. Panel **c**, Asia. Panel **d**, Europe. Panel **e**: Oceania. The node color represents the Goal Sustainability Index (GSI). The colored SDG indicators are those

with revealed comparative advantage (RCA) > 0, indicating that the region has a revealed comparative advantage (or specialization) in these indicators. The node size corresponds to the RCA value for each indicator.

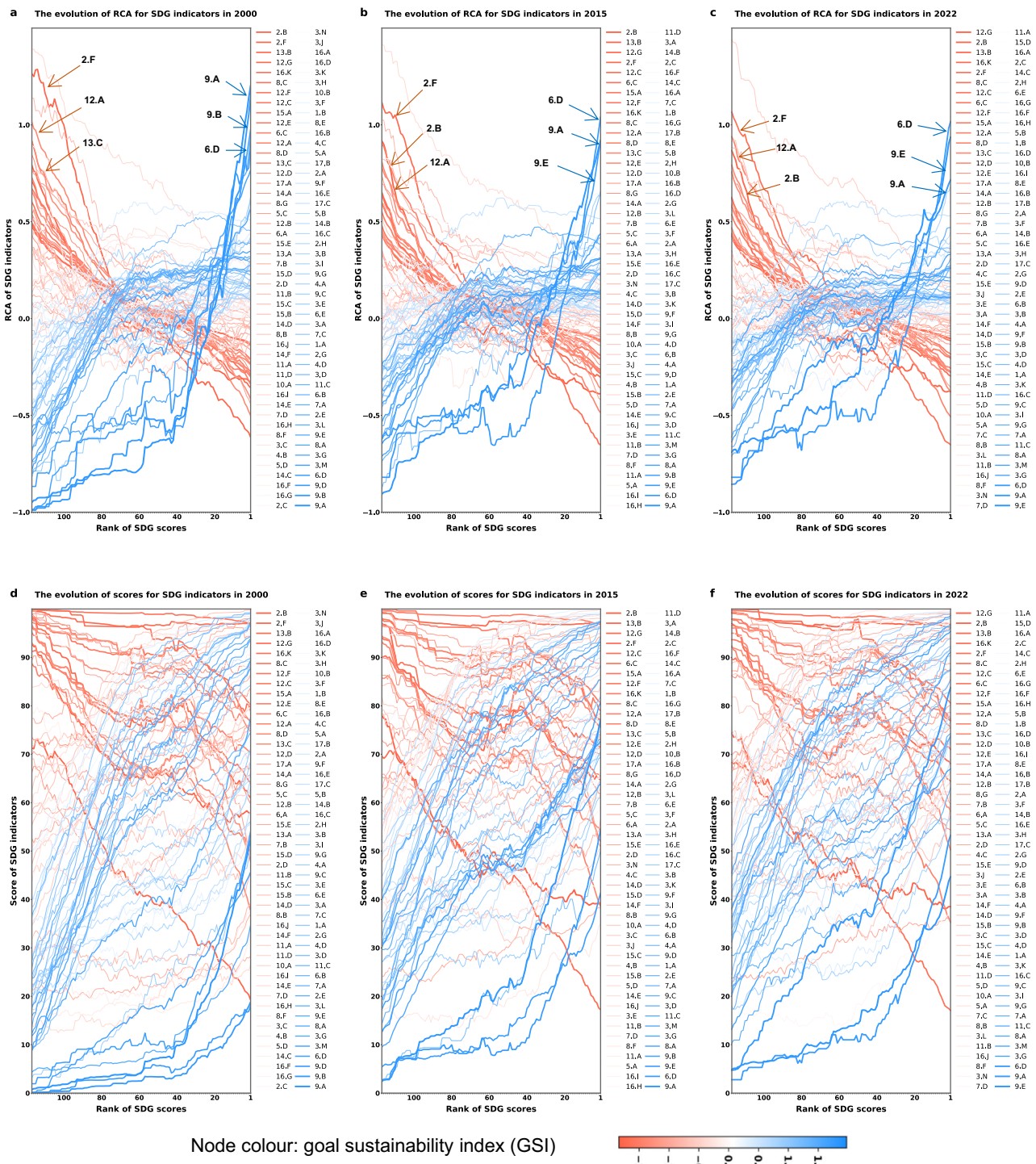

**Fig. 3 | The changing of RCA and absolute score of SDG indicators for each window group alongside the rank of SDG scores in 2000, 2015 and 2022.** Panels **a**–**c**, The changing of SDG indicators in the average of revealed comparative advantage (RCA) for each window group of countries. Panels **d**–**f**, The changing of SDG indicators in the average of absolute scores for each window group of countries. The indicator color ranging from red to blue, indicates a small to large value of goal sustainability index (GSI) of each indicator (see Methods). To discern potential evolutionary patterns underlying the trajectories of sustainable development and elucidate the 'dumbbell' structure observed within the SDG space, we use a moving-window technique[18,35]. This method allows us mitigating short-term fluctuations

and accentuating the long-term dynamics of nations' RCA over time. Direct observation, without this smoothing, renders the evolutionary patterns of countries' pursuit to achieve the SDG objectives ambiguous as evidenced by Supplementary Fig. 6. The issue stems from the fact that countries may exhibit divergent performances despite being at comparable stages of sustainable development. To address this, we determined the window size by grouping 50 countries at a time, resulting in 117 window groups (i.e., countries 1–50, 2–51, …, 117–166, with countries sorted from lowest to highest SDG rank). Please see Methods and the sensitivity analysis in Supplementary Fig. 7.

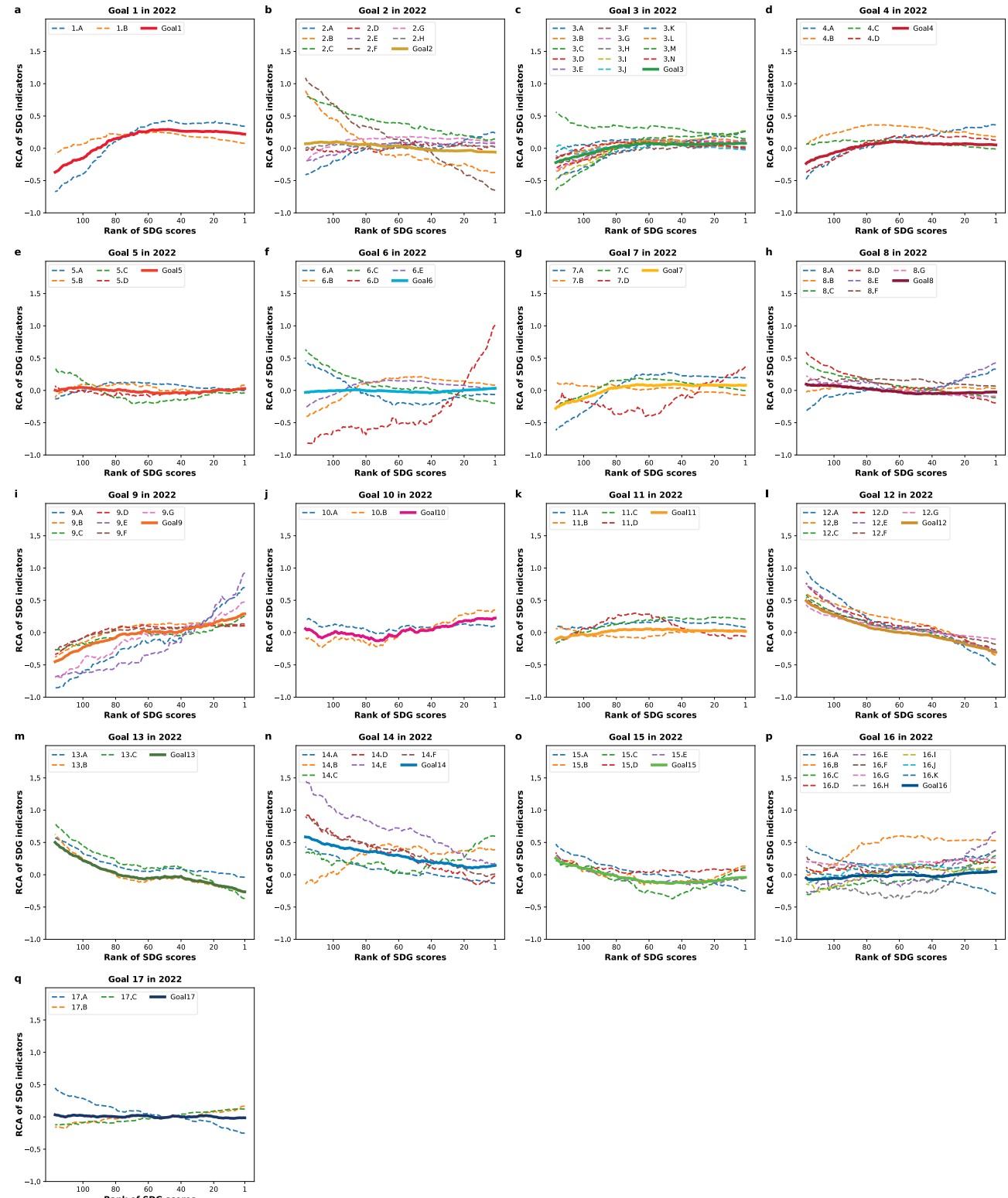

**Fig. 4 | The changing of RCA of 96 SDG indicators divided by their 17 respective goals for each window group alongside the rank of SDG scores in 2022.** Panels **a**–**q**, The change of RCA of SDG indicators across 17 goals for each window group, aligned with the rank of SDG scores in 2022. The x-axis of each figure is the rank of SDG scores, and the y-axis is the average values of revealed comparative advantage (RCA) for each window group of countries. RCA is used to assess which country specializes in which area (see Methods). We examine the impact of countries' growing SDG rankings on the RCA values using a moving-window method, and set the window size by choosing 50 countries as a country group to yield 117 window groups (i.e., countries 1–50, 2–51, ..., 117–166) (see Methods).

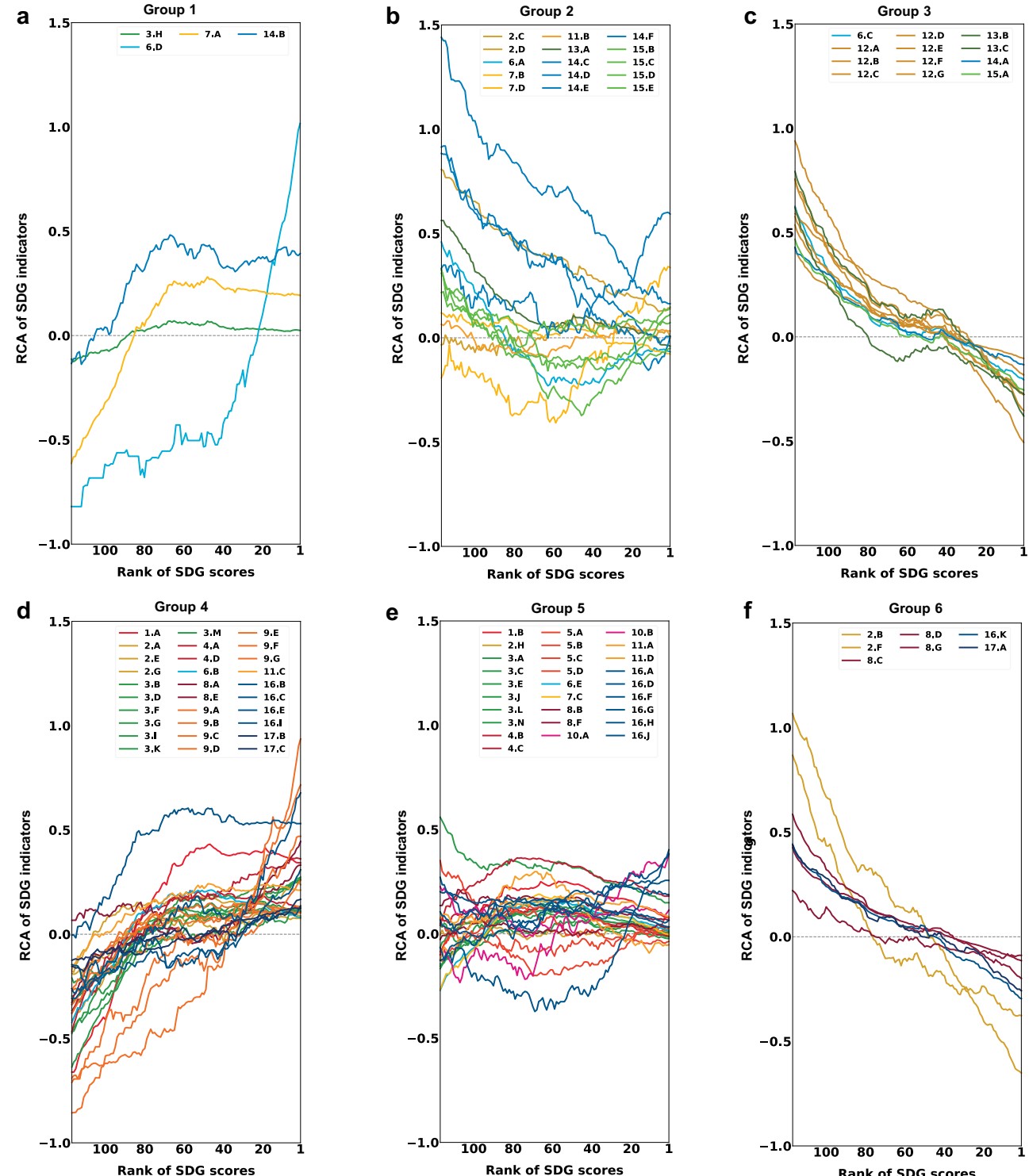

**Fig. 5 | The change in revealed comparative advantage (RCA) for six groups of SDG indicators across window groups ranked by SDG scores in 2022.** Panels **a**–**c**, show the condition for three groups of environment-related indicators, categorized by the goal sustainability index (GSI). **a** Group 1, *GSI* > 0.5; **b** Group 2, −1.0 ≤ *GSI* ≤ 0.5; **c** Group 3, *GSI* < −1.0. Panels **d**–**f**, display the conditions for three groups of non-environment-related indicators. **d** Group 4, *GSI* > 0.5; **e** Group 5, −1.0 ≤ *GSI* ≤ 0.5; **f** Group 6, *GSI* < −1.0. Note: the window size was determined by grouping 50 countries at a time, resulting in 117 window groups (i.e., countries 1–50, 2–51, …, 117–166).

The 'orphaned' indicators in the SDGs are concentrated in the left side of Fig. 6e, where the values of GSI for indicators are low. Specifically, when a country develops from Stage 1 to Stage 2, it loses its comparative advantage in the areas of overnutrition (e.g., 2.B, 2.F) and environmental impacts (e.g., 6.C, 12.A, 13.C). Countries which progress from Stage 2 to Stage 3 probably place a lesser emphasis on the issues of overnutrition (2.F) and fish caught related indicators (14.D, 14.E, 14.F). When countries develop from Stage 3 to Stage 4, they lose further specialization in overnutrition (2.F) and waste and emissions (12.A, 13.C). These easily overlooked indicators across stage changing, are important forewarning indicators for the formulation of sustainable development policies. On the other side, the high-GSI-value indicators

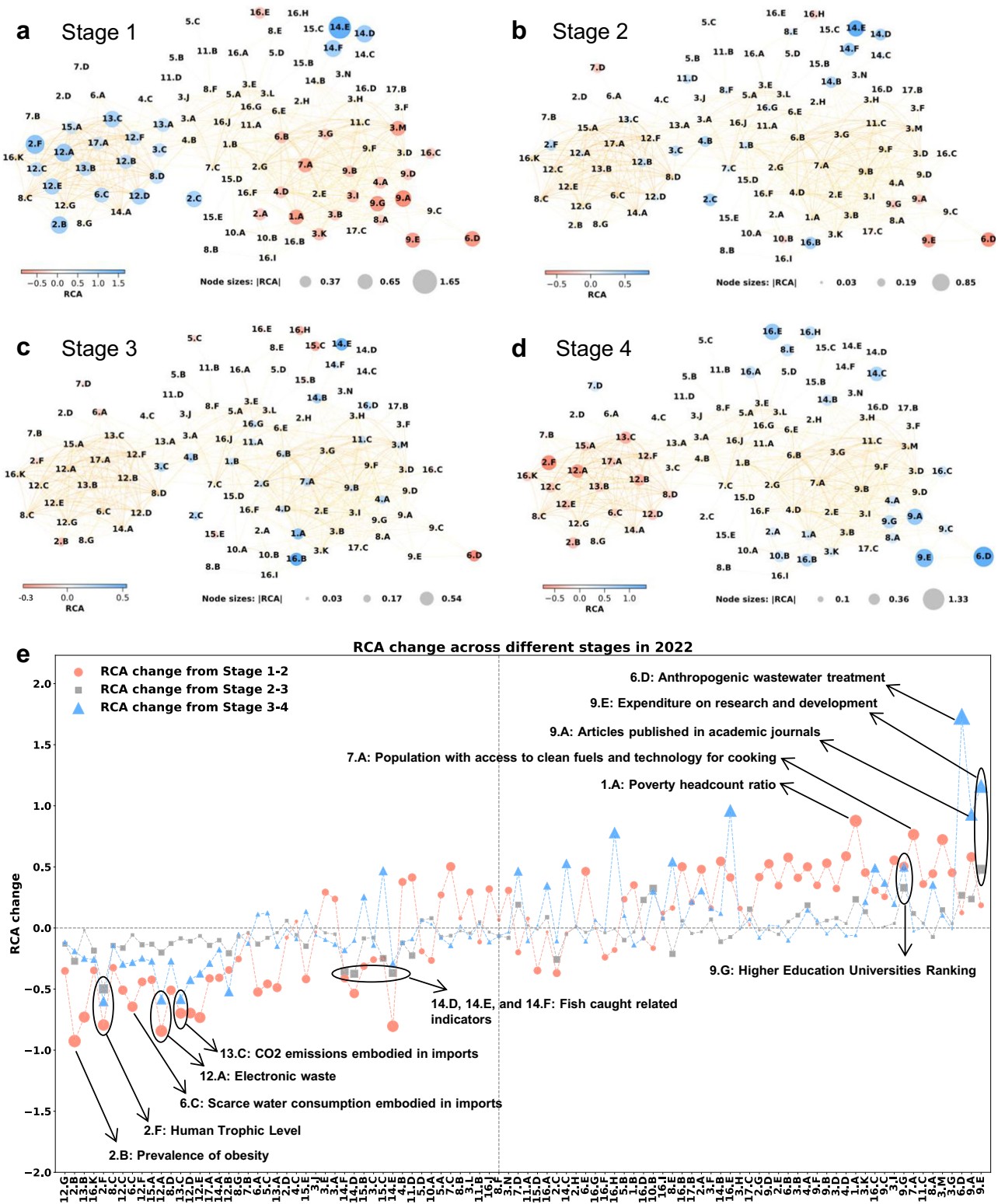

**Fig. 6 | The SDG space of four-stage country groups in 2022.** Countries are evenly divided into four stages according to their SDG scores: Panel **a**, Stage 1 (least developed overall) = Primary stage. Panel **b**, Stage 2 = Low medium stage. Panel **c**, Stage 3 = High medium stage. Panel **d**, Stage 4 (most developed overall) = Advanced stage. Panel **e**, The changes in RCA between stages. In panels **a**–**d**, the blue nodes represent the top 20 SDG indicators in values of revealed comparative advantage (RCA), implying comparative advantages (see Methods), and the red nodes represent the bottom 20 SDG indicators in values of RCA, implying comparative disadvantages. The node size represents the absolute value of RCA in each country group. When designing sustainable development strategies, countries can set the number of highlight indicators in the SDG space according to their needs (e.g., Top 10, 20, 30 indicators, etc.). In **e**, the node size represents the absolute value of RCA change for each country group.

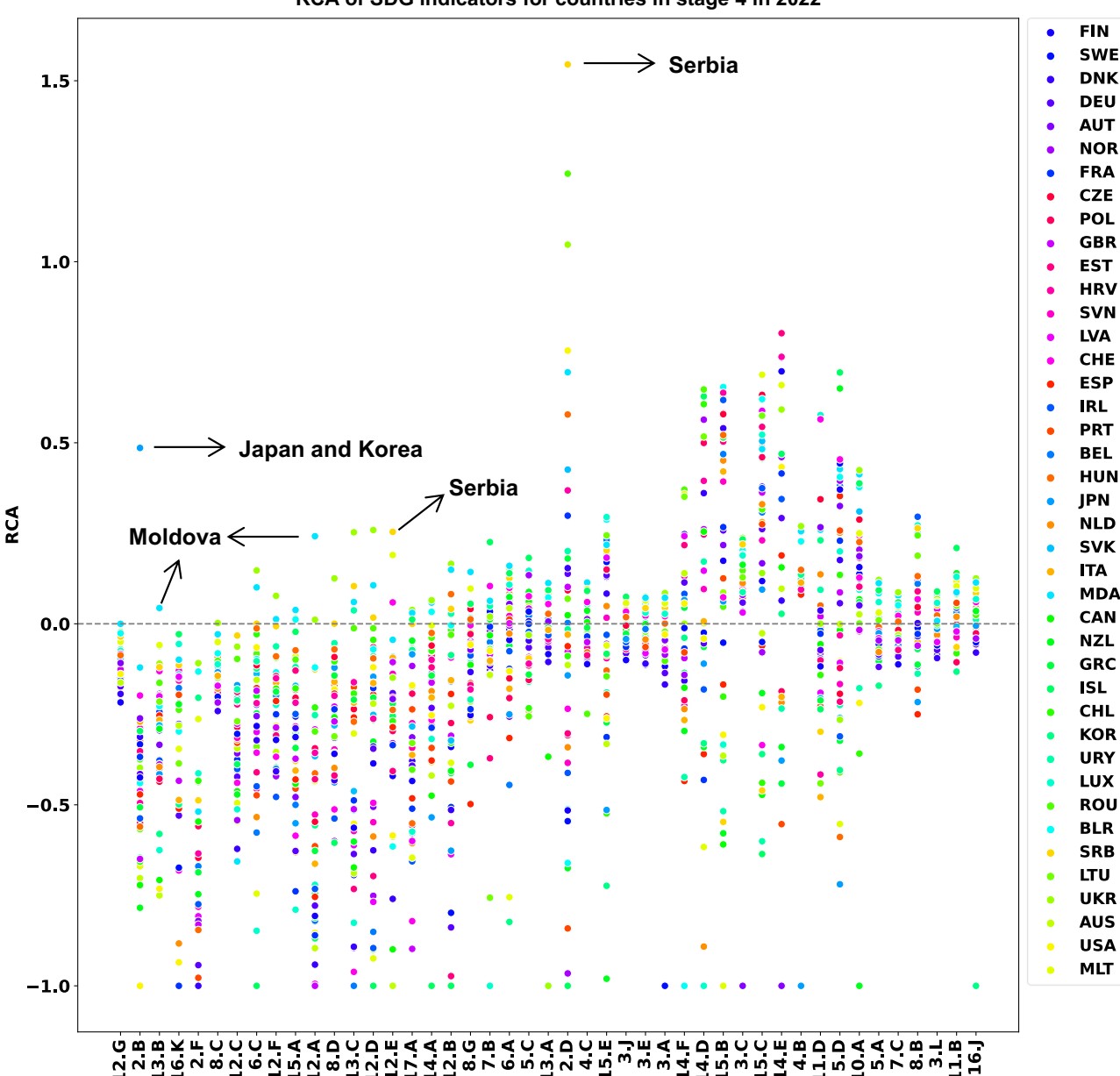

**Fig. 7 | The performance of countries at Stage 4 in 2022.** RCA is revealed comparative advantage, which is used to assess which country specializes or has relative advantage in which area. Only SDG indicators with goal sustainability index (GSI) values lower than 0 are presented, and they are ranked by GSI from low value to high value in the x-axe. These low-GSI indicators are mainly related to overnutrition, environment, and climate goals, in which countries with low SDG scores normally have relative advantages.

on the right side of Fig. 6e, are what countries specialize in when they progress to higher sustainable development stage. These indicators are mainly related to poverty eradication (1.A) and livelihood security (7.A) for Stage 1, scientific research (9.E, 9.G) for Stage 2, and environmental protection (6.D) and expenditure on research and development (9.E) for Stage 3.

The GSI indicator, derived from the product space and economic complexity approach, categorizes SDG indicators that countries tend to over- or under-specialize in during their development process. This categorization can help inform the formulation of sustainable development policies. The details of changes in RCA between stages in 2000, 2015 and 2022 are presented in Supplementary Tables 14–22.

Can countries avoid losing specialization and creating orphan SDG indicators when they progress to higher stages? For the most part no. This is because of both, the way in which SDGs are defined (with goals that are sometimes opposites, such as over- and undernutrition)[18,36], and because measures of specialization, such as RCA are relative measures, meaning that improvements in the score of one indicator must decrease the specialization in other indicators. Figure 7 illustrates this strong trade-off. Still, there are a few exceptions. For example, Serbia specializes in indicators of 2.D (Sustainable nitrogen management) and 12.E (Production-based nitrogen emissions), Japan and Korea perform very well in 2.B (Prevalence of obesity), and Moldova has relative advantages in 12.A (Electronic waste) and 13.B (CO2 emissions from fossil fuel combustion and cement

production). These anomalies can be attributed to distinctive industrial structures (i.e., the composition and organization of industries within a country's economy) or national cultures. For instance, Serbia's nitrogen management and Moldova's $CO_2$ emissions from fossil fuel combustion and cement production are outcomes of their specific industrial frameworks. Similarly, Japan's traditional dietary practices and culinary habits (i.e., its food culture) influence its unique sustainable development indicators.

## Discussion

On the road to achieving the 2030 Agenda for Sustainable Development, monitoring and assessing progress and recognizing sustainable development patterns for countries over time are key priorities for national and international institutions[37]. To this end, understanding each country's relative strengths and weaknesses regarding SDG areas may help design future better indicators and development policies.

This study unveiled a bipolar world, with countries having high SDG scores specialized in poverty reduction, good health and well-being, treatment of wastewater and air pollution, access to clean energy and water, innovation, and government administration and countries with low SDG scores having relative advantages in over-nutrition, embodied social and environmental impacts in international trade, and resource use, waste and emissions. Moreover, at different stages of sustainable development, certain areas – including environmental quality, overnutrition, and impacts of international trade – are left under-specialized by different countries. This bipolar world revealed by the SDG space is supported by previous studies[38–40] that have identified global disparities in SDGs, and it can be partly explained by the trade-offs between SDGs[12,18,41–44]. For instance, the SDGs related to economic development and basic livelihood security (e.g., SDGs 1, 2, 3, 4, 7, 8, 11), are often associated with higher environmental and resource footprints[45–48] and increased greenhouse gas emissions[49], which can hinder progress on SDGs 12 and 13[50]. These trends align with the concept of the environmental Kuznets curve[51,52], which suggests that when countries have a lower GDP per capita, the pursuit of economic growth tends to take precedence over environmental preservation concerns.

Additionally, scientists have warned that global increases in affluence have consistently driven resource use and pollutant emissions to rise more rapidly than technological improvements have been able to mitigate. These impacts are primarily driven by affluent citizens[49]. Therefore, it is imperative to achieve absolute decoupling of economic growth from resource consumption and pollutant emissions through various approaches, such as technological advancements, shifts in consumption patterns, and the adoption of more effective policies[53].

Distinct from earlier notable studies that assessed countries' SDG performance[3–8], such as those ranking countries and evaluating their progress towards achieving the SDGs[6] or exploring development patterns[12–19], our study introduces several key features that set it apart.

First, we employed a comparative framework to identify each country's relative strengths and weaknesses, as well as to pinpoint 'orphaned' SDG indicators in specific development areas. This comparative analysis, which contextualizes performance by benchmarking different variables or entities against each other, offers a clearer understanding of each country's relative standing. In contrast, absolute scoring (Fig. 3d–f) can obscure these insights by focusing solely on an overarching SDG score without the nuanced understanding that comes from comparison with other entities and SDGs.

Second, we utilized the 'product space' method and the broader economic complexity approach as effective tools for pattern recognition. These methods help identify national sustainable development trends and reveal whether countries specialize in certain goals. For instance, the derived GSI indicator divides SDG indicators into low-GSI

indicators that countries tend to overlook and high-GSI indicators that countries tend to specialize in during their development process (Figs. 3, 5, and 6). Additionally, we delineated the development trajectories of nations based on the historical performance of all countries, differing from the Sustainable Development Report[6], which evaluates whether a country is on track based on its individual performance.

Third, we introduced the concept of the 'SDG space' as a high-resolution tool to monitor countries' performance across 96 SDG indicators. This tool visualizes countries' relative performance and assists in guiding the formulation of development strategies (Supplementary Figs. 12–177). Moreover, we developed a dedicated website to present 3818 SDG spaces for 166 nations spanning the years 2000 to 2022: http://www.spacelab.team/#/SDGSpace. Unlike the traditional product space[20], which typically ranks countries and guides them towards advanced products (e.g., motor vehicles and electronics), the SDG space is designed to assist countries monitor their progress within the SDG framework, identifying relative strengths and weaknesses, rather than ranking countries.

In terms of policy guidance, the comparative insights and identified trends in sustainable development trajectories can provide targeted recommendations (Fig. 8). From a static perspective, countries' performance on the indicators can be categorized into four areas (Fig. 8b, d, f, h): Area I (high RCA and SDG Scores) represents an ideal situation. Area II (high RCA but low SDG scores) where indicators are relatively difficult for a specific country to improve, as other countries are also likely to struggle in making progress and providing successful examples. Area III (low RCA and SDG Scores) represents the worst situation, where both relative and absolute performance on these indicators is poor. Area IV (low RCA but relatively high SDG scores) where indicators may be easier to improve, as other countries might be performing better and could offer models for success.

From a dynamic viewpoint, the revealed patterns of trajectories can help forewarn potential challenges for countries (Fig. 8a, c, e, g, Fig. 6e and Supplementary Tables 18, 20, and 22). For Ethiopia, a representative country at Stage 1 (Primary stage), priority should be given to addressing indicators with the lowest SDG scores and Revealed Comparative Advantage (RCA) in Area III (Fig. 8b), such as 6.B (Population using at least basic sanitation services) and 6.D (Anthropogenic wastewater that receives treatment). For India, in addition to the indicators in Area III (Fig. 8d), attention should be directed towards biodiversity-related indicators, such as 15.C (Mean area that is protected in terrestrial sites important to biodiversity) and 15.E (Red List Index of species survival). India exhibits comparative weaknesses in these two indicators, and as it transitions to the next stage, it is likely to face greater challenges in these areas (see the red ▽ in Fig. 8c). For China, as shown in red ▽ in Fig.8e, China would face challenges in realizing indicators 13.B ($CO_2$ emissions from fossil fuel combustion and cement production), 14.F (Fish caught by trawling or dredging), and 17.A (Corporate Tax Haven), in which the country has already shown comparative weakness. Attention should also be given to indicators regarding overnutrition (2.B, 2.F), in which China has relative strength but may lose it in the future (as forewarned by Fig. 6e). A recent study provided evidence that the prevalence of obesity of adults in China is more than doubled between 2004 (3.1%) and 2018 (8.1%)[54]. It is notable that indicators related to overnutrition (2.B, 2.F) are some of the main 'orphaned' indicators in the SDG framework, not only for high-income countries, but also for low-income ones[55]. For the USA (Fig. 8g, h), beyond the indicators in Area III − such as overnutrition (2.B, 2.F) and waste and emissions (12.A, 12.C, 12.E, 13.B) − focus should also be placed on addressing social and environmental impacts linked to imports (8.D, 12.F, 14.A), inequality (10.A), and Corporate Tax Haven (17.A) in Area IV, despite these indicators having absolute scores above the global average.

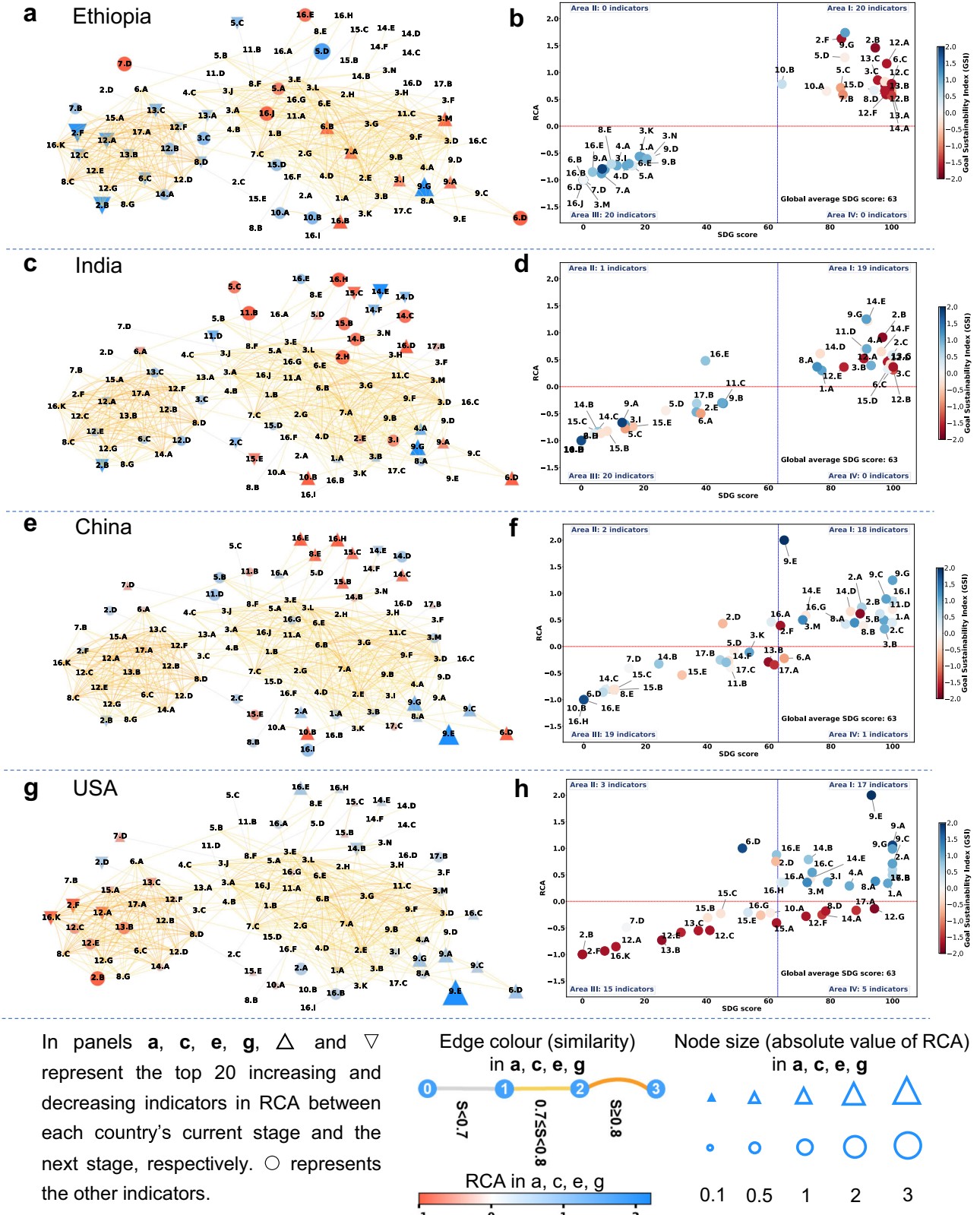

In panels **a**, **c**, **e**, **g**, △ and ▽ represent the top 20 increasing and decreasing indicators in RCA between each country's current stage and the next stage, respectively. ○ represents the other indicators.

Edge colour (similarity) in **a**, **c**, **e**, **g**

Node size (absolute value of RCA) in **a**, **c**, **e**, **g**

RCA in a, c, e, g

These cases illustrate how the structured analysis of nations' sustainable development trajectories can provide insights for shaping development policies. However, it is imperative to view these patterns not as deterministic 'destinies' (or fates) but rather as early warnings that can guide preventative strategies. This recognition underscores that countries, even at similar stages of development, may exhibit distinct approaches to specific SDG indicators, particularly those that are prone to being overlooked (Fig.7).

As we peer past the 2030 mark, it becomes imperative for the United Nations and the global community to undertake a comprehensive review of the SDG indicator framework. Notably, countries with low SDG scores perform 'well' in indicators related to

**Fig. 8 | The SDG space and indicator performance for four representative countries in 2022.** Panels **a**, **b**, Ethiopia (Stage 1). Panels **c**, **d**, India (Stage 2). Panels **e**, **f**, China (Stage 3). Panels **g**, **h**, United States (Stage 4). In panels **a**, **c**, **e**, **g**, blue nodes represent the top 20 SDG indicators by revealed comparative advantage (RCA) values, while red nodes represent the bottom 20. Note: Fig. 8 displays only the top and bottom 20 SDG indicators based on RCA values; the remaining indicators are not highlighted. The size of each node corresponds to the absolute RCA value. Countries are evenly divided into four stages according to their absolute score rankings: Stage 1 (Primary stage), Stage 2 (Low medium stage), Stage 3 (High medium stage), and Stage 4 (Advanced stage). Stage 4 is further divided into two halves to identify the top 20 increasing and decreasing indicators in RCA for the USA (in the first half of Stage 4) between its current stage and the next stage (see Supplementary Table 1). In panels **b**, **d**, **f**, **h**, scatter diagrams are divided into four areas by the lines at RCA = 0 and the global average SDG score of 63 in 2022. Area I: Countries with high RCA and SDG Scores for these indicators have reached a relatively ideal situation. Area II: Countries with high RCA but low SDG scores for these indicators face relative difficulties in improving them, likely due to low global averages for these indicators. Area III: Countries with low RCA and SDG Scores for these indicators show both relative and absolute underperformance, indicating that these indicators should be prioritized for improvement. Area IV: Countries with low RCA but relatively high SDG scores for these indicators demonstrate poor relative performance but strong absolute performance.

overnutrition (2.B, 2.F) and waste and emissions indicators (12.A, 12.D, 13.B, 13.C, 14.D, 14.E). However, they may lose these relative advantages as they progress in their pursuit of greater wealth (i.e., income) and well-being. Moreover, within the same goals, the performance of SDG indicators can vary greatly (Fig. 4). For instance, in Goal 2 (No hunger), the indicators of 2.B (Prevalence of obesity) and 2.F (Human Trophic Level) are quite different from other indicators and are continuously under-specialized. If we just look at the performance of Goals, rather than the specific indicators within them, the different performance of unique indicators can be easily overlooked. All of the above can affect the scientific evaluation of a country's sustainable development state. In addition to 'orphaned indicators', policy attention is suggested to shift to 'underserved populations', such as indigenous people, who constitute 5% of the world's population, but make up 15% of the world's poor[56]. Lastly, the United Nations and the international community are encouraged to identify 'underperforming countries'[57] with less progress in SDGs to improve the whole sustainability level of the world.

To address the above challenges, the United Nations and the international community should continue to support these 'orphaned indicators'. For example, the OECD can aid countries with low SDG scores in promoting renewable energy development through the Clean Energy Finance and Investment Mobilisation (CEFIM) Programme[58]. To improve the biodiversity indicators, the newly launched Global Biodiversity Framework (GBF) Fund[59] can provide special aid for indigenous people and countries with low SDG scores (e.g., the SDG score below 60, see Supplementary Table 1). The European Fund for Sustainable Development Plus (EFSD+) Programme[60] is another good example of financial mechanisms supporting countries in developing their 'orphaned indicators'.

In addition, local investments in tangible economic activities and infrastructure — that is the 'real economy' — that directly contribute to production, employment, and economic growth are essential for promoting sustainable development. These include the allocation of financial resources to industries such as manufacturing, agriculture, services, and infrastructure projects like transportation, renewable energy, and advanced telecommunications. Such local investments are crucial for fostering innovation and creating inclusive economic opportunities. However, the relevant SDG targets and indicators in these areas, particularly those under UN SDGs 8 and 9, are ill-defined. Notably, none of the indicators used to measure these development-related targets adequately address these issues, which represents a significant weakness in the SDG framework. This issue warrants further attention.

Our approach, techniques, and findings require a few clarifications as well as consideration of several limitations. We acknowledge that some of the sustainable development indicators for which certain countries exhibit an RCA (a specialization) might reflect an intentional, premeditated national preference over time, such as the environmental impacts of imports, as part of a larger importation policy. While our data allow us to identify which advantages — or, as they are also referred to in this study (and in Economic Complexity) 'specializations' — are likely to diminish as a country develops, they do not enable us to

differentiate between intentional (i.e., preferred, premeditated) and unintentional specializations. Indeed, 'unintentional specializations' may arise due to a country's geography or geology, such as being landlocked or possessing mineral resource reserves[61]. Additionally, other detrimental factors, such as colonial legacy, may predetermine a country's development trajectory[61].

To clarify, the use of RCA to discuss the performance of countries with low SDG scores on certain SDG indicators is done within the context of the Product Space approach. However, it does not imply that these countries have intentionally pursued this RCA, nor does it suggest that the historical reasons and circumstances underpinning it are inherently 'advantageous'. Using the terms 'advantage' or 'specialization' outside the context of the Product Space approach – when referring to a country meeting certain SDG targets – may be misleading, especially when these 'advantages' may reflect low-income status rather than deliberate policies or capabilities. For example, several indicators, such as lower environmental footprints and reduced health risks like obesity, may result from limited resources. At the same time, this does not inherently render them disadvantages — a low ecological footprint can be considered a positive attribute, regardless of its underlying causes. This invites further discussion and underscores the need for future research.

Moreover, our intention is not to suggest that countries with low SDG scores should maintain their current status to preserve these 'advantages' (RCAs). Instead, the aim of this study is to use the RCA framework to identify national sustainable development trajectories as well as areas where targeted interventions are needed to enhance development outcomes.

In the same vein, the degree of agency we can attribute to national governments in such circumstances (e.g., 'being landlocked', 'being resource-endowed', and 'being post-colonial') when developing a relative specialization remains a matter of debate. Addressing the distinction between intentional and unintentional specializations, as well as the role of national governments' agency, requires further study and may involve subjective judgments. While this study opens avenues for such discussions, the interpretation of RCAs as normative preferences lies outside its scope.

On a related note, it should be emphasized that local variations at subnational levels and units of analysis (regional, provincial, urban, peri-urban, rural) in socio-economic conditions, culture, governance, policy environments, and institutional capacities are important factors that may not be fully captured by our aggregated data and global trends. We acknowledge that even within the same country, SDG performance and strategies may differ between regions and between urban and rural areas[62]. Due to data limitations, this study focuses on global and national sustainable development through 96 SDG indicators. Future research could extend this analysis to regional, state, and city levels.

Lastly, due to the difference in political and economic conditions[63,64], disparities in data availability and quality across countries may result in biased representations of SDG performance, limiting the generalizability and comparability of our findings. With growing calls to enhance the SDG database[65,66], future studies should

critically and comparatively examine SDG indicators and the pathways toward their realization.

## Methods

To reveal SDG development patterns, we provide a lens by constructing the relatedness networks of SDGs, and by employing the measures and techniques of revealed comparative advantage (RCA), goal (country) sustainability index (GSI or CSI), and the SDG space. The relatedness networks of SDGs, derived from data collected between 2000 and 2022, measures the overall affinity between a specific SDG indicator and a country to uncover the SDG development patterns, such as how a country gains or loses a comparative advantage in an SDG indicator during its evolution.

### Revealed comparative advantage (RCA)

We use the revealed comparative advantage (RCA) in SDGs to determine a country's specializations, evidenced by scores of 17 Sustainable Development Goals and their related 96 indicators. The RCA is based on the Ricardian comparative advantage concept, which commonly refers to the Balassa index[33].

The RCA of country $c$ in SDG indicator $i$ is defined as Eq. (1):

$$RCA_{c,i} = RCA'_{c,i} - 1 = \frac{Score(c,i)/\sum_i Score(c,i)}{\sum_c Score(c,i)/\sum_{c,i} Score(c,i)} - 1 \quad (1)$$

where $Score(c,i)$ is the score on SDG indicator $i$ that country $c$ has, $\sum_i Score(c,i)$ is country $c$'s overall SDG indicator score, $\sum_c Score(c,i)$ is the total score on SDG indicator $i$ across all countries, and $\sum_{c,i} Score(c,i)$ is the total score across all countries and indicators.

$RCA'_{c,i}$, which is the main part of RCA, is the ratio of two shares. The numerator is the share of a country's score on a given indicator in its overall score across the 96 indicators, and the denominator is the share of the world score on the same indicator in total scores across all countries and indicators.

If country $c$ has a higher share of SDG indicator $i$ in terms of SDG scores than the world average, then $RCA_{c,i} > 0$, and that country is considered to have specialization in SDG indicator $i$. For example, suppose the share of China's SDG score on indicator 2.G (Prevalence of undernourishment) in its overall score across the 96 indicators is higher than the world average share of 2.G. In that case China is regarded as having relative advantage in 2.G.

### The country sustainability index (CSI) and goal sustainability index (GSI)

Inspired by the economic complexity index (ECI) and product complexity index (PCI) set out by Hidalgo and Hausmann[23], we propose a method, which is equivalent to a spectral clustering algorithm, to calculate the country sustainability index (CSI) and goal sustainability index (GSI) based on countries' RCA in SDG indicators. The method for calculating the CSI − which is related to matrix factorization and dimensionality reduction technique − provides a powerful way to summarise the SDG performance of countries. In general, the CSI values show a positive correlation with the SDG scores of countries (Supplementary Fig. 8).

The algorithm is equivalent to finding the eigenvalues of a matrix and also related to a spectral clustering algorithm[21,22] dividing SDG indicators (countries) into two groups, SDG indicators with higher and lower GSI values (countries with higher and lower CSI values). As a result of a reflections algorithm, countries with higher (or lower) CSI values are more likely to have comparative advantage in SDG indicators with higher (or lower) GSI values. Since CSI values are positively correlated with countries' SDG scores, we can infer which set of goals countries with higher (or lower) SDG scores have a comparative advantage.

The CSI (GSI), similar to the ECI (PCI), is defined through an iterative, self-referential method of reflections algorithm. After setting initial values for the CSI and GSI (e.g., the number of indicators a country specializes in, and the number of countries that specialize in each indicator), we calculate a country's CSI by taking the average GSI values of the SDG indicators in which the country has a relative advantage. In the same way, we calculate an SDG indicator' GSI by the average CSI values of countries with a relative advantage. And then, we recursively use the obtained GSI and CSI values to correct each other until the values are stable. The GSI and CSI can help gain significant insights into SDG development patterns.

### The SDG space and country space

We construct the SDG space from the SDG proximity matrix obtained from countries' RCA in SDG indicators, where nodes represent the SDG indicators and edges represent the similarity between them. The proximity between SDG indicators in the nation-indicator bipartite network is defined by the conditional probability that two indicators are co-specialized within a nation. Similarly, we construct the country space from the country proximity matrix obtained from countries' RCA in SDG indicators, where nodes represent countries and edges represent the similarity between them.

To identify the structure of the SDG space, we use a path-length cost-function method named the kamada_kawai layout algorithm[67] to expose two main clusters. Subsequently, we define the membership of each SDG indicator in the two central communities in the network using the Girvan−Newman method[68]. The Girvan−Newman algorithm identifies communities by progressively eliminating graph edges. At each stage, it traditionally eliminates the 'most useful' edge with the highest betweenness centrality. As the graph fragments, the tightly-knit community structure is revealed. The networks connecting SDG indicators tend to have a dumbbell structure.

The proximity between SDG indicator $i$ and $j$ is defined as the product of the pairwise conditional probabilities of a country having an advantage ($RCA_{c,i} > 0$) on one indicator given an advantage in the other ($RCA_{c,j} > 0$), as shown in Eq. (2):

$$\emptyset_{i,j} = P\left(RCA_{c,i} > 0, |RCA_{c,j} > 0\right) * P\left(RCA_{c,j} > 0, |RCA_{c,i} > 0\right) \quad (2)$$

$\emptyset$ is a 96 × 96 matrix that captures the proximity between pairs of SDG indicators, which is equal to the cosine similarity ($S_{i,j}$). The cosine similarity ($S_{i,j}$) can be calculated by Eqs. (3) and (4):

$$S_{i,j} = \frac{\sum_c RCA''_{c,i} * RCA''_{c,j}}{\sqrt{\sum_c RCA''^2_{c,i}} \sqrt{\sum_c RCA''^2_{c,j}}} \quad (3)$$

where

$$RCA''_{c,i} = \begin{cases} 1, \text{ if } RCA_{c,i} > 0 \\ 0, \text{ if } RCA_{c,i} \leq 0 \end{cases} \quad (4)$$

$RCA_{c,i} > 0$ means country $c$ has a comparative advantage in the indicator $i$, while $RCA_{c,i} \leq 0$ means country $c$ doesn't have a comparative advantage in the indicator $i$.

The resulting similarity $S_{i,j}$ ranges from 0 to 1, where 1 means the probability that each country simultaneously has comparative advantage over the two indicators is 100%, and 0 means when a country has a comparative advantage in one indicator, it always has a relative disadvantage in the other. The in-between values indicate the synergy degree of the two indicators. We call the similarity $S_{i,j}$ as the synergy similarity. By contrast, the trade-off similarity is defined as $T.S._{i,j} = 1 - S_{i,j}$, which also ranges from 0 to 1, but the meanings are opposite to $S_{i,j}$, where 1 means when a country has a comparative advantage in one indicator, it always has a relative disadvantage in the

other. The in-between values indicate the trade-off degree of the two indicators.

To provide a visualization that includes all 96 SDG indicators, we reached all nodes by calculating the maximum spanning tree, which consists of the 95 edges that maximize the added proximity of the tree. To show the strong edges, we overlay all edges with a proximity greater than a certain threshold of 0.7. We perform a sensitivity analysis of the threshold settings (Supplementary Fig. 9) and find that the determination of the proximity threshold does not affect the SDG space's structure. We only consider the proximity threshold to be larger than 0.5, which indicates the probability of two indicators having similarity is greater than 50%.

## The relatedness networks of SDGs

Using the RCA, CSI, GSI, and SDG space, we recognize the relatedness networks of SDGs, which illustrate the affinity between a specific SDG indicator and a country. On a global level, by displaying the GSI values of SDG indicators in the SDG space, we identify which groups of countries have revealed comparative advantage in which groups of SDG indicators since the CSI values are related to the GSI values. At the national scale, the SDG space with each country's RCA in various SDG indicators shows each country's SDG performance. By analyzing the structures of the relatedness networks of SDGs and the patterns of how countries evolve in SDGs, we provide maps for the world and each country to know where they are and where they will go in SDG development.

## The moving-window approach

We examine the impact of countries' rising SDG ranks on the RCA values of SDG indicators and use a moving-window method[18,35] to smooth out the fluctuation and highlight the RCA trends. In all, we sort 166 countries along the SDG rank gradient from the lowest to the highest SDG rank. We set the moving-window size at 50, resulting in a total of 117 windows or country groups (that is, the 1st country group where countries rank 1–50, the 2nd country group where countries rank 2–51, …, the 117th country group where countries rank 117–166). To test the sensitivity of moving-window size on results, we use different sizes (from 20 to 70) to compare the trends and the turning points (i.e., intersections). Similar development paths are found under all moving-window sizes (Supplementary Fig. 7). The smaller the size is, the more volatile the RCA lines are and the flatter the X shape is. We set a middle size of 50 to prevent the X-shaped trend from being too volatile or flat.

## Data

We collected the data of SDG scores from the Sustainable Development Report (SDR) 2023[6] via the following website[34]. Although SDG Index rankings and scores from one edition cannot be compared with the results from previous editions, the latest report provides time series for the SDG Index, calculated retroactively using the same indicators and methods. The scores on individual SDG goals and indicators indicate percentages of optimal performance. The difference between any score and the maximum value of 100 is therefore the distance in percentage points that a country must overcome to reach optimum SDG performance. This ensures that all indicators are comparable. To ensure data continuity and quality, we extracted data for 96 indicators from 98 indicators for 166 countries from 2000 to 2022, while 2 indicators which cannot cover most of countries are excluded. Although the SDG agenda begins in 2015, similar to the SDRs[6,69], the starting year for our analysis is 2000, in order to provide a more comprehensive review of the historical data. The selected SDG indicator list is shown in Tables 1 and 2.

The SDR includes data from both official and unofficial sources. Most of the data come from international organizations which have extensive and rigorous data validation processes. The other data sources (around a third) come from household surveys, civil society organizations and networks, and peer-reviewed journals. The details of the data sources can be found in the Sustainable Development Report[6]. In addition, the related SDG Index methodology and datasets have undergone multiple peer reviews and have been used to substantiate previous notable studies in this field[6,69–71]. Naturally, the dataset has its advantages and disadvantages. The advantages include its broad coverage and consistency over time, which allow for meaningful comparisons across countries and time periods. The disadvantages include a lack of granularity needed to capture local (e.g., urban level) variations and specific economic activities (e.g., industries) within countries. Additionally, some indicators are not fully applicable to all countries (e.g., 14.C. Mean area that is protected in marine sites important to biodiversity).

## Robustness check

To ensure the robustness of our findings, we evaluated the results from the following perspectives: (1) whether a consistent SDG space structure can be reproduced using data from different years; (2) whether the SDG space structure remains similar (i.e., stable) after excluding overlapping indicators; (3) whether the structure is consistent across different data sources; (4) whether the bipolar world results hold under varying similarity thresholds; and (5) whether consistent evolution patterns of SDG indicators emerge with different moving window sizes.

Supplementary Fig. 3 demonstrates that a similar SDG space structure is observed across the years of 2000, 2015, and 2022. This consistency indicates that our finding of a bipolar world within the SDG space is robust and not influenced by the choice of years.

Notably, some SDG indicators may overlap, such as 2.B (Prevalence of obesity) and 2.F (Human Trophic Level) in this study. To ensure the robustness of our results, we removed the overlapping indicators, retaining only one unique indicator, and re-examined the patterns (Supplementary Fig. 10). The findings remained robust, confirming the consistency of our analysis.

To validate the robustness of our results using different data sources, we collected additional data from the SDG database of United Nations Statistics Division[72]. For consistency in comparison, we included only the data on targets that matched the 96 indicators used in this study (see Tables 1 and 2 for the matching process). Data source: https://unstats.un.org/sdgs/dataportal/database. As shown in Fig. 1 (results of SDR 2023 database) and Supplementary Fig. 11 (results of UNSD SDG database), the SDG spaces exhibit a similar structure.

In this study, we generated the SDG spaces by setting the similarity threshold between SDG indicators at 0.7. We tested the robustness of our results by varying the similarity values. As seen in Supplementary Fig. 9, the indicator clusters remain consistent, supporting the SDG space structure.

We also examined whether varying the moving-window sizes would yield similar evolution patterns of SDG indicators. Supplementary Fig. 7 shows that the evolution patterns of SDG indicators and the ranking of countries' SDG scores in 2022 remain consistent across moving-window sizes of 20, 30, 40, 50, 60, and 70.

Overall, our findings demonstrate robustness across various tests and conditions.

## Data availability

Source data are provided with this paper. The source data are deposited in the Zenodo database under accession code [https://zenodo.org/records/14238743][73]. The SDG spaces of all 166 countries are provided on the website: http://www.spacelab.team/#/SDGSpace. The original data of the SDG scores from 2000 to 2022 in the Sustainable Development Report (SDR) 2023[6] can be accessed by the following website: https://dashboards.sdgindex.org. Source data are provided with this paper.

## Code availability
The code for the main figure of the article can be found at the Zenodo database under accession code [https://zenodo.org/records/14238743][73]. For the detailed calculation explanation (technically) of country sustainability index (CSI) and goal sustainability index (GSI), similar to economic complexity index (ECI) and product complexity index (PCI), please refer to the OEC website [https://oec.world/en/resources/methods#eci-technically], or contact the corresponding authors.

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

## Acknowledgements

We gratefully acknowledge the financial support of National Natural Science Foundation of China (No. 52070034 and 52470207 received by H.W., and 71961147003 received by W.Q.C.). Fengmei Ma and Heming Wang acknowledge the helpful discussion with Peng Wang and Xinyi Geng at Institute of Urban Environment, Chinese Academy of Sciences. C.A.H. acknowledges the support of European Union LearnData, GA no. 101086712 a.k.a. 101086712-LearnDataHORIZON-WIDERA–2022-TALENTS–01 (https://cordis.europa.eu/project/id/101086712), IAST funding from the French National Research Agency (ANR) under grant ANR–17-EURE-0010 (Investissements d'Avenir program), and European Lighthouse of AI for Sustainability ELIAS [101120237-HOR-IZON-CL4-2022-HUMAN-02]. Y.Z.Z. acknowledges the financial support of National Social Science Fund of China (No. 23ATJ006).

## Author contributions

H.W., A.T., C.A.H., W.Q.C., and H.S. conceived the study. F.M., H.W., A.T., and C.A.H. led the writing with input from W.Q.C., J.Z., Y.Z., Y.Z.Z., Y.G.Z, and F.M. performed the analyses, with support from C.A.H. on methodology, from H.W. on datasets, and from Y.Z. and J.Z. on figures and website design. Y.G.Z., B.F., Y.Z.Z., and W.Q.C. advised the policy implications. F.M., H.W., A.T., C.A.H., H.S, Y.Z., J.Z., W.Q.C., Y.Z.Z., Y.G.Z, and B.F. reviewed and commented on the manuscript.

## Funding

## Competing interests

The authors declare no competing interests.
