## [Peer Review File · Nature Communications]

REVIEWER COMMENTS

Reviewer #1 (Remarks to the Author):

This paper applies the “product space” technique developed by Hidalgo et al. (2007) to construct an SDG space for countries. Analyzing data for 166 countries between 2000 and 2022, the results suggest that there exists a bipolar world, with distinct groups of countries specializing in specific targets. The results also point to under-developed SDG targets and three country development trajectories: i) under-specialization in targets relating to environmental quality and overnutrition; ii) specialization in targets relating to poverty eradication and education; and iii) a follow-up emphasis on targets pertaining to labor rights, and research and development.

While the research questions are relevant and the results appear reasonable (but note some of them are not exactly new findings), the analysis leaves room for improvement. I have some comments below that are in no particular order that may help improve the paper.

Major comments

1. The paper can make new, important contributions on two fronts: cross-country analysis and country-specific analysis. While the paper demonstrates to some degree that it could analyze cross-country trends over time, it generally falls short of offering the level of detailed country-specific analysis that can provide country-specific relevant advice for actionable policies. So both types of analysis will need more improvement. I will go into more detailed feedback on each below.

2. For cross-country analysis, the main results in the paper are that developed countries have a relative advantage related to zero hunger, living security, economic development, labor rights, national statistical system performance, and protection of life on land. In contrast, developing countries have a relative advantage in SDG targets related to overnutrition, social and environmental impacts of imports, resource use and carbon emissions, and corporate tax havens. I am not sure I am convinced by this somewhat simple dichotomy of findings. Two populous developing countries, China and India, account for roughly more than 40% of the population in the developing world. These two countries have long been known to struggle with environmental degradation issues, including air and water pollution. Some emerging countries with a large population such as Bangladesh and Vietnam also face with similar environmental challenges recently, both of these countries frequently top the list of the globally most polluting cities. In fact, on page 17, the authors acknowledge themselves that China has a problem with CO₂ emission from fossil fuel and cement production. These contradictory results need careful explanation.

3. Regarding country-specific analysis, I similarly find the brief discussion on China (page 17) inadequate. The paper would add practically useful advice for countries if it can zoom in on a

couple of countries as case studies (perhaps with different development trajectories), where it applies the proposed framework of analysis and subsequently clearly identifies the weaknesses and strengths for each country. For now, the current analysis appears to be too vague and high-level to be useful for any particular country.

4. I would also like to see more engagement and comparison of the paper findings with those in the existing literature. For example, (despite the shortcomings discussed in point 2 above), how does the dichotomy of findings for developed and developing countries discussed above relate to previous studies? Are they completely new? Are they backed up by at least some previous qualitative analysis? I have similar questions with the country development trajectories discussed in Figure 3 and Figure 6.

5. The paper seems to take for granted that the SDG targets (and indicators) are non-overlapping. This is not true, as discussed in many previous studies. In fact, a useful robustness check is to better understand whether these non-overlapping (similar) targets would end up having similar score/ color/ node size in the map?

6. Certain SDG goals/ targets may involve tradeoffs for countries, particularly poorer countries. For example, faster poverty reduction may often come with faster environmental degradation. The paper does not discuss this issue at all. One implication is that should these tradeoff targets end up, by definition, in the red and blue clusters of the proposed framework?

7. In general, are there any test statistics (or math quantities) to show that the bipolar patterns in the figures are not artificially caused by certain parameters? In other words, could the authors show different robustness checks to confirm these bipolar patterns? In addition, besides a bipolar pattern, could there be instead a spectrum/ range of values where countries may form several smaller clusters on this range instead of the bipolar pattern?

8. To better anchor the paper findings with the current literature, it would be useful to assess how the country score rankings using the proposed framework compares with other overall SDG indexes such as that employed by the Sustainable Development Reports. In particular, it would be useful to pick out some countries that change ranks using different methods for more discussion.

Other minor comments

9. Not all readers are familiar with the SDGs, so it could be useful to add a table listing each goal and target to the end of the Method section for easy reference.

10. Since the authors obtain their data from the Sustainable Development Report 2023, how do their findings compare with those in this report? To what extent these data affect the paper findings (compared with other data sources, such as from or the UN or the World Bank)?

Reviewer #2 (Remarks to the Author):

The manuscript provides a valuable contribution to the analysis of countries' pathways towards greater sustainability as envisioned in the SDG framework through the lens of development economics. The study compares the trajectories of 166 countries (from 2000-22), and explores whether countries' performance follows distinct patterns by considering 96 global indicators from the SDR. The approach chosen is systematic and sophisticated, employing methodologies innovatively in the context of sustainable development. Overall, the manuscript contributes to advancing knowledge in the field of sustainable development, and can inform more effective strategies for achieving the SDGs across different countries. Notably, I sincerely appreciate that the authors want to make their results on SDG spaces publicly available and easily accessible through the dedicated web-interface.

However, there are some points I would like to raise which I hope can support the authors in improving the manuscript:

1. The authors' choice of referring to the assessed indicators with capital letters (such as 1A, 1B...) can be misleading, as some of the official SDG targets referring to the means of implementation under the respective goals are named alike, although with lower case letters (1a, 1b, ...). If the authors want to stick to this format, they should be explicit about the distinction.

2. Relatedly, the authors could mention earlier in the manuscript that the study uses (proxy) indicators for assessing performance on SDG targets from the SDR (not SDG targets), and that these not always correspond to the official SDG indicators. In this respect, they could further explain why they decided to do so (I suppose due to issues of data availability in the official data set). In Supplementary table 1, consider adding a column indicating whether the indicator is an official SDG indicator ("UNSD Match" in column S, tab "Codebook" in SDR2023_data excel file).

3. While being aware that the terms "developed" vs. "developing" countries is commonly used, I would encourage the authors to refrain from this terminology, particularly since they do not specify where this categorization stems from and is based on. In my opinion, it would be better to distinguish between countries by income levels (from low- to high-income countries) and rename the groups accordingly.

4. In this respect, I strongly encourage the authors to revise the regional groupings in Fig. 2. If the authors want to display regional differences, they should stick to official regional groupings (e.g., (aggregates of) those indicated in SDR, or the M49 standard). Distinguishing between Africa/East & South Asia/Latin America and the Caribbean vs. Industrialized Countries is an unfortunate choice.
5. The findings and insights derived from the method may not be universally applicable to all countries or regions, as they are based on aggregated data and global trends. The authors might want to acknowledge more explicitly that local variations in socio-economic conditions, culture, governance, policy environments, and institutional capacities may not be adequately captured, but might be necessary to consider for providing a more comprehensive understanding of SDG development patterns.
6. The authors also might want to acknowledge that some indicators are not applicable to all countries in their entirety (e.g., indicator 14.C on protected areas in marine sites not relevant for landlocked countries).
7. L129-31: I think this statement should be rephrased and needs further (potential) explanation. First, some of the indicators for which “developing” countries have an RCA could indeed be a preference (e.g. environmental impacts of imports). Second, your data allows you for stating which advantages/specializations are likely to be decreasing when a country “develops”. But does this qualify to speak about (un)intentional specialization? Maybe it would be useful to further reflect on the RCA construction and its limitations/implications in this respect as well.
8. The meaning and interpretation of the network links (rather called edges, and I would recommend to use this term consistently throughout the manuscript) between SDGs and the degree values are barely being addressed in the text.
9. L286-90: I would appreciate if the authors could elaborate on how they interpret these findings.
10. To better highlight and acknowledge the differences shown in Fig.6, the discussion section would benefit from an additional example besides China.
11. The interpretation of proximity values as indicators of synergy and trade-off should be cautiously considered, recognizing the complexity of sustainable development dynamics.
12. Finally but importantly, some of the research aims set out in the beginning in the manuscript (whether countries taking different paths have achieved different degrees of success / whether and under which circumstances countries make progress towards their SDG commitment) have not been sufficiently answered by the authors. While I highly appreciate the extensive data and visualization presented (also in the supplementary material), the authors could have used more space for interpreting and reflecting on their results (sometimes less can be more).

Minor comments:

13. L52: Please correct “constitute” to “constitutes”

14. L55: Are we not already beyond the halfway point to meet the SDGs?
15. L62: Please be consistent in the use of italics for goal descriptions
16. L65: Key development dimensions such as?
17. L68: Variables or entities such as?
18. L94-96: Briefly describe here what “specialization” in the RCA means in your study
19. L98: Explain here briefly what the CSI and GSI cover
20. L102: I would change “correlation” to “relationship to” to avoid misunderstandings (or correspondence, as used in Supplementary Table 1)
21. L184 vs L194: Parenthesis “(Fig 3a-c)” inconsistent
22. L210-11: Please elaborate on what you mean with “moved ahead” in a few words
23. L241-42: Please add here that countries were sorted from lowest to highest SDG rank
24. Fig 5: Add headings (Stage 1-4) similar to other figures in manuscript
25. L246: “The conditions of 17 respective goals” could be reformulated to be more precise
26. L322: When speaking of SDGs related to “economic development and basic livelihood security (e.g., SDGs 1,2,3,4,7)”, you should clearly include SDG 8 and might want to consider also SDG 11
27. L324-27: You could also refer here more strongly to SDG-related research, indicating a general trend in prioritization of social and economic SDGs over environmental ones (e.g. Chapter 6: “Planetary integrity” (Kotzé et al., 2022) in “The Political Impact of the Sustainable Development Goals Transforming Governance Through Global Goals?”, pp. 140 – 171; DOI: <https://doi.org/10.1017/9781009082945.007>)
28. L369: “and the other targets are not shown” – consider rephrasing this
29. L370: “countries” needs to be written with a capital letter
30. L377-79: Consider is you can justify the sentence “these are not their active choices” based on your analysis (see my comment above)
31. L398-402: These limitations should be addressed more explicitly earlier in the manuscript (see my comment above)
32. L434-455: If I understand correctly, GSI (similar to PCI) aims at reflecting the complexity in achieving more sustainable outcomes. I would suggest including this explanation in this subsection, and maybe also already earlier in the manuscript
33. L438: Please correct method “is” instead of “are”
34. L441: You refer to SDG score in the text, but to SDG index value in Supplementary Fig. 8 – please check your figures, text and tables for consistency
35. L450: Further details on the criteria for setting initial values would be helpful

36. L463: Can you provide any reference to the kamada_kawai layout algorithm

37. L463: Consider rephrasing “And then” to “Subsequently”

38. L494: Please correct space with small “s”

39. L524-5: Please indicate that you refer to 98 “global” indicators (not targets) available, since the dataset contains additional (e.g. regional) data as well. Furthermore, you could reflect a bit more on the quality and potential limitation of the SDR data. Despite efforts to include data from a range of sources, there may still be gaps in coverage, particularly for certain regions or SDG targets. Uneven availability of data across countries and sectors could result in skewed representations of SDG performance, limiting the generalizability of the findings.

40. Supplementary material, S87, Fig 9: “c” for Figure 9c. should be written in bold

Reviewer #3 (Remarks to the Author):

The paper is well written and the authors invested a lot of efforts in explaining the methodology and the potential value of its output by providing valuable orientation for countries to find out to what extent they are on track in regard to the UN SDGs to be achieved by 2030.

It is argued that so far country trajectories in the pursuit of the UN Sustainable Development Goals have been poorly explored because the methodology to do so was largely missing. The authors propose to change this by making use of the methodology developed to capture a country's degree economic complexity(EC). Its product space method employs network analysis to reveal the relatedness, or affinity, between economies and applies it to cluster the measured targets of the different UN SDGs highlighting also trajectories of specialization patterns of regions and nations. The resulting ‘SDG space’ reveals the sustainable development trajectories of 166 nations for the period 2000 to 2022. The SDG space then allows to identify ‘orphaned’ areas with insufficient progress while highlighting future challenges in realizing SDGs.

In their mapping and analysis of country-based UN SDG performance and evolution, the authors largely rely on data collected by numerous international institutions to assess if countries are on track in regard to the numerous targets for each of the 17 UN SDGs. In this context, they heavily rely on the 2023 Sustainable Development Report that indexed and evaluated the evolving country performance in regard to the UN SDGs and its targets. Based on secondary macro data analysis, they have identified and visualized 3,818 SDG spaces for 166 nations spanning the years 2000 to 2022 (<http://www.spacelab.team/#/SDGSpace>) and found underlying structures that largely mirror the fact the low income countries and high income countries face different challenges in meeting the different UN SDGs.

I wonder however to what extent it makes sense to talk of 'comparative advantage' of developing countries (based on RCA Assessment) in meeting certain SDG Goals and Targets that largely mirror the fact that they are low income countries with a lower environment footprint and lower health risk exposure related to problems that largely affect affluent countries (e.g. obesity). None of these countries probably have an interest in remaining 'competitive' with regard to these goals and targets because they largely reflect the lack of ability to cover the essential material needs of the people. This is also reflected in the obvious fact that they perform poorly with regard to UN SDG 1,2,4, 8 and 9 that improve with economic development. So talking of a comparative advantage may be misleading. Another weakness is the relatively uncritical embrace of the secondary data gathered on the macro-level by international institutions. They tend to start from the misleading baseline assumption (reflected in many UN Reports on the SDGs) that state bureaucracies are in a position to enable a country to meet the UN SDGs and its targets. The state may provide well-designed policies to incentivize innovation and entrepreneurship moving into a publicly desirable direction; but, ultimately, inclusive and sustainable change is not enabled by regulation and protest but investments in the real economy. UN SDG 8 and 9 emphasize this point, but none of the indicators used to measure its development-related targets actually care about entrepreneurship on the ground. This is a major weakness of gathering data only on the macro-level because they do not allow for any differentiation within a country (emergence of vibrant and innovative economic ecosystems in unexpected places). There is a lack of references on these controversial aspects behind the official UN SDGs rhetoric.

Nevertheless, applying the EC methodology to do a mapping that shows the trajectories of countries in their pursuit of the UN SDGs and its targets may help provide valuable orientation for policy makers by making it transparent that the main enemy of sustainability in high income countries is affluence (leading to overuse of natural resources, large greenhouse gas emissions and material waste) whereas it is poverty in low income countries (lack of ability to provide the right to food, health, education, employment, etc as well as depletion of natural resources and political instability due to population growth and lack of economic opportunities). This should also be made more explicit in the paper. The methodology is surely sound but the authors need to drop the 'comparative advantage' language and they will have to highlight the limits of the approach in view of the exclusive reliance on macro-level data. Finally, the paper does not contain any concluding remarks that describe major findings as well as weaknesses that may have to be addressed in future research.

Response to Referees for Nature Communications manuscript NCOMMS-24-21670, The Sustainable Development Trajectories of Nations

Acknowledgements

We would like to thank the anonymous reviewers, for providing insightful comments and constructive suggestions for improving and balancing the manuscript, adding important explanations, discussions and examples. We have responded to each in the respective sections below.

General remarks

We enthusiastically submit our revised manuscript, in light of reviewers' helpful remarks and overall endorsement (reviewer 1, "...the paper can make new, important contributions... the paper would add practically useful advice for countries..."; reviewer 2, "...the manuscript provides a valuable contribution to the analysis of countries' pathways towards greater sustainability... the approach chosen is systematic and sophisticated, employing methodologies innovatively in the context of sustainable development... the manuscript contributes to advancing knowledge in the field of sustainable development... I sincerely appreciate that the authors want to make their results on SDG spaces publicly available and easily accessible through the dedicated web-interface"; reviewer 3, "...the paper is well written and the authors invested... efforts in explaining the methodology and the potential value of its output by providing valuable orientation for countries to find out to what extent they are on track in regard to the UN SDGs to be achieved by 2030... the resulting 'SDG space' allows to identify 'orphaned' areas with insufficient progress while highlighting future challenges in realizing SDGs...").

We confirm that this manuscript is 5,262 words in length (excluding Abstract, Methods, and References, and Figure captions), slightly above the 5,000 words limit – to allow major revisions, to acknowledge limitations, and to satisfy the review process as a whole.

We note that the majority of reviewers' comments do not relate to issues of article aims and scope, structure, analytical approach, or contribution to knowledge with respect to our original submission. Rather, reviewers' comments mainly focus on requests for additional explanations (e.g., methods; RCA values, CSI and GSI values), discussions and elaborations on findings (e.g., comparisons to the UN dataset, and previous selected studies), and examples to emphasize the value of the Product Space approach and associated techniques (e.g., country-specific analysis and comparisons), along with acknowledging several limitations (e.g., overlap of indicators withing the original SDG framework).

While we were able to address all reviewers' comments to – what we hope is – a satisfactory extent, we note that the word limits of the Article format confine the extent to which we were able to provide further analyses warranted by a number of comments (e.g., we extend our country-specific analysis, but cannot discuss all countries; at the same time, our online webpage allows further, open-access, analyses and comparisons). Nonetheless, the revised manuscript pays attention to all these important matters (e.g., country-specific analysis, elaboration of the

results, discussing the comparative advantage approach).

We hope that our revisions have given suitable recognition of these matters, while keeping the content of our manuscript in line with our aim of providing a single, succinct piece that highlights the potential of the Product Space approach and techniques to reveal useful patterns underpinning the sustainable development trajectories of nations, as well as providing somewhat of a "road map" for future policies and studies in this consequential space.

We wish to thank the reviewers, for the opportunity to elaborate, and precise our arguments, and improve the manuscript as a whole.

Reviewer #1

General Remarks of Reviewer #1

This paper applies the “product space” technique developed by Hidalgo et al. (2007) to construct an SDG space for countries. Analyzing data for 166 countries between 2000 and 2022, the results suggest that there exists a bipolar world, with distinct groups of countries specializing in specific targets. The results also point to under-developed SDG targets and three country development trajectories: i) under-specialization in targets relating to environmental quality and overnutrition; ii) specialization in targets relating to poverty eradication and education; and iii) a follow-up emphasis on targets pertaining to labor rights, and research and development. While the research questions are relevant and the results appear reasonable (but note some of them are not exactly new findings), the analysis leaves room for improvement. I have some comments below that are in no particular order that may help improve the paper.

Response to 1

Thank you for your summary of our manuscript and your overall support regarding the research questions and results, as well as your recommendations for improvement. We greatly appreciate your feedback, and observations concerning the need for further clarification and elaboration on the cross-country and country-specific analyses. These insights enhance the rigor of our study.

In response, we have undertaken substantial revisions to address these comments, ensuring that the manuscript aligns with the journal’s guidelines while also enriching the analysis to better serve the objectives of our research.

Comment 1.1

1. The paper can make new, important contributions on two fronts: cross-country analysis and country-specific analysis. While the paper demonstrates to some degree that it could analyze cross-country trends over time, it generally falls short of offering the level of detailed country-specific analysis that can provide country-specific relevant advice for actionable policies. So both types of analysis will need more improvement. I will go into more detailed feedback on each below.

Response to 1.1

We fully agree with this comment. The original manuscript could have provided a more detailed exploration of country-specific and cross-country developments over time. In this revised version, we have included such analyses for China (see Fig. 8).

To further strengthen the manuscript, we have incorporated additional case studies (or use cases) to better illustrate the advantages of our approach. Specifically, for the country-specific analysis, we have added more case studies and designed new analytical charts that highlight potential future policy directions for the countries studied.

For the cross-country analysis, we have enhanced our approach by categorizing environment-related SDG indicators separately from other SDG indicators, allowing for a more nuanced analysis.

Detailed responses to each of your specific comments, along with the corresponding revisions, are presented in the point-by-point response that follows.

Comment 1.2

2. For cross-country analysis, the main results in the paper are that developed countries have a relative advantage related to zero hunger, living security, economic development, labor rights, national statistical system performance, and protection of life on land. In contrast, developing countries have a relative advantage in SDG targets related to overnutrition, social and environmental impacts of imports, resource use and carbon emissions, and corporate tax havens. I am not sure I am convinced by this somewhat simple dichotomy of findings. Two populous developing countries, China and India, account for roughly more than 40% of the population in the developing world. These two countries have long been known to struggle with environmental degradation issues, including air and water pollution. Some emerging countries with a large population such as Bangladesh and Vietnam also face with similar environmental challenges recently, both of these countries frequently top the list of the globally most polluting cities. In fact, on page 17, the authors acknowledge themselves that China has a problem with CO₂ emission from fossil fuel and cement production. These contradictory results need careful explanation.

Response to 1.2

Thank you for this comment. We agree that the previous manuscript did not sufficiently summarize the findings from the cross-country analysis. To address this, we have revised the section titled "Underlying Structures of the SDG Space of Nations" in the manuscript. Specifically, we have rephrased the description of the SDG space as follows:

Lines 134-142:

Countries at an advanced stage tend to have relative advantages in indicators related to poverty reduction (1.A), hunger reduction (2.E, 2.G), good health and well-being (3.B, 3.D, 3.F, 3.G, 3.I, 3.K, 3.M), wastewater and air pollution treatment (3.H, 6.D, 14.B), access to clean energy and water (6.B, 6.E, 7.A), industry, innovation and infrastructure (Goal 9), as well as government administration (16.B, 16.C, 16.I, 17.B, 17.C) . In contrast, countries at a primary stage show relative advantages in achieving SDG indicators related to overnutrition (2.B, 2.F), embodied social and environmental impacts in international trade (6.C, 8.C, 8.D, 12.D, 12.F,

13.C, 14.A, 15.A, 16.A), resource use, waste and emissions (6.A, 7.B, 12.A, 12.C, 12.E, 12.G, 13.B), and issues related to corporate tax havens (17.A).

Furthermore, in the section "Sustainable Development Paths of Nations," we categorized the environment-related and non-environment-related SDG indicators into six distinct groups using the Goal Sustainability Index (GSI) (please refer to the figure below). Our analysis revealed that for both environment-related and non-environment-related SDG indicators, the RCA values for higher GSI indicators tend to increase as the national SDG index grows. Conversely, the RCA values for lower GSI indicators decrease as the national SDG index grows. This finding provides a solid foundation for summarizing our findings.

We have modified the corresponding text as follows:

Lines 259-275:

To identify more refined trends, we categorize the SDG indicators into six groups based on their GSI values (Fig. 5): three groups of environment-related indicators (Table 1 in Methods) and three groups of non-environment-related indicators (Table 2 in Methods). Our findings reveal that environment-related indicators with high GSI values primarily involve wastewater treatment, air pollution control, and access to clean energy, which tend to be specialized by countries at advanced stages of development. On the other hand, indicators with low GSI values, such as trade-related pollution and emissions, as well as waste and emissions from production and living, are more commonly associated with countries in the early stages of development.

Similarly, non-environment-related indicators with high GSI values predominantly include no poverty, zero hunger, good health and well-being, quality education, industry, innovation and infrastructure, and effective government administration. In contrast, those with low GSI values mainly involve overnutrition and the social impacts embodied in international trade.

For both environment-related and non-environment-related SDG indicators, the RCA values for higher GSI indicators increase with the growth of the national SDG index, while the RCA values for lower GSI indicators tend to decrease as the national SDG index rises.

Fig. 5 | The change in RCA for six groups of SDG indicators across window groups ranked by SDG scores in 2022. Panels **a**, **b**, and **c**, show the condition for three groups of environment-related indicators, categorized by the goal sustainability index (GSI). **a**, Group 1, $GSI > 0.5$; **b**, Group 2, $-1.0 \leq GSI \leq 0.5$; **c**, Group 3, $GSI < -1.0$. Panels **d**, **e**, and **f**, display the conditions for three groups of non-environment-related indicators. **d**, Group 4, $GSI > 0.5$; **e**, Group 5, $-1.0 \leq GSI \leq 0.5$; **f**, Group 6, $GSI < -1.0$. Note: the window size was determined by grouping 50 countries at a time, resulting in 117 window groups (i.e., countries 1-50, 2-51, ..., 117-166).

Finally, we have carefully reviewed and addressed any contradictory results to ensure consistency throughout the manuscript. In particular, we have added more case studies and revised the analysis for China, which is categorized as Stage 3 (High-medium stage), in the

Discussion Section. Please refer to our response to your Comment 1.3 for detailed information on these revisions.

Comment 1.3

3. Regarding country-specific analysis, I similarly find the brief discussion on China (page 17) inadequate. The paper would add practically useful advice for countries if it can zoom in on a couple of countries as case studies (perhaps with different development trajectories), where it applies the proposed framework of analysis and subsequently clearly identifies the weaknesses and strengths for each country. For now, the current analysis appears to be too vague and high-level to be useful for any particular country.

Response to 1.3

Thank you for your valuable and constructive suggestion. In line with your recommendation, we have incorporated additional case studies that focus on country-specific analyses. Furthermore, we have developed scatter diagrams to illustrate potential future policy directions for different countries.

Area I: Countries with high RCA and SDG scores for SDG indicators have reached a relatively ideal situation.

Area II: Countries with high RCA for SDG indicators but low SDG scores find it relatively difficult to improve these indicators, likely because the global average for the indicator is low. This suggests it could be challenging for other countries to enhance these indicators and provide successful examples.

Area III: Countries with low RCA and SDG scores for SDG indicators demonstrate poor relative and absolute performance on these indicators, indicating these indicators should be prioritized for improvement.

Area IV: Countries with low RCA for SDG indicators but relatively high SDG scores show poor relative performance but strong absolute performance. Improvement in these indicators may be more feasible, as the high global average for these indicators suggests that other countries are performing better and could serve as a reference for success.

By categorizing countries' performance on these indicators, we can offer more tailored and precise policy recommendations.

The detailed revisions are as follows:

Lines 418-432:

In terms of policy guidance, the comparative insights and identified trends in sustainable development trajectories can provide targeted recommendations (Fig. 8). From a static perspective, countries' performance on the indicators can be categorized into four areas (Fig. 8b, d, f, h): Area I (high RCA and SDG Scores) represents an ideal situation. Area II (high RCA

but low SDG scores) where indicators are relatively difficult for a specific country to improve, as other countries are also likely to struggle in making progress and providing successful examples. Area III (low RCA and SDG Scores) represents the worst situation, where both relative and absolute performance on these indicators is poor. Area IV (low RCA but relatively high SDG scores) where indicators may be easier to improve, as other countries might be performing better and could offer models for success.

Taking the USA as an example (Fig. 8h), beyond the indicators in Area III — such as overnutrition (2.B, 2.F) and waste and emissions (12.A, 12.C, 12.E, 13.B) — focus should also be placed on addressing social and environmental impacts linked to imports (8.D, 12.F, 14.A), inequality (10.A), and Corporate Tax Haven (17.A) in Area IV, despite these indicators having absolute scores above the global average.

Additionally, in our original manuscript, we categorized countries into four development stages: Stage 1 (Primary stage), Stage 2 (Low medium stage), Stage 3 (High medium stage), and Stage 4 (Advanced stage). From a dynamic viewpoint, the revealed patterns of trajectories can serve as early warnings of potential challenges that countries may face as they progress from a lower to a higher stage of sustainable development.

Lastly, we offered a detailed presentation of SDG spaces for 166 countries from 2000 to 2022 in our Supplementary Materials and on a dedicated website (<http://www.spacelab.team/#/SDGSpace>). This allows further analyses.

Fig. 8 | The SDG space and indicator performance for four representative countries in 2022. Panels a-b, Ethiopia (Stage 1). Panels c-d, India (Stage 2). Panels e-f, China (Stage 3). Panels g-h, United States (Stage 4). In panels a-h, blue nodes represent the top 20 SDG indicators by RCA values, while red nodes represent the bottom 20. The remaining indicators are not highlighted. The size of each node corresponds to the absolute RCA value. Countries are evenly divided into four stages according to their absolute score rankings: Stage 1 (Primary stage), Stage 2 (Low medium stage), Stage 3 (High medium stage), and Stage 4 (Advanced stage). Stage 4 is further divided into two halves to identify the top 20 increasing and decreasing

indicators in RCA for the USA (in the first half of Stage 4) between its current stage and the next stage (see Supplementary Table 1). In panels **b**, **d**, **f**, **h**, scatter diagrams are divided into four areas by the lines at $RCA = 0$ and the global average SDG score of 63. Area I: Countries with high RCA and SDG Scores for these indicators have reached a relatively ideal situation. Area II: Countries with high RCA but low SDG scores for these indicators face relative difficulties in improving them, likely due to low global averages for these indicators. Area III: Countries with low RCA and SDG Scores for these indicators show both relative and absolute underperformance, indicating that these indicators should be prioritized for improvement. Area IV: Countries with low RCA but relatively high SDG scores for these indicators demonstrate poor relative performance but strong absolute performance.

Comment 1.4

4. I would also like to see more engagement and comparison of the paper findings with those in the existing literature. For example, (despite the shortcomings discussed in point 2 above), how does the dichotomy of findings for developed and developing countries discussed above relate to previous studies? Are they completely new? Are they backed up by at least some previous qualitative analysis? I have similar questions with the country development trajectories discussed in Figure 3 and Figure 6.

Response to 1.4

Thank you for your helpful suggestion. Based on this comment, we have restructured the discussion to provide a more thorough comparison between previous relevant studies and our current research.

Regarding your question: “Are they backed up by at least some previous qualitative analysis?”, we have addressed this in the Discussion Section:

Lines 374-382:

This bipolar world revealed by the SDG space is supported by previous studies³⁷⁻³⁹ that have identified global disparities in SDGs, and it can be partly explained by the trade-offs between SDGs^{12,18,40-43}. For instance, the SDGs related to economic development and basic livelihood security (e.g., SDGs 1, 2, 3, 4, 7, 8, 11), are often associated with higher environmental and resource footprints⁴⁴⁻⁴⁷ and increased greenhouse gas emissions⁴⁸, which can hinder progress on SDGs 12 and 13⁴⁹. These trends align with the concept of the environmental Kuznets curve^{50,51}, which suggests that in the early stages of a nation’s economic development, the pursuit of economic growth tends to take precedence over environmental preservation concerns.

Regarding the comparison with previous studies, we have addressed this as follows:

Lines 390-417:

Distinct from earlier notable studies that assessed countries’ SDG performance³⁻⁸, such as those ranking countries and evaluating their progress towards achieving the SDGs⁶ or exploring

development patterns¹²⁻¹⁹, our study introduces several key features that set it apart.

First, we employed a comparative framework to identify each country's relative strengths and weaknesses, as well as to pinpoint 'orphaned' SDG indicators in specific development areas. This comparative analysis, which contextualizes performance by benchmarking different variables or entities against each other, offers a clearer understanding of each country's relative standing. In contrast, absolute scoring (Figs. 3d-f) can obscure these insights by focusing solely on an overarching SDG score without the nuanced understanding that comes from comparison with other entities and SDGs.

Second, we utilized the 'product space' method and the broader economic complexity approach as effective tools for pattern recognition. These methods help identify national sustainable development trends and reveal whether countries specialize in certain goals. For instance, the derived GSI indicator divides SDG indicators into low-GSI indicators that countries tend to overlook and high-GSI indicators that countries tend to specialize in during their development process (Figs. 3, 5, 6). Additionally, we delineated the development trajectories of nations based on the historical performance of all countries, differing from the Sustainable Development Report⁶, which evaluates whether a country is on track based on its individual performance.

Third, we introduced the concept of the 'SDG space' as a high-resolution tool to monitor countries' performance across 96 SDG indicators. This tool visualizes countries' relative performance and assists in guiding the formulation of development strategies (Supplementary Figs. 12-177). Moreover, we developed a dedicated website to present 3,818 SDG spaces for 166 nations spanning the years 2000 to 2022: <http://www.spacelab.team/#/SDGSpace>. Unlike the traditional product space²⁰, which typically ranks countries and guides them towards advanced products (e.g., motor vehicles and electronics), the SDG space is designed to assist countries monitor their progress within the SDG framework, identifying relative strengths and weaknesses, rather than ranking countries.

Comment 1.5

5. The paper seems to take for granted that the SDG targets (and indicators) are non-overlapping. This is not true, as discussed in many previous studies. In fact, a useful robustness check is to better understand whether these non-overlapping (similar) targets would end up having similar score/ color/ node size in the map?

Response to 1.5

Thank you for your comment. The authors do not take for granted that the SDG targets (and indicators) are non-overlapping. To reflect this recognition, a reference source (<https://www.nature.com/articles/s41598-021-01801-6>) was added to the Methods section, acknowledging the potential overlap between several SDG indicators.

Moreover, in response to your comment, we screened six groups of similar indicators: (1.A &

1.B), (2.B & 2.F), (2.E & 2.H), (3.G & 3.L), (10.A & 10.B), and (14.D, 14.E & 14.F). To conduct a robustness check, we retained only the non-overlapping indicators within these groups and redrew the SDG space map (Supplementary Fig. 10). The resulting map exhibits patterns similar to those observed in Fig. 1 of the main text, confirming the consistency of our findings.

The revisions are as follows:

Lines 709-739 in Robustness check of Methods (please also see our response to 1.7):

To ensure the robustness of our findings, we evaluated the results from the following perspectives: (1) whether a consistent SDG space structure can be reproduced using data from different years; (2) whether the SDG space structure remains similar (i.e., stable) after excluding overlapping indicators; (3) whether the structure is consistent across different data sources; (4) whether the bipolar world results hold under varying similarity thresholds; and (5) whether consistent evolution patterns of SDG indicators emerge with different moving window sizes.

... ..

(2) Notably, some SDG indicators may overlap, such as 2.B (*Prevalence of obesity*) and 2.F (*Human Trophic Level*) in this study. To ensure the robustness of our results, we removed the overlapping indicators, retaining only one unique indicator, and re-examined the patterns (Supplementary Fig. 10). The findings remained robust, confirming the consistency of our analysis.

Supplementary Fig. 10 in the Supplementary Materials is as follows:

Supplementary Figure 10 | The SDG space in 2022 with non-overlapping indicators. We screened six groups of similar indicators: (1.A and 1.B), (2.B and 2.F), (2.E and 2.H), (3.G and 3.L), (10.A and 10.B), and (14.D, 14.E, and 14.F). For the robustness check, we retained only

the non-overlapping indicators within these groups. The node colour represents Goal Sustainability Index (GSI), and the node size corresponds to the node degree, indicating the number of edges connected to the node.

Comment 1.6

6. Certain SDG goals/ targets may involve tradeoffs for countries, particularly poorer countries. For example, faster poverty reduction may often come with faster environmental degradation. The paper does not discuss this issue at all. One implication is that should these tradeoff targets end up, by definition, in the red and blue clusters of the proposed framework?

Response to 1.6

Thank you for this comment. Indeed, certain SDG goals and targets may involve tradeoffs for countries. Although critiquing the SDG framework is not the purpose of our study, we agree with your observation. Following your suggestion, we have highlighted this issue in the Discussion Section.

Moreover, we have added several reference sources to support this claim, including:

Fader, M., Cranmer, C., Lawford, R., & Engel-Cox, J. (2018). Toward an understanding of synergies and trade-offs between water, energy, and food SDG targets. Frontiers in Environmental Science, 6, 112.

Singh, G. G., Cisneros-Montemayor, A. M., Swartz, W., Cheung, W., Guy, J. A., Kenny, T. A., ... & Ota, Y. (2018). A rapid assessment of co-benefits and trade-offs among Sustainable Development Goals. Marine Policy, 93, 223-231.

Scherer, L., Behrens, P., De Koning, A., Heijungs, R., Sprecher, B., & Tukker, A. (2018). Trade-offs between social and environmental Sustainable Development Goals. Environmental Science & Policy, 90, 65-72.

Barbier, E. B., & Burgess, J. C. (2019). Sustainable development goal indicators: Analyzing trade-offs and complementarities. World Development, 122, 295-305.

We have modified the text as follows:

Lines 374-382:

This bipolar world revealed by the SDG space is supported by previous studies³⁷⁻³⁹ that have identified global disparities in SDGs, and it can be partly explained by the trade-offs between SDGs^{12,18,40-43}. For instance, the SDGs related to economic development and basic livelihood security (e.g., SDGs 1, 2, 3, 4, 7, 8, 11), are often associated with higher environmental and resource footprints⁴⁴⁻⁴⁷ and increased greenhouse gas emissions⁴⁸, which can hinder progress on SDGs 12 and 13⁴⁹. These trends align with the concept of the environmental Kuznets curve^{50,51}, which suggests that in the early stages of a nation's economic development, the pursuit of economic growth tends to take precedence over environmental preservation concerns.

Comment 1.7

7. In general, are there any test statistics (or math quantities) to show that the bipolar patterns in the figures are not artificially caused by certain parameters? In other words, could the authors show different robustness checks to confirm these bipolar patterns? In addition, besides a bipolar pattern, could there be instead a spectrum/ range of values where countries may form several smaller clusters on this range instead of the bipolar pattern?

Response to 1.7

Thank you for your comment and suggestion.

In the previous manuscript, we conducted statistical tests for the bipolar patterns and performed a sensitivity analysis on the threshold settings (Supplementary Fig. 9). Our analysis revealed that the determination of the proximity threshold does not affect the structure of the SDG Space.

We apologize for not making this clearer in the original text.

In this revised manuscript, we have added the robustness check part in the Methods section. In addition, following your suggestions, we examined the robustness of our results from the following perspectives: (1) whether a consistent SDG space structure can be reproduced with data from different years; (2) whether the SDG space structure remains similar after excluding overlapping indicators; (3) whether the structure is consistent across different data sources; (4) whether the bipolar world results hold under varying similarity thresholds; and (5) whether consistent evolution patterns of SDG indicators emerge with different moving window sizes.

The revisions are as follows.

Lines 709-739, Robustness check Section of the Methods:

To ensure the robustness of our findings, we evaluated the results from the following perspectives: (1) whether a consistent SDG space structure can be reproduced using data from different years; (2) whether the SDG space structure remains similar (i.e., stable) after excluding overlapping indicators; (3) whether the structure is consistent across different data sources; (4) whether the bipolar world results hold under varying similarity thresholds; and (5) whether consistent evolution patterns of SDG indicators emerge with different moving window sizes.

(1) Supplementary Fig. 3 demonstrates that a similar SDG space structure is observed across the years of 2000, 2015, and 2022. This consistency indicates that our finding of a bipolar world within the SDG space is robust and not influenced by the choice of years.

(2) Notably, some SDG indicators may overlap, such as 2.B (*Prevalence of obesity*) and 2.F (*Human Trophic Level*) in this study. To ensure the robustness of our results, we removed the overlapping indicators, retaining only one unique indicator, and re-examined the patterns (Supplementary Fig. 10). The findings remained robust, confirming the consistency of our analysis.

(3) To validate the robustness of our results using different data sources, we collected additional data from the UNSD SDG database. For consistency in comparison, we included only the data on targets that matched the 96 indicators used in this study (see Tables 1 and 2 in Methods for the matching process). Data source: <https://unstats.un.org/sdgs/dataportal/database>. As shown in Fig. 1 (SDR 2023 database) and Supplementary Fig. 11 (UNSD SDG database), the SDG spaces exhibit a similar structure.

(4) In this study, we generated the SDG spaces by setting the similarity threshold between SDG indicators at 0.7. We tested the robustness of our results by varying the similarity values. As seen in Supplementary Fig. 9, the indicator clusters remain consistent, supporting the SDG space structure.

(5) We also examined whether varying the moving-window sizes would yield similar evolution patterns of SDG indicators. Supplementary Fig. 7 shows that the evolution patterns of SDG indicators and the ranking of countries' SDG scores in 2022 remain consistent across moving-window sizes of 20, 30, 40, 50, 60, and 70.

Overall, our findings demonstrate robustness across various tests and conditions.

Supplementary Figs. 3, 7, 9-11 in the Supplementary Materials are as follows:

Supplementary Figure 3 | The evolution of SDG space. Panels a, c, and e show the global SDG space in 2000, 2015, and 2022, respectively. Panels b, d, and f display the similarity of SDG indicators in 2000, 2015, and 2022, respectively.

Supplementary Figure 7 | The evolution of SDG indicators alongside the ranking of countries' SDG scores in 2022 with varying moving-window sizes. Panels a, b, c, d, e, and f display the results using moving-window sizes of 20, 30, 40, 50, 60, and 70, respectively.

Supplementary Figure 9 | Sensitivity analysis of SDG indicators' similarity in 2022. Panel **a**, SDG indicators with similarity ≥ 0.6 . Panel **b**, SDG indicators with similarity ≥ 0.7 . Panel **c**, SDG indicators with similarity ≥ 0.8 .

Supplementary Figure 10 | The SDG space in 2022 with non-overlapping indicators. We screened six groups of similar indicators: (1.A and 1.B), (2.B and 2.F), (2.E and 2.H), (3.G and 3.L), (10.A and 10.B), and (14.D, 14.E, and 14.F). For the robustness check, we retained only the non-overlapping indicators within these groups. The node colour represents Goal Sustainability Index (GSI), and the node size corresponds to the node degree, indicating the number of edges connected to the node.

Supplementary Figure 11 | The SDG space in 2022 using UN SDG data. The node colour represents Goal Sustainability Index (GSI), and the node size indicates the node degree, representing the number of edges connected to the node. This SDG space was produced using data from the UNSD SDG targets framework, which were matched with the 96 SDG indicators (see Tables 1-2 in Methods). Data source: <https://unstats.un.org/sdgs/dataportal/database>.

Comment 1.8

8. To better anchor the paper findings with the current literature, it would be useful to assess how the country score rankings using the proposed framework compares with other overall SDG indexes such as that employed by the Sustainable Development Reports. In particular, it would be useful to pick out some countries that change ranks using different methods for more discussion.

Response to 1.8

Thank you for this comment. In the original manuscript, we compared the Country Sustainability Index (CSI) with the SDG Index scores.

We found that the CSI values are positively correlated with the SDG Index scores of countries (please see Lines 509-510 in Methods and Supplementary Fig. 8 in the Supplementary Materials). Please see this comparison below.

It is important to note that our focus is not on the rankings of countries in SDG performance. Instead, we concentrate on the patterns underlying the sustainable development trajectories of nations across indicators and over time, and whether countries have so-called ‘orphaned’ (i.e., overlooked) SDG indicators in certain development areas.

Therefore, we did not add discussions on countries’ rankings in this revised manuscript, as it is outside the scope and aims of our study.

Lines 590-591:

In general, the CSI values show a positive correlation with the SDG scores of countries (Supplementary Fig. 8).

Supplementary Figure 8 | The relationship between the country sustainability index (CSI) and the SDG index of countries in 2022. Panel **a** shows the CSI value versus SDG index value. Panel **b** indicates the CSI rank versus the SDG index rank.

Comment 1.9

Other minor comments

9. Not all readers are familiar with the SDGs, so it could be useful to add a table listing each goal and target to the end of the Method section for easy reference.

Response to 1.9

Thank you for your suggestion

Based on your recommendation, we have added two tables listing each indicator to the end of the Data Section of Methods.

The detailed revision is as follows (Lines 699-707):

Table 1 | Environment-related SDG indicators ranked by the Goal Sustainability Index (GSI) and their alignment with UN SDG taxonomies. The environment-related indicators are divided into three groups based on their GSI values. Group 1: $GSI > 0.5$. Group 2: $-1.0 \leq GSI \leq 0.5$. Group 3: $GSI < -1.0$.

Group	GSI in 2022	Indicator code in this study	Indicator code in SDR 2023	Indicator definitions	UNSD target	UNSD match
1	1.79	6.D	sdg6_wastewat	Anthropogenic wastewater that receives treatment	6.3.1	Match
	1.08	7.A	sdg7_cleanfuel	Population with access to clean fuels and technology for cooking	7.1.2	Closely aligned
	0.83	3.H	sdg3_pollmort	Age-standardized death rate attributable to household air pollution and ambient air pollution	3.9.1	Match
	0.79	14.B	sdg14_cleanwat	Ocean Health Index: Clean Waters score	14.1.1	Closely aligned
2	0.27	14.C	sdg14_cpma	Mean area that is protected in marine sites important to biodiversity	14.5.1	Closely aligned
	0.27	2.C	sdg2_pestexp	Exports of hazardous pesticides	3.9	Closely aligned
	0.19	15.D	sdg15_forchg	Permanent deforestation	15.2	Closely aligned
	0.05	7.D	sdg7_renewcon	Renewable energy share in total final energy consumption	7.2.1	Match
	-0.03	11.B	sdg11_pm25	Annual mean concentration of particulate matter of less than 2.5 microns in diameter (PM2.5)	11.6.2	Match
	-0.21	14.E	sdg14_fishstocks	Fish caught from overexploited or collapsed stocks	14.4.1	Closely aligned
	-0.27	15.C	sdg15_cpta	Mean area that is protected in terrestrial sites important to biodiversity	15.1.2	Match
	-0.30	15.B	sdg15_cpfa	Mean area that is protected in freshwater sites important to biodiversity	15.1.2	Match
	-0.34	14.D	sdg14_discard	Fish caught that are then discarded	14.4	Closely aligned
	-0.34	14.F	sdg14_trawl	Fish caught by trawling or dredging	14.4	Closely aligned
	-0.57	15.E	sdg15_redlist	Red List Index of species survival	15.5.1	Match
	-0.65	2.D	sdg2_snmi	Sustainable Nitrogen Management Index	2.4	Closely aligned
	-0.72	13.A	sdg13_co2export	CO2 emissions embodied in fossil fuel exports	13.2	Closely aligned
	-0.82	6.A	sdg6_freshwat	Freshwater withdrawal	6.4.2	Match
	-0.95	7.B	sdg7_co2twh	CO2 emissions from fuel combustion per total electricity output	7.2	Closely aligned
	3	-1.26	12.B	sdg12_explastic	Exports of plastic waste	12.4
-1.40		14.A	sdg14_biomar	Marine biodiversity threats embodied in imports	14.4	Closely aligned
-1.54		12.E	sdg12_nprod	Production-based nitrogen emissions	9.4	Closely aligned
-1.59		12.D	sdg12_nimport	Nitrogen emissions embodied in imports	9.4	Closely aligned
-1.59		13.C	sdg13_co2import	CO2 emissions embodied in imports	13.2	Closely aligned
-1.62		12.A	sdg12_ewaste	Electronic waste	12.4.2	Match
-1.64		15.A	sdg15_biofrwter	Terrestrial and freshwater biodiversity threats embodied in imports	15.5	Closely aligned
-1.66		12.F	sdg12_so2import	SO2 emissions embodied in imports	9.4	Closely aligned
-1.67		6.C	sdg6_scarcew	Scarce water consumption embodied in imports	6.4	Closely aligned
-1.67		12.C	sdg12_msw	Municipal solid waste	12.5	Closely aligned
-1.76		13.B	sdg13_co2gcp	CO2 emissions from fossil fuel combustion and cement production	13.2.2	Closely aligned
-1.84	12.G	sdg12_so2prod	Production-based SO2 emissions	9.4	Closely aligned	

Table 2 | Non-environment-related SDG indicators ranked by the Goal Sustainability Index (GSI) and their alignment with UN SDG taxonomies. The non-environment-related indicators are divided into three groups based on their GSI values. Group 4: $GSI > 0.5$. Group 5: $-1.0 \leq GSI \leq 0.5$. Group 6: $GSI < -1.0$.

Group	GSI in 2022	Indicator code in this study	Indicator code in SDR 2023	Indicator definitions	UNSD target	UNSD match	
4	1.98	9.E	sdg9_rdex	Expenditure on research and development	9.5.1	Match	
	1.87	9.A	sdg9_articles	Articles published in academic journals	9.5	Closely aligned	
	1.30	3.G	sdg3_neonat	Neonatal mortality rate	3.2.2	Match	
	1.29	3.M	sdg3_uhc	Universal health coverage (UHC) index of service coverage	3.8.1	Match	
	1.23	8.A	sdg8_accounts	Adults with an account at a bank or other financial institution or with a mobile-money-service provider	8.10.2	Match	
	1.17	11.C	sdg11_slums	Proportion of urban population living in slums	11.1.1	Match	
	1.08	9.G	sdg9_uni	The Times Higher Education Universities Ranking: Average score of top 3 universities	-	Not in UNSTATS	
	1.08	3.I	sdg3_swb	Subjective well-being	3.4	Closely aligned	
	1.08	9.C	sdg9_lpi	Logistics Performance Index: Quality of trade and transport-related infrastructure	9.1	Closely aligned	
	1.07	16.C	sdg16_cpi	Corruption Perceptions Index	16.5.1, 16.5.2	Closely aligned	
	1.06	3.K	sdg3_traffic	Traffic deaths	3.6.1	Match	
	1.05	1.A	sdg1_lmipov	Poverty headcount ratio at \$3.65/day	1.1.1	Match	
	1.04	4.D	sdg4_second	Lower secondary completion rate	4.1.2	Match	
	1.03	3.D	sdg3_lifex	Life expectancy at birth	3.1:3.9	Closely aligned	
	1.02	9.B	sdg9_intuse	Population using the internet	17.8.1	Match	
	1.01	9.F	sdg9_roads	Rural population with access to all-season roads	9.1.1	Match	
	1.01	4.A	sdg4_earlyedu	Participation rate in pre-primary organized learning	4.2.2	Closely aligned	
	0.98	3.B	sdg3_fertility	Adolescent fertility rate	3.7.2	Match	
	0.97	6.B	sdg6_sanita	Population using at least basic sanitation services	6.2.1	Closely aligned	
	0.95	2.E	sdg2_stunting	Prevalence of stunting in children under 5 years of age	2.2.1	Match	
	0.94	9.D	sdg9_mobuse	Mobile broadband subscriptions	9.c.1, 17.6.1	Closely aligned	
	0.91	2.G	sdg2_undernsh	Prevalence of undernourishment	2.1.1	Match	
	0.84	17.C	sdg17_statperf	Statistical Performance Index	17.18.1: 17.19.2	Closely aligned	
	0.81	16.E	sdg16_exprop	Expropriations are lawful and adequately compensated	16.6	Closely aligned	
	0.75	3.F	sdg3_ncds	Age-standardized death rate due to cardiovascular disease, cancer, diabetes, or chronic respiratory disease in adults aged 30–70 years	3.4.1	Match	
	0.66	2.A	sdg2_crlyld	Cereal yield	2.3, 2.4	Closely aligned	
	0.66	17.B	sdg17_govex	Government spending on health and education	1.a.1	Closely aligned	
	0.58	16.B	sdg16_clabor	Children involved in child labor	8.7.1	Closely aligned	
	0.52	8.E	sdg8_rights	Fundamental labor rights are effectively guaranteed	8.8.2	Match	
	0.51	16.I	sdg16_safe	Population who feel safe walking alone at night in the city or area where they live	16.1.4	Match	
	5	0.47	10.B	sdg10_palma	Palma ratio	10.1	Closely aligned
		0.47	16.D	sdg16_detain	Unsentenced detainees	16.3.2	Match

Group	GSI in 2022	Indicator code in this study	Indicator code in SDR 2023	Indicator definitions	UNSD target	UNSD match
	0.42	1.B	sdg1_wpc	Poverty headcount ratio at \$2.15/day	1.1.1	Match
	0.42	5.B	sdg5_familypl	Demand for family planning satisfied by modern methods	3.7.1	Match
	0.39	16.H	sdg16_rsf	Press Freedom Index	16.1	Closely aligned
	0.37	16.F	sdg16_homicides	Homicides	16.1.1	Match
	0.36	16.G	sdg16_justice	Access to and affordability of justice	16.3.1, 16.3.3	Closely aligned
	0.30	6.E	sdg6_water	Population using at least basic drinking water services	6.1.1	Closely aligned
	0.28	2.H	sdg2_wasting	Prevalence of wasting in children under 5 years of age	2.2.2	Match
	0.26	16.A	sdg16_admin	Timeliness of administrative proceedings	16.6	Closely aligned
	0.05	11.A	sdg11_pipedwat	Access to improved water source, piped	11.1	Closely aligned
	0.05	3.N	sdg3_vac	Surviving infants who received 2 WHO-recommended vaccines	3.b.1	Closely aligned
	0.00	8.F	sdg8_slavery	Victims of modern slavery	8.7	Closely aligned
	-0.01	16.J	sdg16_u5reg	Birth registrations with civil authority	16.9.1	Match
	-0.07	3.L	sdg3_u5mort	Mortality rate, under-5	3.2.1	Match
	-0.13	8.B	sdg8_adjgrowth	Adjusted GDP growth	8.1.1	Closely aligned
	-0.15	7.C	sdg7_elecac	Population with access to electricity	7.1.1	Match
	-0.15	5.A	sdg5_edat	Ratio of female-to-male mean years of education received	4.5.1	Closely aligned
	-0.15	10.A	sdg10_gini	Gini coefficient	10.1	Closely aligned
	-0.15	5.D	sdg5_parl	Seats held by women in national parliament	5.5.1	Match
	-0.19	11.D	sdg11_transport	Satisfaction with public transport	11.2.1	Closely aligned
	-0.19	4.B	sdg4_literacy	Literacy rate	4.6.1	Match
	-0.27	3.C	sdg3_hiv	New HIV infections	3.3.1	Match
	-0.37	3.A	sdg3_births	Births attended by skilled health personnel	3.1.2	Match
	-0.39	3.E	sdg3_matmort	Maternal mortality rate	3.1.1	Match
	-0.51	3.J	sdg3_tb	Incidence of tuberculosis	3.3.2	Match
	-0.57	4.C	sdg4_primary	Net primary enrollment rate	4.1.2	Closely aligned
	-0.82	5.C	sdg5_lfpr	Ratio of female-to-male labor force participation rate	5.5	Closely aligned
	-1.06	8.G	sdg8_unemp	Unemployment rate	8.5.2	Match
	-1.45	17.A	sdg17_cohaven	Corporate Tax Haven Score	-	Not in UNSTATS
	-1.60	8.D	sdg8_impstav	Victims of modern slavery embodied in imports	8.7	Closely aligned
6	-1.67	8.C	sdg8_impacc	Fatal work-related accidents embodied in imports	8.8.1	Closely aligned
	-1.68	2.F	sdg2_trophic	Human Trophic Level	-	Not in UNSTATS
	-1.69	16.K	sdg16_weaponsexp	Exports of major conventional weapons	16.1	Closely aligned
	-1.81	2.B	sdg2_obesity	Prevalence of obesity, BMI \geq 30	2.2	Closely aligned

Comment 1.10

10. Since the authors obtain their data from the Sustainable Development Report 2023, how do their findings compare with those in this report? To what extent these data affect the paper findings (compared with other data sources, such as from or the UN or the World Bank)?

Response to 1.10

Indeed, we have obtained the SDG data from the Sustainable Development Report (SDR) 2023. The SDR 2023 provides the primary data used in our analysis, as detailed in the Data Section of the Methods in our original submission. It is a reputable and widely used source that has undergone multiple peer reviews and has been utilized in numerous previous analyses.

To enhance the context in our manuscript, we have supplemented the introduction with additional information about the SDR dataset, particularly highlighting its advantages and disadvantages.

Lines 690-698:

In addition, the related SDG Index methodology and datasets have undergone multiple peer reviews and have been used to substantiate previous notable studies in this field^{6,66-68}. Naturally, the dataset has its advantages and disadvantages. The advantages include its broad coverage and consistency over time, which allow for meaningful comparisons across countries and time periods. The disadvantages include a lack of granularity needed to capture local (e.g., urban level) variations and specific economic activities (e.g., industries) within countries. Additionally, some indicators are not fully applicable to all countries (e.g., 14.C. *Mean area that is protected in marine sites important to biodiversity*).

Regarding the comparison with the findings from the Sustainable Development Report 2023, which ranked countries and evaluating their progress towards achieving the SDGs, this study has the following key features as added in the Discussion Section:

Lines 390-417:

Distinct from earlier notable studies that assessed countries' SDG performance³⁻⁸, such as those ranking countries and evaluating their progress towards achieving the SDGs⁶ or exploring development patterns¹²⁻¹⁹, our study introduces several key features that set it apart.

First, we employed a comparative framework to identify each country's relative strengths and weaknesses, as well as to pinpoint 'orphaned' SDG indicators in specific development areas. This comparative analysis, which contextualizes performance by benchmarking different variables or entities against each other, offers a clearer understanding of each country's relative standing. In contrast, absolute scoring (Figs. 3d-f) can obscure these insights by focusing solely on an overarching SDG score without the nuanced understanding that comes from comparison with other entities and SDGs.

Second, we utilized the 'product space' method and the broader economic complexity approach as effective tools for pattern recognition. These methods help identify national sustainable development trends and reveal whether countries specialize in certain goals. For instance, the derived GSI indicator divides SDG indicators into low-GSI indicators that countries tend to overlook and high-GSI indicators that countries tend to specialize in during their development process (Figs. 3, 5, 6). Additionally, we delineated the development trajectories of nations based on the historical performance of all countries, differing from the Sustainable

Development Report⁶, which evaluates whether a country is on track based on its individual performance.

Third, we introduced the concept of the ‘SDG space’ as a high-resolution tool to monitor countries’ performance across 96 SDG indicators. This tool visualizes countries’ relative performance and assists in guiding the formulation of development strategies (Supplementary Figs. 12-177). Moreover, we developed a dedicated website to present 3,818 SDG spaces for 166 nations spanning the years 2000 to 2022: <http://www.spacelab.team/#/SDGSpace>. Unlike the traditional product space²⁰, which typically ranks countries and guides them towards advanced products (e.g., motor vehicles and electronics), the SDG space is designed to assist countries monitor their progress within the SDG framework, identifying relative strengths and weaknesses, rather than ranking countries.

We would like to mention that, although this was referred to as a 'minor comment', addressing it alone required four full weeks of several rounds of data analysis and multiple discussions among the authorship team. This was due to the relatively poor quality of the UN SDG data, which necessitated extensive cleaning and standardization by our team—a time-consuming process. Following this, we had to construct a new SDG Space using the UN data for comparison and perform robustness checks.

As reviewers can now see in Supplementary Fig. 11, we identified a similar bipolar structure, which confirms the robustness of our original results. We have supplemented this information in the Robustness Check section of the Methods (please also refer to our response to Comment 1.7).

Lines 724-730:

(3) To validate the robustness of our results using different data sources, we collected additional data from the UNSD SDG database. For consistency in comparison, we included only the data on targets that matched the 96 indicators used in this study (see Tables 1 and 2 in Methods for the matching process). Data source: <https://unstats.un.org/sdgs/dataportal/database>. As shown in Fig. 1 (SDR 2023 database) and Supplementary Fig. 11 (UNSD SDG database), the SDG spaces exhibit a similar structure.

Supplementary Figure 11 | The SDG space in 2022 using UN SDG data. The node colour represents Goal Sustainability Index (GSI), and the node size indicates the node degree, representing the number of edges connected to the node. This SDG space was produced using data from the UNSD SDG targets framework, which were matched with the 96 SDG indicators (see Tables 1-2 in Methods). Data source: <https://unstats.un.org/sdgs/dataportal/database>.

Concluding remark

Thank you once again for all your invaluable suggestions, which have significantly enhanced the quality of our paper.

Reviewer #2

General Remarks of Reviewer #2

The manuscript provides a valuable contribution to the analysis of countries' pathways towards greater sustainability as envisioned in the SDG framework through the lens of development economics. The study compares the trajectories of 166 countries (from 2000-22), and explores whether countries' performance follows distinct patterns by considering 96 global indicators from the SDR. The approach chosen is systematic and sophisticated, employing methodologies innovatively in the context of sustainable development. Overall, the manuscript contributes to advancing knowledge in the field of sustainable development, and can inform more effective strategies for achieving the SDGs across different countries. Notably, I sincerely appreciate that the authors want to make their results on SDG spaces publicly available and easily accessible through the dedicated web-interface. However, there are some points I would like to raise which I hope can support the authors in improving the manuscript:

Response to 2

We acknowledge this summary of the content of our manuscript, and endorsement of our aims, analytical approach, and aspired contribution to literature and practice, including via a dedicated open-access web-interface. Thank you.

We further thank you for your comments and suggestions. Please see our response to your comments in the following.

Comment 2.1

1. The authors' choice of referring to the assessed indicators with capital letters (such as 1A, 1B...) can be misleading, as some the official SDG targets referring to the means of implementation under the respective goals are named alike, although with lower case letters (1a, 1b, ...). If the authors want to stick to this format, they should be explicit about the distinction.

Response to 2.1

Thank you for this remark. In the previous version, we explained the difference in indexing SDG targets and indicators by the UN and by the 2023 Sustainable Development Report (upon which this study relies).

This explanation was included in the SI (Supplementary Table 1), not in the main text.

Following this comment, and Comment 1.9 above, we have provided an explanation of the

indexing system in the main text and transferred Supplementary Table 1 into the Methods section.

We have modified the text as follows (Lines 699-707):

Table 1 | Environment-related SDG indicators ranked by the Goal Sustainability Index (GSI) and their alignment with UN SDG taxonomies. The environment-related indicators are divided into three groups based on their GSI values. Group 1: $GSI > 0.5$. Group 2: $-1.0 \leq GSI \leq 0.5$. Group 3: $GSI < -1.0$.

Group	GSI in 2022	Indicator code in this study	Indicator code in SDR 2023	Indicator definitions	UNSD target	UNSD match
1	1.79	6.D	sdg6_wastewat	Anthropogenic wastewater that receives treatment	6.3.1	Match
	1.08	7.A	sdg7_cleanfuel	Population with access to clean fuels and technology for cooking	7.1.2	Closely aligned
	0.83	3.H	sdg3_pollmort	Age-standardized death rate attributable to household air pollution and ambient air pollution	3.9.1	Match
	0.79	14.B	sdg14_cleanwat	Ocean Health Index: Clean Waters score	14.1.1	Closely aligned
2	0.27	14.C	sdg14_cpma	Mean area that is protected in marine sites important to biodiversity	14.5.1	Closely aligned
	0.27	2.C	sdg2_pestexp	Exports of hazardous pesticides	3.9	Closely aligned
	0.19	15.D	sdg15_forchg	Permanent deforestation	15.2	Closely aligned
	0.05	7.D	sdg7_renewcon	Renewable energy share in total final energy consumption	7.2.1	Match
	-0.03	11.B	sdg11_pm25	Annual mean concentration of particulate matter of less than 2.5 microns in diameter (PM2.5)	11.6.2	Match
	-0.21	14.E	sdg14_fishstocks	Fish caught from overexploited or collapsed stocks	14.4.1	Closely aligned
	-0.27	15.C	sdg15_cpta	Mean area that is protected in terrestrial sites important to biodiversity	15.1.2	Match
	-0.30	15.B	sdg15_cpfa	Mean area that is protected in freshwater sites important to biodiversity	15.1.2	Match
	-0.34	14.D	sdg14_discard	Fish caught that are then discarded	14.4	Closely aligned
	-0.34	14.F	sdg14_trawl	Fish caught by trawling or dredging	14.4	Closely aligned
	-0.57	15.E	sdg15_redlist	Red List Index of species survival	15.5.1	Match
	-0.65	2.D	sdg2_snmi	Sustainable Nitrogen Management Index	2.4	Closely aligned
	-0.72	13.A	sdg13_co2export	CO2 emissions embodied in fossil fuel exports	13.2	Closely aligned
	-0.82	6.A	sdg6_freshwat	Freshwater withdrawal	6.4.2	Match
	-0.95	7.B	sdg7_co2twh	CO2 emissions from fuel combustion per total electricity output	7.2	Closely aligned
	3	-1.26	12.B	sdg12_explastic	Exports of plastic waste	12.4
-1.40		14.A	sdg14_biomar	Marine biodiversity threats embodied in imports	14.4	Closely aligned
-1.54		12.E	sdg12_nprod	Production-based nitrogen emissions	9.4	Closely aligned
-1.59		12.D	sdg12_nimport	Nitrogen emissions embodied in imports	9.4	Closely aligned
-1.59		13.C	sdg13_co2import	CO2 emissions embodied in imports	13.2	Closely aligned
-1.62		12.A	sdg12_ewaste	Electronic waste	12.4.2	Match
-1.64		15.A	sdg15_biofrwter	Terrestrial and freshwater biodiversity threats embodied in imports	15.5	Closely aligned
-1.66		12.F	sdg12_so2import	SO2 emissions embodied in imports	9.4	Closely aligned
-1.67		6.C	sdg6_scarcew	Scarce water consumption embodied in imports	6.4	Closely aligned
-1.67		12.C	sdg12_msw	Municipal solid waste	12.5	Closely aligned
-1.76		13.B	sdg13_co2gcp	CO2 emissions from fossil fuel combustion and cement production	13.2.2	Closely aligned
-1.84		12.G	sdg12_so2prod	Production-based SO2 emissions	9.4	Closely aligned

Table 2 | Non-environment-related SDG indicators ranked by the Goal Sustainability Index (GSI) and their alignment with UN SDG taxonomies. The non-environment-related indicators are divided into three groups based on their GSI values. Group 4: $GSI > 0.5$. Group 5: $-1.0 \leq GSI \leq 0.5$. Group 6: $GSI < -1.0$.

Group	GSI in 2022	Indicator code in this study	Indicator code in SDR 2023	Indicator definitions	UNSD target	UNSD match	
4	1.98	9.E	sdg9_rdex	Expenditure on research and development	9.5.1	Match	
	1.87	9.A	sdg9_articles	Articles published in academic journals	9.5	Closely aligned	
	1.30	3.G	sdg3_neonat	Neonatal mortality rate	3.2.2	Match	
	1.29	3.M	sdg3_uhc	Universal health coverage (UHC) index of service coverage	3.8.1	Match	
	1.23	8.A	sdg8_accounts	Adults with an account at a bank or other financial institution or with a mobile-money-service provider	8.10.2	Match	
	1.17	11.C	sdg11_slums	Proportion of urban population living in slums	11.1.1	Match	
	1.08	9.G	sdg9_uni	The Times Higher Education Universities Ranking: Average score of top 3 universities	-	Not in UNSTATS	
	1.08	3.I	sdg3_swb	Subjective well-being	3.4	Closely aligned	
	1.08	9.C	sdg9_lpi	Logistics Performance Index: Quality of trade and transport-related infrastructure	9.1	Closely aligned	
	1.07	16.C	sdg16_cpi	Corruption Perceptions Index	16.5.1, 16.5.2	Closely aligned	
	1.06	3.K	sdg3_traffic	Traffic deaths	3.6.1	Match	
	1.05	1.A	sdg1_lmipov	Poverty headcount ratio at \$3.65/day	1.1.1	Match	
	1.04	4.D	sdg4_second	Lower secondary completion rate	4.1.2	Match	
	1.03	3.D	sdg3_lifex	Life expectancy at birth	3.1:3.9	Closely aligned	
	1.02	9.B	sdg9_intuse	Population using the internet	17.8.1	Match	
	1.01	9.F	sdg9_roads	Rural population with access to all-season roads	9.1.1	Match	
	1.01	4.A	sdg4_earlyedu	Participation rate in pre-primary organized learning	4.2.2	Closely aligned	
	0.98	3.B	sdg3_fertility	Adolescent fertility rate	3.7.2	Match	
	0.97	6.B	sdg6_sanita	Population using at least basic sanitation services	6.2.1	Closely aligned	
	0.95	2.E	sdg2_stunting	Prevalence of stunting in children under 5 years of age	2.2.1	Match	
	0.94	9.D	sdg9_mobuse	Mobile broadband subscriptions	9.c.1, 17.6.1	Closely aligned	
	0.91	2.G	sdg2_undernsh	Prevalence of undernourishment	2.1.1	Match	
	0.84	17.C	sdg17_statperf	Statistical Performance Index	17.18.1: 17.19.2	Closely aligned	
	0.81	16.E	sdg16_exprop	Expropriations are lawful and adequately compensated	16.6	Closely aligned	
	0.75	3.F	sdg3_ncds	Age-standardized death rate due to cardiovascular disease, cancer, diabetes, or chronic respiratory disease in adults aged 30–70 years	3.4.1	Match	
	0.66	2.A	sdg2_crlyld	Cereal yield	2.3, 2.4	Closely aligned	
	0.66	17.B	sdg17_govex	Government spending on health and education	1.a.1	Closely aligned	
	0.58	16.B	sdg16_clabor	Children involved in child labor	8.7.1	Closely aligned	
	0.52	8.E	sdg8_rights	Fundamental labor rights are effectively guaranteed	8.8.2	Match	
	0.51	16.I	sdg16_safe	Population who feel safe walking alone at night in the city or area where they live	16.1.4	Match	
	5	0.47	10.B	sdg10_palma	Palma ratio	10.1	Closely aligned
		0.47	16.D	sdg16_detain	Unsentenced detainees	16.3.2	Match

Group	GSI in 2022	Indicator code in this study	Indicator code in SDR 2023	Indicator definitions	UNSD target	UNSD match
	0.42	1.B	sdg1_wpc	Poverty headcount ratio at \$2.15/day	1.1.1	Match
	0.42	5.B	sdg5_familypl	Demand for family planning satisfied by modern methods	3.7.1	Match
	0.39	16.H	sdg16_rsf	Press Freedom Index	16.1	Closely aligned
	0.37	16.F	sdg16_homicides	Homicides	16.1.1	Match
	0.36	16.G	sdg16_justice	Access to and affordability of justice	16.3.1, 16.3.3	Closely aligned
	0.30	6.E	sdg6_water	Population using at least basic drinking water services	6.1.1	Closely aligned
	0.28	2.H	sdg2_wasting	Prevalence of wasting in children under 5 years of age	2.2.2	Match
	0.26	16.A	sdg16_admin	Timeliness of administrative proceedings	16.6	Closely aligned
	0.05	11.A	sdg11_pipedwat	Access to improved water source, piped	11.1	Closely aligned
	0.05	3.N	sdg3_vac	Surviving infants who received 2 WHO-recommended vaccines	3.b.1	Closely aligned
	0.00	8.F	sdg8_slavery	Victims of modern slavery	8.7	Closely aligned
	-0.01	16.J	sdg16_u5reg	Birth registrations with civil authority	16.9.1	Match
	-0.07	3.L	sdg3_u5mort	Mortality rate, under-5	3.2.1	Match
	-0.13	8.B	sdg8_adjgrowth	Adjusted GDP growth	8.1.1	Closely aligned
	-0.15	7.C	sdg7_elecac	Population with access to electricity	7.1.1	Match
	-0.15	5.A	sdg5_edat	Ratio of female-to-male mean years of education received	4.5.1	Closely aligned
	-0.15	10.A	sdg10_gini	Gini coefficient	10.1	Closely aligned
	-0.15	5.D	sdg5_parl	Seats held by women in national parliament	5.5.1	Match
	-0.19	11.D	sdg11_transport	Satisfaction with public transport	11.2.1	Closely aligned
	-0.19	4.B	sdg4_literacy	Literacy rate	4.6.1	Match
	-0.27	3.C	sdg3_hiv	New HIV infections	3.3.1	Match
	-0.37	3.A	sdg3_births	Births attended by skilled health personnel	3.1.2	Match
	-0.39	3.E	sdg3_matmort	Maternal mortality rate	3.1.1	Match
	-0.51	3.J	sdg3_tb	Incidence of tuberculosis	3.3.2	Match
	-0.57	4.C	sdg4_primary	Net primary enrollment rate	4.1.2	Closely aligned
	-0.82	5.C	sdg5_lfpr	Ratio of female-to-male labor force participation rate	5.5	Closely aligned
	-1.06	8.G	sdg8_unemp	Unemployment rate	8.5.2	Match
	-1.45	17.A	sdg17_cohaven	Corporate Tax Haven Score	-	Not in UNSTATS
	-1.60	8.D	sdg8_imp Slav	Victims of modern slavery embodied in imports	8.7	Closely aligned
6	-1.67	8.C	sdg8_impacc	Fatal work-related accidents embodied in imports	8.8.1	Closely aligned
	-1.68	2.F	sdg2_trophic	Human Trophic Level	-	Not in UNSTATS
	-1.69	16.K	sdg16_weaponsexp	Exports of major conventional weapons	16.1	Closely aligned
	-1.81	2.B	sdg2_obesity	Prevalence of obesity, BMI \geq 30	2.2	Closely aligned

Comment 2.2

2. Relatedly, the authors could mention earlier in the manuscript that the study uses (proxy) indicators for assessing performance on SDG targets from the SDR (not SDG targets), and that these not always correspond to the official SDG indicators. In this respect, they could further

explain why they decided to do so (I suppose due to issues of data availability in the official data set). In Supplementary table 1, consider adding a column indicating whether the indicator is an official SDG indicator (“UNSD Match” in column S, tab “Codebook” in SDR2023_data excel file).

Response to 2.2

Thank you for this suggestion. In response to this comment and Comment 2.1 above, we have added the “UNSD Match” in new tables, Tables 1-2. Please refer to the tables provided in our response to Comment 2.1. Additionally, we have changed the term “targets” to “indicators,” which are obtained from the SDR.

Following your suggestion, we have also introduced the selection of these SDG indicators at the beginning of the main text.

Lines 109-113:

The indicators covered in this study are carefully indexed and evaluated in the Sustainable Development Report (SDR) 2023⁶, with details provided in Tables 1-2 of Methods, including their relationship to the official UNSD SDG targets. Please note that, for clarity in the figures, the SDG indicator abbreviations used in this study (e.g., 1.A) are simplified versions of the indicators provided in the SDR report and do not correspond directly to the UNSD SDG targets.

Comment 2.3

3. While being aware that the terms “developed” vs. “developing” countries is commonly used, I would encourage the authors to refrain from this terminology, particularly since they do not specify where this categorization stems from and is based on. In my opinion, it would be better to distinguish between countries by income levels (from low- to high-income countries) and rename the groups accordingly.

Response to 2.3

Thank you for this suggestion. Following your comment, we have replaced the terms "developed" and "developing" countries with "countries at a primary stage of development" and "countries at an advanced stage of development", respectively, to align with our categorization of countries into quartiles based on their SDG index score rankings — Stage 1 (Primary stage), Stage 2 (Low-medium stage), Stage 3 (High-medium stage), and Stage 4 (Advanced stage).

Comment 2.4

4. In this respect, I strongly encourage the authors to revise the regional groupings in Fig. 2. If the authors want to display regional differences, they should stick to official regional groupings

(e.g., (aggregates of) those indicated in SDR, or the M49 standard). Distinguishing between Africa/East & South Asia/Latin America and the Caribbean vs. Industrialized Countries is an unfortunate choice.

Response to 2.4

Thank you for this suggestion. Following your comment, we have revised the regional groupings according to M49 standard.

We have modified the text as follows:

Lines 170-181:

Fig. 2 presents strong regional specialization patterns within the SDG space. The SDG indicators with $RCA > 0$ for each region are colour-coded, indicating that the region has a relative advantage in these indicators. Africa occupies the red cluster on the left side of the SDG space (Fig. 2a), primarily associated with overnutrition, embodied social and environmental impacts in international trade, and waste and emissions. In contrast, Europe occupies the blue cluster on the right side (Fig. 2d), which includes indicators related to poverty and hunger reduction, good health and well-being, education, innovation, and government administration. The Americas, Asia, and Oceania regions exhibit relatively similar patterns (Figs. 2b, 2c, 2e), with a more even distribution and smaller RCA values for their specialized indicators. However, Oceania has more advantages in high-GSI indicators within the blue cluster. The SDG space of all 166 nations ranked by SDG scores in 2000, 2015, and 2022, are shown in Supplementary Figs. 12-177.

Fig. 2 | The SDG space of different regions of the world in 2022. Panel a, Africa. Panel b, Americas. Panel c, Asia. Panel d, Europe. Panel e, Oceania. The node colour represents the Goal Sustainability Index (GSI). The coloured SDG indicators are those with $RCA > 0$, indicating that the region has a revealed comparative advantage (or specialization) in these indicators. The node size corresponds to the RCA value for each indicator.

Comment 2.5

5. The findings and insights derived from the method may not be universally applicable to all countries or regions, as they are based on aggregated data and global trends. The authors might want to acknowledge more explicitly that local variations in socio-economic conditions, culture, governance, policy environments, and institutional capacities may not be adequately captured,

but might be necessary to consider for providing a more comprehensive understanding of SDG development patterns.

Response to 2.5

We agree that local variations in socio-economic conditions, culture, governance, policy environments, and institutional capacities are crucial factors that may not be fully captured by our aggregated data and global trends.

In response, we have explicitly acknowledged this limitation in the revised manuscript and emphasized the importance of considering these local variations when using SDG development patterns to guide countries' policies.

Additionally, we have introduced three more representative countries to demonstrate how our SDG evaluation framework can be applied in different contexts. Please also refer to our complementary response (and discussion in response) to Comment 2.10, below.

The detailed revisions are as follows:

Lines 181-183:

It is notable that the findings and insights derived from this method may not be universally applicable to all countries in each region, given the local variations in socio-economic conditions, culture, governance, and policy environments.

In view of this Comment, and Comments 2.7 and 3.2, we have added a subsection entitled "Limitations and clarifications on the Product Space terminology", to acknowledge the data limitations, clarify the use of the Product Space techniques and terminology, and discuss their implications.

Lines 504-556:

Limitations and clarifications on the Product Space terminology

In this study, we constructed 'SDG spaces' for 166 nations from 2000 to 2022, revealing a bipolar world while identifying 'orphaned' development indicators by different nations. These patterns, along with the SDG space, provide a high-resolution tool for evaluating progress and offering targeted guidance for countries in achieving the SDGs. In doing so, we have applied the Product Space and Economic Complexity framework, analytical techniques, and terminology, including 'revealed comparative advantage' (RCA) and 'specializations'.

We acknowledge that some of the sustainable development indicators for which certain countries exhibit an RCA (a specialization) might reflect an intentional, premeditated national preference over time, such as the environmental impacts of imports, as part of a larger importation policy. While our data allow us to identify which advantages — or, as they are also referred to in this study (and in Economic Complexity) 'specializations' — are likely to diminish as a country develops, they do not enable us to differentiate between intentional (i.e., preferred, premeditated) and unintentional specializations. Indeed, 'unintentional

specializations' may arise due to a country's geography or geology, such as being landlocked or possessing mineral resource reserves⁶⁰. Additionally, other detrimental factors, such as colonial legacy, may predetermine a country's development trajectory⁶⁰.

To clarify, the use of RCA to discuss developing countries' performance on certain SDG indicators is done within the context of the Product Space approach. However, it does not imply that these countries have intentionally pursued this RCA, nor does it suggest that the historical reasons and circumstances underpinning it are inherently 'advantageous'. Using the terms 'advantage' or 'specialization' outside the context of the Product Space approach – when referring to a country meeting certain SDG targets – may be misleading, especially when these 'advantages' may reflect low-income status rather than deliberate policies or capabilities. For example, several indicators, such as lower environmental footprints and reduced health risks like obesity, may result from limited resources. At the same time, this does not inherently render them disadvantages — a low ecological footprint can be considered a positive attribute, regardless of its underlying causes. This invites further discussion and underscores the need for future research.

Moreover, our intention is not to suggest that so-called developing countries (or any country, for that matter) should maintain their current status to preserve these 'advantages' (RCAs). Instead, the aim of this study is to use the RCA framework to identify national sustainable development trajectories as well as areas where targeted interventions are needed to enhance development outcomes.

In the same vein, the degree of agency we can attribute to national governments in such circumstances (e.g., 'being landlocked', 'being resource-endowed', and 'being post-colonial') when developing a relative specialization remains a matter of debate. Addressing the distinction between intentional and unintentional specializations, as well as the role of national governments' agency, requires further study and may involve subjective judgments. While this study opens new avenues for such discussions, the interpretation of RCAs as normative preferences lies outside its scope.

On a related note, it should be emphasized that local variations at subnational levels and units of analysis (regional, provincial, urban, peri-urban, rural) in socio-economic conditions, culture, governance, policy environments, and institutional capacities are important factors that may not be fully captured by our aggregated data and global trends. We acknowledge that even within the same country, SDG performance and strategies may differ between regions and between urban and rural areas⁶¹. Due to data limitations, this study focuses on global and national sustainable development through 96 SDG indicators. Future research could extend this analysis to regional, state, and city levels.

Lastly, disparities in data availability across countries and economic sectors could lead to biased representations of SDG performance, limiting the generalizability of the findings. With increasing calls to improve the SDG database^{62,63}, future studies should critically and comparatively investigate SDG indicators and the pathways toward their realization.

Comment 2.6

6. The authors also might want to acknowledge that some indicators are not applicable to all countries in their entirety (e.g., indicator 14.C on protected areas in marine sites not relevant for landlocked countries).

Response to 2.6

Thank you for noticing this. We acknowledge that some indicators, such as indicator 14.C, are not applicable to all countries. Following this comment, we have added a note to address this limitation in the Data section of the Methods.

The detailed revision is as follows:

Data section of Methods (Lines 692-698):

Naturally, the dataset has its advantages and disadvantages. The advantages include its broad coverage and consistency over time, which allow for meaningful comparisons across countries and time periods. The disadvantages include a lack of granularity needed to capture local (e.g., urban level) variations and specific economic activities (e.g., industries) within countries. Additionally, some indicators are not fully applicable to all countries (e.g., 14.C. *Mean area that is protected in marine sites important to biodiversity*).

Comment 2.7

7. L129-31: I think this statement should be rephrased and needs further (potential) explanation. First, some of the indicators for which “developing” countries have an RCA could indeed be a preference (e.g. environmental impacts of imports). Second, your data allows you for stating which advantages/specializations are likely to be decreasing when a country “develops”. But does this qualify to speak about (un)intentional specialization? Maybe it would be useful to further reflect on the RCA construction and its limitations/implications in this respect as well.

Response to 2.7

Thank you for this comment.

We acknowledge that some of the sustainable development indicators for which “developing” countries exhibit an RCA might reflect a national preference over time, such as the environmental impacts of imports, as part of a larger importation policy. While our data allows us to identify which advantages, or as they are referred to in this study, ‘specializations’, are likely to diminish as a country develops, it does not enable us to differentiate between intentional (preferred) and unintentional specializations. In this regard, unintentional specializations may arise due to a country’s geography or geology, such as being landlocked or possessing mineral resource reserves (*Sachs, 2006*). Additionally, other detrimental factors,

such as colonial legacy, may predetermine a country's development trajectory (*Sachs, 2006*). The extent of agency we attribute to national governments in such circumstances to develop a relative specialization remains a matter of debate. Addressing this distinction between intentional and unintentional specializations requires further study and might involve subjective judgments. This study opens new vistas to such discussions, yet the interpretation of RCAs as preferences lies outside its scope.

In view of this Comment, and Comments 2.5 and 3.2, we have added a subsection entitled "Limitations and clarifications on the Product Space terminology", to acknowledge the data limitations, clarify the use of the Product Space techniques and terminology, and discuss their implications.

Please refer to our response to Comment 2.5 above, and to lines 503-555 in the revised manuscript.

Reference: *Sachs, J. D. (2006). The end of poverty: Economic possibilities for our time. Penguin.*

Comment 2.8

8. The meaning and interpretation of the network links (rather called edges, and I would recommend to use this term consistently throughout the manuscript) between SDGs and the degree values are barely being addressed in the text.

Response to 2.8

Thank you for your suggestion. Following this comment we have modified the text by changing 'links' into 'edges', and added a sentence as follows (Lines 131-132):

The indicators in these two clusters have relatively higher node degrees, indicating that they are connected to a larger number of edges (Fig. 1a).

Since the network edges and node degrees are not the focus of this study, we did not delve into their interpretation.

Comment 2.9

9. L286-90: I would appreciate if the authors could elaborate on how they interpret these findings.

Response to 2.9

In view of this comment, we have added the interpretation of these findings as follows:

Lines 337-343:

These anomalies can be attributed to distinctive industrial structures (i.e., the composition and

organization of industries within a country's economy) or national cultures. For instance, Serbia's nitrogen management and Moldova's CO₂ emissions from fossil fuel combustion and cement production are outcomes of their specific industrial frameworks. Similarly, Japan's traditional dietary practices and culinary habits (i.e., its food culture) influence its unique sustainable development indicators.

Comment 2.10

10. To better highlight and acknowledge the differences shown in Fig.6, the discussion section would benefit from an additional example besides China.

Response to 2.10

Thank you for your suggestion. In response, we have added three additional countries as case studies in the Discussion Section. We have modified the text as follows:

Lines 418-432:

In terms of policy guidance, the comparative insights and identified trends in sustainable development trajectories can provide targeted recommendations (Fig. 8). From a static perspective, countries' performance on the indicators can be categorized into four areas (Fig. 8b, d, f, h): Area I (high RCA and SDG Scores) represents an ideal situation. Area II (high RCA but low SDG scores) where indicators are relatively difficult for a specific country to improve, as other countries are also likely to struggle in making progress and providing successful examples. Area III (low RCA and SDG Scores) represents the worst situation, where both relative and absolute performance on these indicators is poor. Area IV (low RCA but relatively high SDG scores) where indicators may be easier to improve, as other countries might be performing better and could offer models for success.

Taking the USA as an example (Fig. 8h), beyond the indicators in Area III — such as overnutrition (2.B, 2.F) and waste and emissions (12.A, 12.C, 12.E, 13.B) — focus should also be placed on addressing social and environmental impacts linked to imports (8.D, 12.F, 14.A), inequality (10.A), and Corporate Tax Haven (17.A) in Area IV, despite these indicators having absolute scores above the global average.

Lines 452-469:

Fig. 8 | The SDG space and indicator performance for four representative countries in 2022. Panels a-b, Ethiopia (Stage 1). Panels c-d, India (Stage 2). Panels e-f, China (Stage 3). Panels g-h, United States (Stage 4). In panels a-h, blue nodes represent the top 20 SDG indicators by RCA values, while red nodes represent the bottom 20. The remaining indicators are not highlighted. The size of each node corresponds to the absolute RCA value. Countries are evenly divided into four stages according to their absolute score rankings: Stage 1 (Primary stage), Stage 2 (Low medium stage), Stage 3 (High medium stage), and Stage 4 (Advanced stage). Stage 4 is further divided into two halves to identify the top 20 increasing and decreasing

indicators in RCA for the USA (in the first half of Stage 4) between its current stage and the next stage (see Supplementary Table 1). In panels **b, d, f, h**, scatter diagrams are divided into four areas by the lines at $RCA = 0$ and the global average SDG score of 63. Area I: Countries with high RCA and SDG Scores for these indicators have reached a relatively ideal situation. Area II: Countries with high RCA but low SDG scores for these indicators face relative difficulties in improving them, likely due to low global averages for these indicators. Area III: Countries with low RCA and SDG Scores for these indicators show both relative and absolute underperformance, indicating that these indicators should be prioritized for improvement. Area IV: Countries with low RCA but relatively high SDG scores for these indicators demonstrate poor relative performance but strong absolute performance.

Comment 2.11

11. The interpretation of proximity values as indicators of synergy and trade-off should be cautiously considered, recognizing the complexity of sustainable development dynamics.

Response to 2.11

We recognize the complexity of sustainable development dynamics and agree that interpreting proximity values as indicators of synergy and trade-off should be approached with caution.

Following this comment, we have added a clarification in the Methods section as follows:

Lines 610-611:

The proximity between SDG indicators in the nation-indicator bipartite network is defined by the conditional probability that two indicators are co-specialized within a nation.

Comment 2.12

12. Finally but importantly, some of the research aims set out in the beginning in the manuscript (whether countries taking different paths have achieved different degrees of success / whether and under which circumstances countries make progress towards their SDG commitment) have not been sufficiently answered by the authors.

While I highly appreciate the extensive data and visualization presented (also in the supplementary material), the authors could have used more space for interpreting and reflecting on their results (sometimes less can be more).

Response to 2.12

Based on the findings and contributions of this study, we have rephrased the research aims as follows:

Lines 55-59:

The year of 2023 marked the halfway point in the implementation of the SDGs², prompting us to question whether universal patterns underpinned nations' sustainable development trajectories, and whether some SDG indicators were neglected or overlooked (referred to herein as 'orphaned') in specific areas, across nations, and over time.

Lines 74-76:

Specifically, we examined whether countries have so-called 'orphaned' SDG indicators in certain development areas and whether there are underlying patterns or rules governing these 'orphaned' areas.

Regarding your comment on interpreting and reflecting on the results, we have added more in-depth interpretation of our findings (e.g., please see our response to Comment 2.9), supplemented additional cases (please see our response to Comment 2.10), and compared our results with previous studies (please see our response to Comment 1.4).

Comment 2.13

13. L52: Please correct "constitute" to "constitutes"

Response to 2.13

Thank you for noticing this. We have corrected the text.

Comment 2.14

14. L55: Are we not already beyond the halfway point to meet the SDGs?

Response to 2.14

Thank you for noticing this. Please see our answer to comment 2.12 above.

Comment 2.15

15. L62: Please be consistent in the use of italics for goal descriptions

Response to 2.15

Thank you for noticing this. We have corrected the text accordingly.

Comment 2.16

16. L65: Key development dimensions such as?

Response to 2.16

In light of this comment, we have modified the text as follows:

Lines 68-69:

...explored key development dimensions (e.g., socioeconomic development, environment, and equality)

Comment 2.17

17. L68: Variables or entities such as?

Response to 2.17

In light of this comment, we have changed ‘variables or entities’ into ‘goals or indicators’.

Lines 71-73:

...that is, comparing different goals or indicators against each other to understand the relative performance or characteristics of each entity in relation to the others.

Comment 2.18

18. L94-96: Briefly describe here what “specialization” in the RCA means in your study

Response to 2.18

In light of this comment, we have modified the text as follows.

Lines 99-101:

For example, if a country’s score for a specific SDG indicator constitutes a higher proportion of its total score across all indicators compared to the world average, that country is considered to be specialized in that SDG indicator.

Comment 2.19

19. L98: Explain here briefly what the CSI and GSI cover

Response to 2.19

In light of this comment, we have modified the text as follows:

Lines 104-108:

The rationale behind the calculation of GSI and CSI indicates that high-CSI countries (most of which are at an advanced stage of development), are more likely to dominate high-GSI indicators, whereas low-CSI countries (most of which are at a primary stage of development) tend to dominate low-GSI indicators overall (see Methods).

Comment 2.20

20. L102: I would change “correlation” to “relationship to” to avoid misunderstandings (or correspondence, as used in Supplementary Table 1)

Response to 2.20

Thank you for this suggestion. We have modified this following your suggestion.

Comment 2.21

21. L184 vs L194: Parenthesis “(Fig 3a-c)” inconsistent

Response to 2.21

Thanks for noticing this. We have modified the text as follows.

Lines 199-200:

The first trend is characterised by a continuous decrease in specialization in low-GSI indicators represented in red in Figs. 3a-c.

Comment 2.22

22. L210-11: Please elaborate on what you mean with “moved ahead” in a few words

Response to 2.22

In light of this comment, we have modified the text as follows.

Lines 224-227:

From a dynamic analysis spanning the years 2000 to 2022, it is observed that while the first

two trends have remained relatively stable, the third trend - particularly the emphasis on goals related to innovation - has advanced, appearing in earlier window groups.

Comment 2.23

23. L241-42: Please add here that countries were sorted from lowest to highest SDG rank

Response to 2.23

Thank you for this suggestion. We have revised the text as follows.

Lines 242-244:

To address this, we determined the window size by grouping 50 countries at a time, resulting in 117 window groups (i.e., countries 1-50, 2-51, ..., 117-166, with countries sorted from lowest to highest SDG rank).

Comment 2.24

24. Fig 5: Add headings (Stage 1-4) similar to other figures in manuscript

Response to 2.24

Thank you for noticing this. We have revised this accordingly.

Comment 2.25

25. L246: “The conditions of 17 respective goals” could be reformulated to be more precise

Response to 2.25

In light of this comment, we have modified the text as follows.

Lines 247-248:

The change of RCA of SDG indicators across 17 goals for each window group, aligned with the rank of SDG scores in 2022.

Comment 2.26

26. L322: When speaking of SDGs related to “economic development and basic livelihood security (e.g., SDGs 1,2,3,4,7)”, you should clearly include SDG 8 and might want to consider also SDG 11

Response to 2.26

In light of this comment, we have modified the text as follows.

Lines 377-378:

For instance, the SDGs related to economic development and basic livelihood security (e.g., SDGs 1, 2, 3, 4, 7, 8, 11).

Comment 2.27

27. L324-27: You could also refer here more strongly to SDG-related research, indicating a general trend in prioritization of social and economic SDGs over environmental ones (e.g. Chapter 6: “Planetary integrity” (Kotzé et al., 2022) in “The Political Impact of the Sustainable Development Goals Transforming Governance Through Global Goals?”, pp. 140 – 171; DOI: <https://doi.org/10.1017/9781009082945.007>

Response to 2.27

Thank you for this suggestion. We have added this reference accordingly.

Comment 2.28

28. L369: “and the other targets are not shown” – consider rephrasing this

Response to 2.28

In light of this comment, we have modified the text as follows.

Line 456:

...the remaining indicators are not highlighted in colour.

Comment 2.29

29. L370: “countries” needs to be written with a capital letter

Response to 2.29

Thank you for noticing this. We have modified this mistake.

Comment 2.30

30. L377-79: Consider is you can justify the sentence “these are not their active choices” based on your analysis (see my comment above)

Response to 2.30

Thank you for your comment. We have revised this sentence as follows:

Lines 473-475:

However, they may lose these relative advantages as they progress in their pursuit of greater wealth (i.e., income) and well-being.

This change is related to our response to comment 2.7 above.

Comment 2.31

31. L398-402: These limitations should be addressed more explicitly earlier in the manuscript (see my comment above)

Response to 2.31

In view of this comment, we have addressed this limitation in the Results Section, and stated it again in the Discussion Section. Please also see our response to Comments 2.5, 2.7 and 3.2.

Lines 181-183:

It is notable that the findings and insights derived from this method may not be universally applicable to all countries in each region, given the local variations in socio-economic conditions, culture, governance, and policy environments.

Lines 545-552:

On a related note, it should be emphasized that local variations at subnational levels and units of analysis (regional, provincial, urban, peri-urban, rural) in socio-economic conditions, culture, governance, policy environments, and institutional capacities are important factors that may not be fully captured by our aggregated data and global trends. We acknowledge that even within the same country, SDG performance and strategies may differ between regions and between urban and rural areas⁶¹. Due to data limitations, this study focuses on global and national sustainable development through 96 SDG indicators. Future research could extend this analysis to regional, state, and city levels.

Comment 2.32

32. L434-455: If I understand correctly, GSI (similar to PCI) aims at reflecting the complexity in achieving more sustainable outcomes. I would suggest including this explanation in this subsection, and maybe also already earlier in the manuscript

Response to 2.32

In this study, the country sustainability index (CSI) and goal sustainability index (GSI) are primarily used within a spectral clustering algorithm.

While it is true that CSI values are positively correlated with the SDG scores of countries and may indicate the complexity of achieving more sustainable outcomes to some extent, this is not the primary reason for using these indexes.

Therefore, we did not state in the main text that CSI and GSI reflect the complexity you refer to.

We explain these two indexes (CSI and GSI) as follows:

Lines 101-108:

Inspired by the economic complexity index (ECI) and product complexity index (PCI) set out by Hidalgo and Hausmann²³, we use a method equivalent to a clustering algorithm^{21,22} to calculate the country sustainability index (CSI) and goal sustainability index (GSI) based on countries' RCA in SDG indicators. The rationale behind the calculation of GSI and CSI indicates that high-CSI countries (most of which are at an advanced stage of development), are more likely to dominate high-GSI indicators, whereas low-CSI countries (most of which are at a primary stage of development) tend to dominate low-GSI indicators overall (see Methods).

Lines 154-160 (in the caption of Fig. 1):

GSI and CSI are twin clustering indicators, which are calculated based on RCA of indicators using the method developed by Hidalgo and Hausmann²³ (see Methods). The algorithm is equivalent to finding the eigenvalues of a matrix and is related to a spectral clustering algorithm that divides SDG indicators (or countries) into two groups: those with higher and lower GSI values (or countries with higher and lower CSI values). As a result of this reflections algorithm, countries with high CSI values tend to dominate the high-GSI indicators, and vice versa.

Lines 323-326:

The GSI indicator, derived from the product space and economic complexity approach, categorizes SDG indicators that countries tend to over- or under-specialize in during their development process. This categorization can help inform the formulation of sustainable development policies.

Lines 402-405:

For instance, the derived GSI indicator divides SDG indicators into low-GSI indicators that countries tend to overlook and high-GSI indicators that countries tend to specialize in during their development process (Figs. 3, 5, 6).

Comment 2.33

33. L438: Please correct method “is” instead of “are”

Response to 2.33

Thank you for noticing this. We have corrected the text accordingly.

Comment 2.34

34. L441: You refer to SDG score in the text, but to SDG index value in Supplementary Fig. 8 – please check your figures, text and tables for consistency

Response to 2.34

Thank you for noticing this. We have corrected the text, following your suggestion.

Comment 2.35

35. L450: Further details on the criteria for setting initial values would be helpful

Response to 2.35

In view of this comment, we have modified the text as follows.

Lines 600-603:

After setting initial values for the CSI and GSI (e.g., the number of indicators a country specializes in, and the number of countries that specialize in each indicator), we calculate a country’s CSI by taking the average GSI values of the SDG indicators in which the country has a relative advantage.

Comment 2.36

36. L463: Can you provide any reference to the kamada_kawai layout algorithm

Response to 2.36

Yes, of course. In view of this comment, we have added this reference.

Reference no.61: Pospisil, L., Hasal, M., Nowakova, J. & Platos, J. Computation of Kamada-Kawai Algorithm Using Barzilai-Borwein Method. in 327–333 (IEEE, 2015).

Comment 2.37

37. L463: Consider rephrasing “And then” to “Subsequently”

Response to 2.37

Thank you for noticing this. We have corrected the text accordingly.

Comment 2.38

38. L494: Please correct space with small “s”

Response to 2.38

Thank you for noticing this. We have corrected the text accordingly.

Comment 2.39

39. L524-5: Please indicate that you refer to 98 “global” indicators (not targets) available, since the dataset contains additional (e.g. regional) data as well. Furthermore, you could reflect a bit more on the quality and potential limitation of the SDR data. Despite efforts to include data from a range of sources, there may still be gaps in coverage, particularly for certain regions or SDG targets. Uneven availability of data across countries and sectors could result in skewed representations of SDG performance, limiting the generalizability of the findings.

Response to 2.39

Thank you for this suggestion. Following this comment we have modified the text as follows.

Lines 553-556:

Lastly, disparities in data availability across countries and economic sectors could lead to biased representations of SDG performance, limiting the generalizability of the findings. With increasing calls to improve the SDG database^{62,63}, future studies should critically and comparatively investigate SDG indicators and the pathways toward their realization.

We also introduced the advantages and disadvantages of the SDR database.

Lines 690-698:

In addition, the related SDG Index methodology and datasets have undergone multiple peer reviews and have been used to substantiate previous notable studies in this field^{6,66-68}. Naturally, the dataset has its advantages and disadvantages. The advantages include its broad coverage and consistency over time, which allow for meaningful comparisons across countries and time periods. The disadvantages include a lack of granularity needed to capture local (e.g., urban level) variations and specific economic activities (e.g., industries) within countries. Additionally, some indicators are not fully applicable to all countries (e.g., 14.C. *Mean area that is protected in marine sites important to biodiversity*).

Comment 2.40

40. Supplementary material, S87, Fig 9: “c” for Figure 9c. should be written in bold

Response to 2.40

Thank you for noticing this. We have corrected the text accordingly.

Concluding remark

Thank you once again for all your invaluable suggestions, which have significantly enhanced the quality of our paper.

Reviewer #3

General Remarks of Reviewer #3

The paper is well written and the authors invested a lot of efforts in explaining the methodology and the potential value of its output by providing valuable orientation for countries to find out to what extent they are on track in regard to the UN SDGs to be achieved by 2030.

It is argued that so far country trajectories in the pursuit of the UN Sustainable Development Goals have been poorly explored because the methodology to do so was largely missing. The authors propose to change this by making use of the methodology developed to capture a country's degree economic complexity (EC). Its product space method employs network analysis to reveal the relatedness, or affinity, between economies and applies it to cluster the measured targets of the different UN SDGs highlighting also trajectories of specialization patterns of regions and nations. The resulting 'SDG space' reveals the sustainable development trajectories of 166 nations for the period 2000 to 2022. The SDG space then allows to identify 'orphaned' areas with insufficient progress while highlighting future challenges in realizing SDGs.

Response to 3

We acknowledge this summary of the content of our manuscript, and overall support of our aims, and aspired contribution to literature and practice.

We further thank you for your comments and suggestions. Please see our response to your comments in the following.

Comment 3.1

In their mapping and analysis of country-based UN SDG performance and evolution, the authors largely rely on data collected by numerous international institutions to assess if countries are on track in regard to the numerous targets for each of the 17 UN SDGs. In this context, they heavily rely on the 2023 Sustainable Development Report that indexed and evaluated the evolving country performance in regard to the UN SDGs and its targets. Based on secondary macro data analysis, they have identified and visualized 3,818 SDG spaces for 166 nations spanning the years 2000 to 2022 (<http://www.spacelab.team/#/SDGSpace>) and found underlying structures that largely mirror the fact the low income countries and high income countries face different challenges in meeting the different UN SDGs.

Response to 3.1

We acknowledge this summary of the analytical approach, and findings of our study, as well noticing the dedicated open-access web-interface.

Comment 3.2

I wonder however to what extent it makes sense to talk of 'comparative advantage' of developing countries (based on RCA Assessment) in meeting certain SDG Goals and Targets that largely mirror the fact that they are low income countries with a lower environment footprint and lower health risk exposure related to problems that largely affect affluent countries (e.g. obesity). None of these countries probably have an interest in remaining 'competitive' with regard to these goals and targets because they largely reflect the lack of ability to cover the essential material needs of the people. This is also reflected in the obvious fact that they perform poorly with regard to UN SDG 1,2,4, 8 and 9 that improve with economic development. So talking of a comparative advantage may be misleading.

Response to 3.2

Thank you for your comment. We acknowledge this issue regarding the use of Revealed Comparative Advantage (RCA) to discuss developing countries' performance on certain SDG Goals and Targets.

This performance is referred to in the Product Space (our analytical approach) as an RCA. It does not, however, imply that countries have intentionally, deliberately, pursued this RCA, nor that the historical reasons and circumstances underpinning it are so-called 'advantageous'.

In this regard, please see our replies to Comments 2.5 and 2.7, above, and corresponding revisions in the main text.

Moreover, while it is true, based on historical experience, that social and environmental indicators *may* improve with economic development, the latter *may also* cause significant environmental harm (e.g., pollution), or widen inequality.

To restate our reply to Comments 2.5 and 2.7 above, the exploration of the underpinnings, or drivers, of these RCAs (or specializations, as they are referred to in Economic Complexity and the Product Space), lie outside the scope of the current study, and can be debated at length.

Regarding the use of RCA, there are several benefits, so to speak, to use a relative (comparative) analytical approach to measure development, rather than using absolute numbers. First, it allows for the identification of unique strengths and weaknesses specific to each country, which may facilitate more tailored and effective policy interventions in the future. Second, it highlights disparities and inequities between countries, drawing attention to areas needing urgent policy and/or international support. Third, it enables tracking progress over time relative to other countries, which can be more meaningful for policy evaluation and improvement.

Finally, this approach helps in understanding the contextual differences that absolute measures might overlook, providing a more nuanced perspective on development.

All this adds to our ability to identify ‘orphan’ indicators — goals that may be overlooked during different stages of a country's development process — in order to address development shortcomings.

Having said that, we acknowledge that talking about an ‘advantage’ in meeting certain SDG Goals and Indicators can be misleading, especially when these advantages may reflect low-income status rather than intentional policies or capabilities. Indeed, many of these indicators, such as lower environmental footprints and reduced health risks like obesity, may be a consequence of limited resources. Still, this does not render them *apriori*, *in and of themselves*, disadvantages (low ecological footprints is a so-called good attribute, it's underpinnings notwithstanding). Indeed, this is a complex territory.

Reflecting on this reply, this is exactly the sort of discussions we so hope to stir and support with our study.

To be clear, our intention is not to suggest that developing countries (or any country for this matter) should maintain their current status to keep these 'advantages'. Instead, our goal is to use the RCA framework to uncover areas where targeted interventions can help improve development outcomes without compromising social and environmental well-being.

In view of this Comment, and Comments 2.5 and 2.7, we have added a subsection entitled "Limitations and clarifications on the Product Space terminology", to acknowledge the data limitations, clarify the use of the Product Space techniques and terminology, and discuss their implications. Please refer to our response to Comment 2.5 above, and to lines 503-555 in the revised manuscript.

Thank you for your feedback, which has helped us clarify and refine our approach.

In addition to these revisions, we have added three additional countries as case studies in the Discussion Section to further explore these issues. Please see the newly designed Fig. 8 below.

We remain bounded by the Article format (and wordcount) limitations, yet hope to inspire and inform such future discussions.

Lines 418-450:

In terms of policy guidance, the comparative insights and identified trends in sustainable development trajectories can provide targeted recommendations (Fig. 8). From a static perspective, countries' performance on the indicators can be categorized into four areas (Fig. 8b, d, f, h): Area I (high RCA and SDG Scores) represents an ideal situation. Area II (high RCA but low SDG scores) where indicators are relatively difficult for a specific country to improve, as other countries are also likely to struggle in making progress and providing successful examples. Area III (low RCA and SDG Scores) represents the worst situation, where both relative and absolute performance on these indicators is poor. Area IV (low RCA but relatively high SDG scores) where indicators may be easier to improve, as other countries might be performing better and could offer models for success.

Taking the USA as an example (Fig. 8h), beyond the indicators in Area III — such as overnutrition (2.B, 2.F) and waste and emissions (12.A, 12.C, 12.E, 13.B) — focus should also be placed on addressing social and environmental impacts linked to imports (8.D, 12.F, 14.A), inequality (10.A), and Corporate Tax Haven (17.A) in Area IV, despite these indicators having absolute scores above the global average.

From a dynamic viewpoint, the revealed patterns of trajectories can help forewarn potential challenges for countries developing from low to high sustainable development stage (Fig. 8a, c, e, g, Fig. 5e and Supplementary Tables 18, 20, 22). Taking China as an example, as shown in red ∇ in Fig.8, China would face challenges in realizing indicators 13.B (*CO₂ emissions from fossil fuel combustion and cement production*), 14.F (*Fish caught by trawling or dredging*), and 17.A (*Corporate Tax Haven*), in which the country has already shown comparative weakness. Attention should also be given to indicators regarding overnutrition (2.B, 2.F), in which China has relative strength but may lose it in the future (as forewarned by Fig. 5e). A recent study provided evidence that the prevalence of obesity of adults in China is more than doubled between 2004 (3.1%) and 2018 (8.1%)⁵³. It is notable that indicators related to overnutrition (2.B, 2.F) are some of the main ‘orphaned’ indicators in the SDG framework, not only for high-income countries, but also for low-income ones⁵⁴.

These cases illustrate how the structured analysis of nations’ sustainable development trajectories can provide insights for shaping development policies. However, it is imperative to view these patterns not as deterministic ‘destinies’ (or fates) but rather as early warnings that can guide preventative strategies. This recognition underscores that countries, even at similar stages of development, may exhibit distinct approaches to specific SDG indicators, particularly those that are prone to being overlooked (Fig.7).

Fig. 8 | The SDG space and indicator performance for four representative countries in 2022. Panels **a-b**, Ethiopia (Stage 1). Panels **c-d**, India (Stage 2). Panels **e-f**, China (Stage 3). Panels **g-h**, United States (Stage 4). In panels **a-h**, blue nodes represent the top 20 SDG indicators by RCA values, while red nodes represent the bottom 20. The remaining indicators are not highlighted. The size of each node corresponds to the absolute RCA value. Countries are evenly divided into four stages according to their absolute score rankings: Stage 1 (Primary stage), Stage 2 (Low medium stage), Stage 3 (High medium stage), and Stage 4 (Advanced stage). Stage 4 is further divided into two halves to identify the top 20 increasing and decreasing

indicators in RCA for the USA (in the first half of Stage 4) between its current stage and the next stage (see Supplementary Table 1). In panels **b, d, f, h**, scatter diagrams are divided into four areas by the lines at $RCA = 0$ and the global average SDG score of 63. Area I: Countries with high RCA and SDG Scores for these indicators have reached a relatively ideal situation. Area II: Countries with high RCA but low SDG scores for these indicators face relative difficulties in improving them, likely due to low global averages for these indicators. Area III: Countries with low RCA and SDG Scores for these indicators show both relative and absolute underperformance, indicating that these indicators should be prioritized for improvement. Area IV: Countries with low RCA but relatively high SDG scores for these indicators demonstrate poor relative performance but strong absolute performance.

Comment 3.3

Another weakness is the relatively uncritical embrace of the secondary data gathered on the macro-level by international institutions. They tend to start from the misleading baseline assumption (reflected in many UN Reports on the SDGs) that state bureaucracies are in a position to enable a country to meet the UN SDGs and its targets. The state may provide well-designed policies to incentivize innovation and entrepreneurship moving into a publicly desirable direction; but, ultimately, inclusive and sustainable change is not enabled by regulation and protest but investments in the real economy. UN SDG 8 and 9 emphasize this point, but none of the indicators used to measure its development-related targets actually care about entrepreneurship on the ground. This is a major weakness of gathering data only on the macro-level because they do not allow for any differentiation within a country (emergence of vibrant and innovative economic ecosystems in unexpected places). There is a lack of references on these controversial aspects behind the official UN SDGs rhetoric.

Response to 3.3

Thank you for this comment.

We acknowledge the importance of investments in the “real economy” in achieving the SDGs, along with well-designed, evidence-based policies, among other areas, in incentivizing innovation.

Investments in the “real economy” refers to financial resources directed towards tangible economic activities and infrastructure that directly contribute to production, employment, and economic growth. This includes investments in industries such as manufacturing, agriculture, services, and infrastructure projects like transportation, energy, and telecommunications. These investments are crucial for driving sustainable development, fostering innovation, and creating inclusive economic opportunities.

Based on your comment, we have added the important role of investments in achieving the SDGs in the Discussion Section as follows:

Lines 494-503:

In addition, local investments in tangible economic activities and infrastructure — that is the ‘real economy’ — that directly contribute to production, employment, and economic growth are essential for promoting sustainable development. These include the allocation of financial resources to industries such as manufacturing, agriculture, services, and infrastructure projects like transportation, renewable energy, and advanced telecommunications. Such local investments are crucial for fostering innovation and creating inclusive economic opportunities. However, the relevant SDG targets and indicators in these areas, particularly those under UN SDGs 8 and 9, are ill-defined. Notably, none of the indicators used to measure these development-related targets adequately address these issues, which represents a significant weakness in the SDG framework. This issue warrants further attention.

Regarding the use of secondary data, we have chosen the 2023 Sustainable Development Report (SDR) as our data source because it provides the most comprehensive and comparable datasets available. The SDR offers time series data for the SDG Index, calculated retroactively using consistent indicators and methods, and covers more than 160 countries worldwide.

To the best of our knowledge, this database is one of the few that provides long-term, comparable SDG indicators, which is crucial for constructing the SDG space and performing robust trend analysis.

Moreover, the SDR dataset has been widely used, and repeatedly scrutinized and reviewed, in numerous studies published in high-impact journals, including *Nature*⁶⁴, *Nature Geoscience*⁶⁵, *Dublin University Press*⁶, and *Cambridge University Press*⁶³.

At the same time, we recognize that the dataset has its own advantages and disadvantages. The advantages include its broad coverage and consistency over time, which allow for meaningful comparisons across countries and time periods. The disadvantages include the lack of granularity needed to capture local variations and emerging economic ecosystems within countries.

In the revised manuscript, we have acknowledged these limitations as follows:

Lines 181-184:

It is notable that the findings and insights derived from this method may not be universally applicable to all countries in each region, given the local variations in socio-economic conditions, culture, governance, and policy environments.

In view of this Comment, and Comments 2.5, 2.7 and 3.2, we have added a subsection entitled "Limitations and clarifications on the Product Space terminology", to acknowledge the data limitations, clarify the use of the Product Space techniques and terminology, and discuss their implications.

Lines 504-556:

Limitations and clarifications on the Product Space terminology

In this study, we constructed ‘SDG spaces’ for 166 nations from 2000 to 2022, revealing a bipolar world while identifying ‘orphaned’ development indicators by different nations. These

patterns, along with the SDG space, provide a high-resolution tool for evaluating progress and offering targeted guidance for countries in achieving the SDGs. In doing so, we have applied the Product Space and Economic Complexity framework, analytical techniques, and terminology, including ‘revealed comparative advantage’ (RCA) and ‘specializations’.

We acknowledge that some of the sustainable development indicators for which certain countries exhibit an RCA (a specialization) might reflect an intentional, premeditated national preference over time, such as the environmental impacts of imports, as part of a larger importation policy. While our data allow us to identify which advantages — or, as they are also referred to in this study (and in Economic Complexity) ‘specializations’ — are likely to diminish as a country develops, they do not enable us to differentiate between intentional (i.e., preferred, premeditated) and unintentional specializations. Indeed, ‘unintentional specializations’ may arise due to a country’s geography or geology, such as being landlocked or possessing mineral resource reserves⁶⁰. Additionally, other detrimental factors, such as colonial legacy, may predetermine a country’s development trajectory⁶⁰.

To clarify, the use of RCA to discuss developing countries’ performance on certain SDG indicators is done within the context of the Product Space approach. However, it does not imply that these countries have intentionally pursued this RCA, nor does it suggest that the historical reasons and circumstances underpinning it are inherently ‘advantageous’. Using the terms ‘advantage’ or ‘specialization’ outside the context of the Product Space approach — when referring to a country meeting certain SDG targets — may be misleading, especially when these ‘advantages’ may reflect low-income status rather than deliberate policies or capabilities. For example, several indicators, such as lower environmental footprints and reduced health risks like obesity, may result from limited resources. At the same time, this does not inherently render them disadvantages — a low ecological footprint can be considered a positive attribute, regardless of its underlying causes. This invites further discussion and underscores the need for future research.

Moreover, our intention is not to suggest that so-called developing countries (or any country, for that matter) should maintain their current status to preserve these ‘advantages’ (RCAs). Instead, the aim of this study is to use the RCA framework to identify national sustainable development trajectories as well as areas where targeted interventions are needed to enhance development outcomes.

In the same vein, the degree of agency we can attribute to national governments in such circumstances (e.g., ‘being landlocked’, ‘being resource-endowed’, and ‘being post-colonial’) when developing a relative specialization remains a matter of debate. Addressing the distinction between intentional and unintentional specializations, as well as the role of national governments’ agency, requires further study and may involve subjective judgments. While this study opens new avenues for such discussions, the interpretation of RCAs as normative preferences lies outside its scope.

On a related note, it should be emphasized that local variations at subnational levels and units of analysis (regional, provincial, urban, peri-urban, rural) in socio-economic conditions, culture, governance, policy environments, and institutional capacities are important factors that may

not be fully captured by our aggregated data and global trends. We acknowledge that even within the same country, SDG performance and strategies may differ between regions and between urban and rural areas⁶¹. Due to data limitations, this study focuses on global and national sustainable development through 96 SDG indicators. Future research could extend this analysis to regional, state, and city levels.

Lastly, disparities in data availability across countries and economic sectors could lead to biased representations of SDG performance, limiting the generalizability of the findings. With increasing calls to improve the SDG database^{62,63}, future studies should critically and comparatively investigate SDG indicators and the pathways toward their realization.

In addition, to validate our results, we have added a sub-section of Robustness check, for example, we compared our results with the newly calculated results using the UN-SDG dataset. Please also see our reply to Comment 1.10 above.

Lines 708-739, Robustness check Section of the Methods:

To ensure the robustness of our findings, we evaluated the results from the following perspectives: (1) whether a consistent SDG space structure can be reproduced using data from different years; (2) whether the SDG space structure remains similar (i.e., stable) after excluding overlapping indicators; (3) whether the structure is consistent across different data sources; (4) whether the bipolar world results hold under varying similarity thresholds; and (5) whether consistent evolution patterns of SDG indicators emerge with different moving window sizes.

(1) Supplementary Fig. 3 demonstrates that a similar SDG space structure is observed across the years of 2000, 2015, and 2022. This consistency indicates that our finding of a bipolar world within the SDG space is robust and not influenced by the choice of years.

(2) Notably, some SDG indicators may overlap, such as 2.B (*Prevalence of obesity*) and 2.F (*Human Trophic Level*) in this study. To ensure the robustness of our results, we removed the overlapping indicators, retaining only one unique indicator, and re-examined the patterns (Supplementary Fig. 10). The findings remained robust, confirming the consistency of our analysis.

(3) To validate the robustness of our results using different data sources, we collected additional data from the UNSD SDG database. For consistency in comparison, we included only the data on targets that matched the 96 indicators used in this study (see Tables 1-2 in Methods for the matching process). Data source: <https://unstats.un.org/sdgs/dataportal/database>. As shown in Fig. 1 (results of SDR 2023 database) and Supplementary Fig. 11 (results of UNSD SDG database), the SDG spaces exhibit a similar structure.

(4) In this study, we generated the SDG spaces by setting the similarity threshold between SDG indicators at 0.7. We tested the robustness of our results by varying the similarity values. As seen in Supplementary Fig. 9, the indicator clusters remain consistent, supporting the SDG space structure.

(5) We also examined whether varying the moving-window sizes would yield similar evolution

patterns of SDG indicators. Supplementary Fig. 7 shows that the evolution patterns of SDG indicators and the ranking of countries' SDG scores in 2022 remain consistent across moving-window sizes of 20, 30, 40, 50, 60, and 70.

Overall, our findings demonstrate robustness across various tests and conditions.

Supplementary Figs. 3, 7, 9-11 in the Supplementary Materials are as follows:

Supplementary Figure 3 | The evolution of SDG space. Panels **a**, **c**, and **e** show the global SDG space in 2000, 2015, and 2022, respectively. Panels **b**, **d**, and **f** display the similarity of SDG indicators in 2000, 2015, and 2022, respectively.

Supplementary Figure 7 | The evolution of SDG indicators alongside the ranking of countries' SDG scores in 2022 with varying moving-window sizes. Panels a, b, c, d, e, and f display the results using moving-window sizes of 20, 30, 40, 50, 60, and 70, respectively.

Supplementary Figure 9 | Sensitivity analysis of SDG indicators' similarity in 2022. Panel **a**, SDG indicators with similarity ≥ 0.6 . Panel **b**, SDG indicators with similarity ≥ 0.7 . Panel **c**, SDG indicators with similarity ≥ 0.8 .

Supplementary Figure 10 | The SDG space in 2022 with non-overlapping indicators. We screened six groups of similar indicators: (1.A and 1.B), (2.B and 2.F), (2.E and 2.H), (3.G and 3.L), (10.A and 10.B), and (14.D, 14.E, and 14.F). For the robustness check, we retained only the non-overlapping indicators within these groups. The node colour represents Goal Sustainability Index (GSI), and the node size corresponds to the node degree, indicating the number of edges connected to the node.

Supplementary Figure 11 | The SDG space in 2022 using UN SDG data. The node colour represents Goal Sustainability Index (GSI), and the node size indicates the node degree, representing the number of edges connected to the node. This SDG space was produced using data from the UNSD SDG targets framework, which were matched with the 96 SDG indicators (see Tables 1-2 in Methods). Data source: <https://unstats.un.org/sdgs/dataportal/database>.

Comment 3.4

Nevertheless, applying the EC methodology to do a mapping that shows the trajectories of countries in their pursuit of the UN SDGs and its targets may help provide valuable orientation for policy makers by making it transparent that the main enemy of sustainability in high income countries is affluence (leading to overuse of natural resources, large greenhouse gas emissions and material waste) whereas it is poverty in low income countries (lack of ability to provide the right to food, health, education, employment, etc as well as depletion of natural resources and political instability due to population growth and lack of economic opportunities). This should also be made more explicit in the paper.

Response to 3.4

Thank you for your helpful suggestion and for acknowledging our contribution in using the economic complexity and product space method to map the trajectories of countries in their

pursuit of the SDGs, providing valuable orientation for policymakers.

Following this comment, we have added a relevant discussion on the explanations of SDG development patterns in the Discussion Section, as follows:

Lines 383-389:

Additionally, scientists have warned that global increases in affluence have consistently driven resource use and pollutant emissions to rise more rapidly than technological improvements have been able to mitigate. These impacts are primarily driven by affluent citizens⁴⁹. Therefore, it is imperative to achieve absolute decoupling of economic growth from resource consumption and pollutant emissions through various approaches, such as technological advancements, shifts in consumption patterns, and the adoption of more effective policies⁵⁰.

Comment 3.5

The methodology is surely sound but the authors need to drop the 'comparative advantage' language and they will have to highlight the limits of the approach in view of the exclusive reliance on macro-level data. Finally, the paper does not contain any concluding remarks that describe major findings as well as weaknesses that may have to be addressed in future research.

Response to 3.5

Thank you for the support of the analytical approach.

To underscore the limitations of the data used, we have modified the text as follows:

Lines 690-698:

In addition, the related SDG Index methodology and datasets have undergone multiple peer reviews and have been used to substantiate previous notable studies in this field^{6,66-68}. Naturally, the dataset has its advantages and disadvantages. The advantages include its broad coverage and consistency over time, which allow for meaningful comparisons across countries and time periods. The disadvantages include a lack of granularity needed to capture local (e.g., urban level) variations and specific economic activities (e.g., industries) within countries. Additionally, some indicators are not fully applicable to all countries (e.g., 14.C. *Mean area that is protected in marine sites important to biodiversity*).

To underscore the key findings along with weaknesses, we have added a subsection of '*Limitations and clarifications on the Product Space terminology*' as follows:

Lines 504-556:

Limitations and clarifications on the Product Space terminology

In this study, we constructed 'SDG spaces' for 166 nations from 2000 to 2022, revealing a bipolar world while identifying 'orphaned' development indicators by different nations. These patterns, along with the SDG space, provide a high-resolution tool for evaluating progress and

offering targeted guidance for countries in achieving the SDGs. In doing so, we have applied the Product Space and Economic Complexity framework, analytical techniques, and terminology, including ‘revealed comparative advantage’ (RCA) and ‘specializations’.

We acknowledge that some of the sustainable development indicators for which certain countries exhibit an RCA (a specialization) might reflect an intentional, premeditated national preference over time, such as the environmental impacts of imports, as part of a larger importation policy. While our data allow us to identify which advantages — or, as they are also referred to in this study (and in Economic Complexity) ‘specializations’ — are likely to diminish as a country develops, they do not enable us to differentiate between intentional (i.e., preferred, premeditated) and unintentional specializations. Indeed, ‘unintentional specializations’ may arise due to a country’s geography or geology, such as being landlocked or possessing mineral resource reserves⁶⁰. Additionally, other detrimental factors, such as colonial legacy, may predetermine a country’s development trajectory⁶⁰.

To clarify, the use of RCA to discuss developing countries’ performance on certain SDG indicators is done within the context of the Product Space approach. However, it does not imply that these countries have intentionally pursued this RCA, nor does it suggest that the historical reasons and circumstances underpinning it are inherently ‘advantageous’. Using the terms ‘advantage’ or ‘specialization’ outside the context of the Product Space approach – when referring to a country meeting certain SDG targets – may be misleading, especially when these ‘advantages’ may reflect low-income status rather than deliberate policies or capabilities. For example, several indicators, such as lower environmental footprints and reduced health risks like obesity, may result from limited resources. At the same time, this does not inherently render them disadvantages — a low ecological footprint can be considered a positive attribute, regardless of its underlying causes. This invites further discussion and underscores the need for future research.

Moreover, our intention is not to suggest that so-called developing countries (or any country, for that matter) should maintain their current status to preserve these ‘advantages’ (RCAs). Instead, the aim of this study is to use the RCA framework to identify national sustainable development trajectories as well as areas where targeted interventions are needed to enhance development outcomes.

In the same vein, the degree of agency we can attribute to national governments in such circumstances (e.g., ‘being landlocked’, ‘being resource-endowed’, and ‘being post-colonial’) when developing a relative specialization remains a matter of debate. Addressing the distinction between intentional and unintentional specializations, as well as the role of national governments’ agency, requires further study and may involve subjective judgments. While this study opens new avenues for such discussions, the interpretation of RCAs as normative preferences lies outside its scope.

On a related note, it should be emphasized that local variations at subnational levels and units of analysis (regional, provincial, urban, peri-urban, rural) in socio-economic conditions, culture, governance, policy environments, and institutional capacities are important factors that may not be fully captured by our aggregated data and global trends. We acknowledge that even

within the same country, SDG performance and strategies may differ between regions and between urban and rural areas⁶¹. Due to data limitations, this study focuses on global and national sustainable development through 96 SDG indicators. Future research could extend this analysis to regional, state, and city levels.

Lastly, disparities in data availability across countries and economic sectors could lead to biased representations of SDG performance, limiting the generalizability of the findings. With increasing calls to improve the SDG database^{62,63}, future studies should critically and comparatively investigate SDG indicators and the pathways toward their realization.

The relative comparative advantage is integral to the analytical approach and methodology (one of the main innovations of this study). Please, kindly refer to our replies to comments 1.3, 2.5, 2.7, 3.2, above.

Concluding remark

Thank you once again for all your invaluable suggestions, which have significantly enhanced the quality of our paper.

REVIEWERS' COMMENTS

Reviewer #1 (Remarks to the Author):

The authors have satisfactorily responded to my most of my questions and suggestions. I think the paper can make useful contributions to the literature. I would recommend it for publication conditional on some final, minor comments below.

1. For completeness, the authors should provide concrete policy suggestions for all the four countries shown in Figure, using four groups they construct and show in Figure 8. These include Ethiopia, China, India and the US. Currently, they only discuss the US and China.

2. The 4 countries in Figure 8 have their indicators mostly falling into two groups, I and III. Does this result similarly hold for other countries, given the dominant bipolar pattern revealed in this study? If so, it is useful to note. Otherwise, it would be more illustrative to show other country examples with their indicators falling more into groups II and IV for richer analysis.

3. Finally, while the authors have provided clearer discussion on the main data source that they analyze and some other data sources, it can be useful to provide a more nuanced note on data quality issues. Since countries are at different statistical capacity levels, the quality of the data they produce (and provide to international organizations) can vary. Furthermore, there is also an aspect of political economy with data generation and comparability. See, for example, Dargent et al (2018) and Dang et al (2023). These can be discussed as caveats or limitations that should be explored in further research.

References

Dargent, E., Lotta, G., Mejía-Guerra, J. A., & Moncada, G. (2018). *The Political Economy of Statistical Capacity in Latin America: Who Wants to Know?*. Inter-American Development Bank.

Dang, H. A. H., Pullinger, J., Serajuddin, U., & Stacy, B. (2023). Statistical performance indicators and index—a new tool to measure country statistical capacity. *Scientific Data*, 10(1), 146.

Reviewer #2 (Remarks to the Author):

I thank the authors for their thoughtful consideration of my comments and suggestions. I am pleased to see that my concerns have been addressed thoroughly and satisfactorily.

Reviewer #3 (Remarks to the Author):

The response of the authors is very detailed and I appreciate that the concerns raised have been taken into account to the greatest extent possible in the revised manuscript. The publication will be of great significance because it highlights the potential of the Product Space approach and techniques to reveal useful patterns underpinning the sustainable development trajectories of nations.

Response to Referees for Nature Communications manuscript NCOMMS-24-21670A, The Sustainable Development Trajectories of Nations (original article title, now revised)

Acknowledgements

We are grateful to the anonymous reviewers and editors for their recognition of this study's contributions and for their invaluable feedback. In particular, we would like to thank Reviewer #1 for their specific and constructive comments, which have helped improve key aspects of the article. We also appreciate Reviewers #2 and #3 for their satisfaction with our previous responses and their continued acknowledgement of the work. Below, we provide detailed responses to Reviewer #1's comments and have made corresponding revisions in the manuscript.

Reviewer #1

General Remarks of Reviewer #1

The authors have satisfactorily responded to my most of my questions and suggestions. I think the paper can make useful contributions to the literature. I would recommend it for publication conditional on some final, minor comments below.

Response to 1

We are pleased to note your satisfaction with our previous response and deeply appreciate your thoughtful and detailed comments. These specific suggestions will further strengthen the rigor of our analysis and enhance the policy relevance of our study's findings.

Comment 1.1

1. For completeness, the authors should provide concrete policy suggestions for all the four countries shown in Figure 8, using four groups they construct and show in Figure 8. These include Ethiopia, China, India and the US. Currently, they only discuss the US and China.

Response to 1.1

We fully agree with your insightful comment. Including concrete policy suggestions for Ethiopia and India will indeed enrich our discussion and provide a more comprehensive perspective. In line with your recommendation, we have incorporated the relevant discussion as follows:

Lines XX:

For Ethiopia, a representative country at Stage 1 (Primary stage), priority should be given to addressing indicators with the lowest SDG scores and Revealed Comparative Advantage (RCA) in Area III (Fig. 8b), such as 6.B (Population using at least basic sanitation services) and 6.D (Anthropogenic wastewater that receives treatment). For India, in addition to the indicators in Area III (Fig. 8d), attention should be directed towards biodiversity-related indicators, such as 15.C (Mean area that is protected in terrestrial sites important to biodiversity) and 15.E (Red List Index of species survival). India exhibits comparative weaknesses in these two indicators, and as it transitions to the next stage, it is likely to face greater challenges in these areas (see the red \square in Fig. 8c).

Comment 1.2

2. The 4 countries in Figure 8 have their indicators mostly falling into two groups, I and III. Does this result similarly hold for other countries, given the dominant bipolar pattern revealed in this study? If so, it is useful to note. Otherwise, it would be more illustrative to show other country examples with their indicators falling more into groups II and IV for richer analysis.

Response to 1.2

Thank you for this valuable comment. We apologize for not explaining this more clearly in the original submission. The reason most indicators for these four countries fall into two groups is that, for clarity and to provide more focused recommendations, we only displayed the top and bottom 20 SDG indicators based on RCA values in Fig. 8. To clarify this point, we have added the following note to the caption of Fig. 8.

Lines XX:

Note: Fig. 8 displays only the top and bottom 20 SDG indicators based on RCA values; the remaining indicators are not highlighted.

Comment 1.3

3. Finally, while the authors have provided clearer discussion on the main data source that they analyze and some other data sources, it can be useful to provide a more nuanced note on data quality issues. Since countries are at different statistical capacity levels, the quality of the data they produce (and provide to international organizations) can vary. Furthermore, there is also an aspect of political economy with data generation and comparability. See, for example, Dargent et al (2018) and Dang et al (2023). These can be discussed as caveats or limitations that should be explored in further research.

References

Dargent, E., Lotta, G., Mejía-Guerra, J. A., & Moncada, G. (2018). The Political Economy of Statistical Capacity in Latin America: Who Wants to Know?. *Inter-American Development*

Bank.

Dang, H. A. H., Pullinger, J., Serajuddin, U., & Stacy, B. (2023). Statistical performance indicators and index—a new tool to measure country statistical capacity. *Scientific Data*, 10(1), 146.

Response to 1.3

Thank you for your constructive suggestion. In line with your recommendation, we have expanded the discussion on data quality issues and highlighted that data generation and comparability can be influenced by various political and economic factors.

The detailed revisions are as follows:

Lines XX:

Lastly, due to the difference in political and economic conditions^{62,63}, disparities in data availability and quality across countries may result in biased representations of SDG performance, limiting the generalizability and comparability of our findings. With growing calls to enhance the SDG database^{64,65}, future studies should critically and comparatively examine SDG indicators and the pathways toward their realization.

Reviewer #2

General Remarks of Reviewer #2

I thank the authors for their thoughtful consideration of my comments and suggestions. I am pleased to see that my concerns have been addressed thoroughly and satisfactorily.

Response to 2

We are pleased to hear that you are satisfied with our responses and revisions. We greatly appreciate your constructive and detailed comments, which have significantly contributed to elevating the quality of the manuscript.

Reviewer #3

General Remarks of Reviewer #3u

The response of the authors is very detailed and I appreciate that the concerns raised have been taken into account to the greatest extent possible in the revised manuscript. The publication

will be of great significance because it highlights the potential of the Product Space approach and techniques to reveal useful patterns underpinning the sustainable development trajectories of nations.

Response to 3

We are pleased to learn that you are satisfied with our responses and revisions. We sincerely appreciate your insightful comments and your recognition of the key contributions made by this study.